# A Review on Gallium Oxide Materials from Solution Processes

**DOI:** 10.3390/nano12203601

**Published:** 2022-10-14

**Authors:** Jung-Lung Chiang, Bharath Kumar Yadlapalli, Mu-I Chen, Dong-Sing Wuu

**Affiliations:** 1Ph.D. Program, Prospective Technology of Electrical Engineering and Computer Science, National Chin-Yi University of Technology, Taichung 41170, Taiwan; 2Department of Materials Science and Engineering, National Chung Hsing University, Taichung 40227, Taiwan; 3Department of Applied Materials and Optoelectronic Engineering, National Chi Nan University, Nantou 54561, Taiwan; 4Innovation and Development Center of Sustainable Agriculture, National Chung Hsing University, Taichung 40227, Taiwan

**Keywords:** gallium oxide, sol-gel, hydrothermal, chemical bath deposition, solvothermal, forced hydrolysis, reflux condensation, electrochemical deposition, photodetector, sensor

## Abstract

Gallium oxide (Ga_2_O_3_) materials can be fabricated via various methods or processes. It is often mentioned that it possesses different polymorphs (α-, β-, γ-, δ- and ε-Ga_2_O_3_) and excellent physical and chemical properties. The basic properties, crystalline structure, band gap, density of states, and other properties of Ga_2_O_3_ will be discussed in this article. This article extensively discusses synthesis of pure Ga_2_O_3_, co-doped Ga_2_O_3_ and Ga_2_O_3_-metal oxide composite and Ga_2_O_3_/metal oxide heterostructure nanomaterials via solution-based methods mainly sol-gel, hydrothermal, chemical bath methods, solvothermal, forced hydrolysis, reflux condensation, and electrochemical deposition methods. The influence of the type of precursor solution and the synthesis conditions on the morphology, size, and properties of final products is thoroughly described. Furthermore, the applications of Ga_2_O_3_ will be introduced and discussed from these solution processes, such as deep ultraviolet photodetector, gas sensors, pH sensors, photocatalytic and photodegradation, and other applications. In addition, research progress and future outlook are identified.

## 1. Introduction

Gallium oxide (Ga_2_O_3_) has been a well-known old material for decades. The element gallium and its compounds were first discovered by a French man, P. E. Lecoq de Boisbaudran [1]. Research on Ga_2_O_3_ began from the investigation on the phase equilibria in the system Al_2_O_3_-Ga_2_O_3_-H_2_O by R. Roy et al. [2]. In 1952, he further demonstrated the existence of five types of Ga_2_O_3_ polymorphs and their stability relations [3]. By observing its optical absorption and photoconductivity, the band gap of bulk single crystals of Ga_2_O_3_ was known as 4.7 eV in 1965 [4]. Early research about Ga_2_O_3_ focused on its basic properties, such as crystal structures, band gap, density of states of electrons, and so on. This part will be described in detail in the next chapter. The crystal quality of Ga_2_O_3_ materials in most scientific reports published in the range of 1960s~1980s was poor or even in amorphous forms [5]. That is why Ga_2_O_3_ had been ignored by most semiconductor researchers and engineers, resulting in its development lagging behind SiC and GaN.

In the 1990s, various methods for growing high-quality and large-sized Ga_2_O_3_ bulk single crystals were developed [5]. In the same period, high quality epitaxial thin films of Ga_2_O_3_ had also been successfully grown as essential parts of more complex devices [6]. Apart from bulk crystals and epitaxial thin films, various forms of Ga_2_O_3_ nanomaterials have made considerable progress in growth methods, basic properties, and device applications due to the rapid development in science and technology [7]. Because of nanoscale size in at least one dimension, nanomaterials have a high aspect ratio, which leads to high activity and some unique quantum-dimensional effects [8]. UV photodetectors [9,10], photocatalysis [11], flat panel display [12], UV filter [13], MOS capacitor [14], MOS structure [15] and optoelectronic devices [16,17] are some of the applications of Ga_2_O_3_.

In principle, the vapor deposition process, either physical vapor deposition (PVD) or chemical vapor deposition (CVD), is the mainstream for the fabrication of gallium oxide thin films and nanomaterials. Generally speaking, thin films made by vapor deposition have a higher uniformity and better quality. The thermal evaporation method, a kind of PVD, followed by CVD, are primarily used in the fabrication of β-Ga_2_O_3_ nanomaterials [6]. However, such processes should be carried out using vacuum equipment, which has higher maintenance costs. Additionally, excessively high temperature, complexity in process, and a limited deposition area are also problems in the vacuum process, especially the potential need for expensive catalysts in the vacuum process of Ga_2_O_3_ nanomaterials [8].

In order to reduce complexity and energy consumption during the manufacturing process, a number of wet chemical processes, also called wet chemistry approaches, which include hydrothermal, sol-gel, chemical bath deposition (CBD), and so on, are promising to synthesize Ga_2_O_3_ thin films and nanomaterials [8]. These methods can achieve large area deposition and high yield through a simple experimental process at a low temperature in an ambient atmosphere [18]. By controlling the amounts of individual precursors, it is easy to tune the final material compositions. Furthermore, these wet chemical solution processes are effective methods to prepare various kinds of nano-powders. That is why solution process has gained more acceptance in recent years [19].

Typically, Ga_2_O_3_ nanomaterials could be obtained by calcining gallium oxide hydroxide (GaOOH), which is a common intermediate product of wet chemistry approaches of gallium oxide and could be shaped with a variety of morphologies, including rod-like, spindle-like, and scroll-like cylindrical structures [20,21]. In comparison to Ga_2_O_3_, many kinds of oxides, such as ZnO, IGZO, and In_2_O_3_, have been widely manufactured by wet chemical solution processes [22]; the number of studies on the wet chemical growth of Ga_2_O_3_ is growing in recent years, as shown in Figure 1.

Therefore, this review will focus on Ga_2_O_3_ fabricated by wet chemical solution processes that include mainly three methods: sol-gel, hydrothermal, and chemical bath deposition. The basic elaboration of the three methods will be presented in the third chapter. The fourth chapter in the main topic of the review is: Ga_2_O_3_ thin films and nanomaterials synthesized by the three main kinds and four other kinds of wet chemistry approaches. The fifth chapter demonstrates the applications of Ga_2_O_3_ synthesized through wet chemical solution processes, such as deep ultraviolet photodetector, gas sensors, pH sensors, photocatalytic and photodegradation, and so on.

There have been numerous reviews on the synthesis of Ga_2_O_3_ in recent years [8,23,24]. Despite mentioning one or two solution-based methods, a thorough discussion on the solution-based synthesis of Ga_2_O_3_ was still required to understand its mechanism of synthesis and changes in properties. This review serves as a comprehensive text for a solution-based Ga_2_O_3_ synthesis.

## 2. Basic Properties of Ga_2_O_3_

Before introducing growth methods and applications of Ga_2_O_3_, some fundamental properties of different Ga_2_O_3_ phases have to be discussed, as these are key in understanding Ga_2_O_3_. Several research papers about this topic have been published through experimental or theoretical investigations [25,26,27,28,29,30,31,32]. The basic properties of Ga_2_O_3_, such as crystalline structures, band structures, and density of states of electrons will be discussed as follows.

### 2.1. Crystalline Structure of Ga_2_O_3_

In general, there are five different polymorphs labelled as α, β, γ, δ, and ε for Ga_2_O_3_ single crystal, first reported by Roy et al. in 1952 [3]. Table 1 contains a list of their lattice parameters and space groups. Among them, only β-Ga_2_O_3_ can be formed as bulk crystals directly from the melt, while the other four metastable crystalline phases can only be obtained as thin films [6]. The scientific reports focused on the material properties and crystal growth of β-Ga_2_O_3_ much more than those on the other four types because β-Ga_2_O_3_ is the most stable phase and is easily fabricated [33,34].

For the large-scale synthesis of crystalline Ga_2_O_3_ by wet chemical methods, gallium oxyhydroxide (α-GaOOH) is a common precursor, from which different gallium oxide phases (α, β, γ, δ, and ε) could be obtained through heat treatment under certain temperatures to dehydrate α-GaOOH [38]. Furthermore, various morphologies of Ga_2_O_3_ nanomaterials can also be obtained indirectly from different morphologies of α-GaOOH precursors [39] because in most cases α-GaOOH nanostructures can transform isomorphously to Ga_2_O_3_ under heat treatment [38]. Therefore, synthesis of Ga_2_O_3_ nanomaterials through thermal treatment of α-GaOOH has become an easy and convenient method [39]. In this section, the crystalline structure and morphology of α-GaOOH will be demonstrated first, followed by different polymorphs of Ga_2_O_3_.

α-GaOOH has an orthorhombic structure (space group: *Pbnm*, lattice parameters: a = 4.5545 Å, b = 9.8007 Å, and c = 2.9738 Å, as shown in Figure 2a), along the c-axis of which α-GaOOH crystals prefer to grow in this direction [40] by continuously adsorbing ${\mathrm{OH}}^{-}$ anions in the solution on the specific crystal plane (001) to facilitate nucleation and to grow into rod-like nanomaterials [8]. In addition, these rod-like GaOOH crystals usually have a prismatic shape with quadrilateral or rhombic cross-sections, which results from the outside embodiment of the stacked unit cells of the orthorhombic structure [41]. In principle, gallium(III) hydroxide Ga(OH)_3_ is the first precursor in the hydrolysis process before it is dehydrated to gallium oxide hydroxide at 100 °C [42].

In principle, a crystal face with less closely packed atoms, which means the higher density of the unsaturated bonds, has more opportunity to absorb anions [44]. Based on the crystalline structure of α-GaOOH, the (001) crystal surface has the least closely packed atoms, compared with the (010) and (100), and thus the most likely to absorb ${\mathrm{OH}}^{-}$ anions, leading to preferred growth at a higher rate along the [001] direction on the α-GaOOH amorphous particles under acidic [8] or neutral conditions [38]. Before hydrothermal treatment, spindle-shaped particles of α-GaOOH are the first resultant for fresh precipitate in an acidic and low-temperature environment [38,43]. The needle-like ends of the spindle-like particles imply faster growth in the [001] direction than the [010] and [100] directions [41]. However, when more ${\mathrm{OH}}^{-}$ anions exist in the solution, or the pH value increases, the (100) and (010) crystal surfaces have more opportunity to absorb ${\mathrm{OH}}^{-}$ anions such as the (001) crystal surface, resulting in isotropic growth to form ellipsoid-like particles composed of well-aligned nanoplatelets [41], nanorod arrays [45], or overlapping rhombi [43]. The major axes of these ellipsoid-shaped particles are along the [001] direction due to faster growth along this direction.

α-GaOOH can be easily transformed to α-Ga_2_O_3_ with morphology preserved by heating α-GaOOH in air between 450 °C and 550 °C [3] because the oxygen anions of the two crystals are both based on the hexagonal close packing (hcp) with preserved stacking sequence of layers in the direction of [100] for α-GaOOH (Figure 2a) and [001] for α-Ga_2_O_3_ (Figure 2b). Additionally, the growth direction changes from the [001] direction of α-GaOOH nanorods to the [010] direction of α-Ga_2_O_3_ nanorods during the dihydroxylation process [43]. α-Ga_2_O_3_ has a hexagonal (or rhombohedral [46]) crystal structure in the space group of *R-3c*, which is commonly called the corundum structure. Gallium ions occupy two-thirds of the octahedral sites in the corundum crystal structure. That is why the Ga^3+^ ions are much closer together than they are in the monoclinic β phase [35], which will be mentioned in the next paragraph. There are 6 Ga_2_O_3_ formula units in every crystallographic cell with lattice parameters *a* = *b* = 4.98 Å, *c* = 13.43 Å, α = β = 90°, and γ = 120° [25].

When α-GaOOH nanorods are heated to 900 °C, the cross sections change from the quadrilateral shape to diamond-like patterns, which is caused by the orthorhombic phase of α-GaOOH changing to the monoclinic phase of β-Ga_2_O_3_ [45]. Other morphologies of α-GaOOH can also be retained after this high-temperature calcination to form different morphologies of β-Ga_2_O_3_ nanomaterials except for the spindle-shaped α-GaOOH [43]. Furthermore, the monoclinic phase β-Ga_2_O_3_ has the best thermal stability up to 1800 °C [7], while the other four metastable polymorphs tend to transform back to β-Ga_2_O_3_ at high temperatures [47], as shown in Figure 3.

β-Ga_2_O_3_ has a monoclinic crystal structure (space group: *C2*/*m*, lattice parameters: *a* = 12.23 Å, *b* = 3.04 Å, *c* = 5.80 Å and a monoclinic angle: β = 103.7° [35]). There are four Ga_2_O_3_ formula units in every crystallographic cell, as shown in Figure 4, with two inequivalent Ga atoms and three inequivalent O atoms in one Ga_2_O_3_ formula unit. Tetrahedrally and octahedrally coordinated gallium atoms are designated as Ga1 and Ga2, respectively. The oxygen atoms are organized in a close-packed “distorted cubic” pattern [48]. O1 (red) stands for the oxygen atoms that share bonds with two Ga2 and one Ga1, while O3 (maroon) has two bonds with Ga1 and one bond with Ga2. O2 (pink) are primarily coupled to Ga2 by 3 bonds and connected to Ga1 with only one bond [49], as shown in Figure 4. O1 and O2 are coordinated threefold, while O3 are coordinated fourfold [6].

As α-Ga_2_O_3_ is obtained by calcining α-GaOOH, γ-Ga_2_O_3_ can be obtained by calcining Ga(OH)_3_ gel, which is the fresh precipitate before it rapidly goes through oxolation and condensation to α-GaOOH [50]. Similar to γ and *η*-Al_2_O_3_, γ-Ga_2_O_3_ possesses a faulty cubic spinel-type structure (AB_2_O_4_-type) [51], as shown in Figure 5a. It belongs to the space group of *Fd-3m* with lattice parameters *a* = *b* = *c* = 8.24 Å and α = β = γ = 90° [36]. First principle calculations were made by expanding the primitive fcc unit cell along the *c*-axis by three times [37]. For such, the unit cell of the normal spinel structure contains 6 tetrahedrally coordinated cations, 12 octahedrally coordinated cations, and 24 oxygen ions. Among the 18 cations, 2 sites were chosen to be vacant. After considering the symmetry, 14 inequivalent configurations were found [37]. Because some of the Ga ions are located on the tetrahedral sites in γ-Ga_2_O_3_, there is particular interest in this phase of Ga_2_O_3_ for possible applications in heterogeneous catalysis [52].

The last two polymorphs, named δ-Ga_2_O_3_ and ε-Ga_2_O_3_, were first discovered by Roy et al. [3] in 1952. The white powder of δ-Ga_2_O_3_ can be prepared by annealing gallium nitrate in the air at 200 °C for 18 h. Further heating the white powder at 500 °C for 6 h, the δ-Ga_2_O_3_ transforms to ε-Ga_2_O_3_ [53]. δ-Ga_2_O_3_ is a body-centered cubic crystal that belongs to the space group of *Ia3* with a unit edge length of *a* = 9.402 Å [37]. δ-Ga_2_O_3_ has a bixbyite structure analogous to that of In_2_O_3_, Mn_2_O_3_, and Ti_2_O_3_ [32,37]. However, thin films of δ-Ga_2_O_3_ have not yet been obtained. As for ε-Ga_2_O_3_, it has a hexagonal crystal structure with lattice parameters *a* = *b* = 2.90 Å, *c* = 9.26 Å, α = β = 90°, and γ = 120° [32], which is similar to κ-Al_2_O_3_ [54,55,56,57] in the space group of *P6_3_mc*, as shown in Figure 5b. This phase is next in stability among all Ga_2_O_3_ polymorphs to β-Ga_2_O_3_ [58,59,60,61]. Furthermore, density functional theory (DFT) calculation and recent experimental results indicate that ε-Ga_2_O_3_ is also an orthorhombic crystal belonging to the space group of *Pna2_1_* [37,54,62], which is an ordered subgroup of the hexagonal *P6_3_mc* [54,63].

### 2.2. Band Gap and Density of States of Electrons for Ga_2_O_3_

The fundamental electrical and optical characteristics of a material are defined by its electronic band structure. With this knowledge, we can know for what proper devices the material could be designed with desired functionalities [6]. Among all the polymorphs of Ga_2_O_3_, β-Ga_2_O_3_ is the most representative of the polymorphs of Ga_2_O_3_. Due to the difficulty in obtaining pure crystalline phases of β-Ga_2_O_3_, much knowledge about this topic comes from theoretical approaches [64,65,66]. Up to now, numerous first-principle computations employing standard density functional theory (DFT) have been used to study the basic electronic structure of β-Ga_2_O_3_ [27,64,67,68,69,70,71]. However, the standard DFT is not directly based on the excited states of electrons [68], causing an underestimated band gap. The hybrid functional [28,72,73,74,75] and GGA + U [76,77,78] approaches used in DFT were proven to provide more accurate results for the experimental band gaps of β-Ga_2_O_3_.

The full-energy band structure along with the corresponding density of states (DOS) of β-Ga_2_O_3_ are shown in Figure 6 as calculated by J. Furthmuller and F. Bechstedt [75]. From Figure 6, we can easily see that the densities of states (DOS) exhibit a direct reflection of the band structure. The maximum of the valence band (above −7.5 eV) is defined as zero energy level. The conduction band (above the zero-energy level) is mainly formed by Ga 4s orbitals. It can be easily seen from Figure 6 that the conduction band minimum (CBM) is located at the Г point independent of the different polymorph of Ga_2_O_3_ [75]. On the contrary, the valence band is mainly formed by O 2p orbitals and is almost flat near 0 eV. Therefore, the precise location of the valence band maximum (VBM) is determined by the atomic configurations or the polymorphs of Ga_2_O_3_ [75].

For the DOS of the monoclinic polymorph (β-Ga_2_O_3_) from Figure 6, there also exists hybridization between Ga 4s and O 2p states in the lower region of the valence band. Furthermore, some Ga 4p states are located in the middle of the valence band of which some Ga 3d states are found near the top. The exact arrangement of the Ga and O atoms has a small impact on the energy distribution of the empty and occupied electronic states due to the strong ionic Ga-O bonding. That is why DOS diagrams of all different Ga_2_O_3_ polymorphs are very similar [75]. In addition to the valence band, there are two groups of core-level bands: bands mainly composed of O 2s orbitals in the energy region from −20.3 eV to −18.5 eV and bands mainly formed by Ga 3d orbitals in the energy region from −16 eV to −14.2 eV [73].

For β-Ga_2_O_3_, the valence band maximum (VBM) seems to be nearly degenerate at the Г and M points. The energy at Г point is 0.04 eV lower than that at M point. In this case, there is an indirect M-Г band gap of 4.83 eV, slightly smaller than the direct band gap of 4.87 eV at Г calculated by J. B. Varley et al. [74]. This result is close to the experimentally observed sharp absorption edge at ~4.9 eV [79]. Because of momentum conservation for the transition at Г point, the transition at Г-Г is more probable than that at M-Г [6]. The weakness of the indirect transition and the slight energy different between the indirect and direct band gaps make β-Ga_2_O_3_ a direct-band gap semiconductor [18]. However, the band gap for the rhombohedral corundum phase is obviously indirect and is roughly 0.25 eV less than the direct gap at Г. Additionally, the cubic bixbyite structure has an indirect band gap which is approximately 0.1 eV lower than the direct gap at Г [75].

Unlike the conduction band with the effective electron mass in the range of 0.24m_e_–0.34m_e_ [28,64,74] (where m_e_ is the free electron mass), the top of the valence band is almost flat, causing a very high effective hole mass (approximately 40 m_e_ along the Г-Z direction [74]), or a very low effective hole mobility for β-Ga_2_O_3_. Therefore, it is impractical for β-Ga_2_O_3_ to be fabricated as a p-type semiconductor [6]. In fact, the holes cannot move freely because they tend to produce localized polarons that are confined by local lattice distortions [72,80,81].

**Figure 6 nanomaterials-12-03601-f006:**
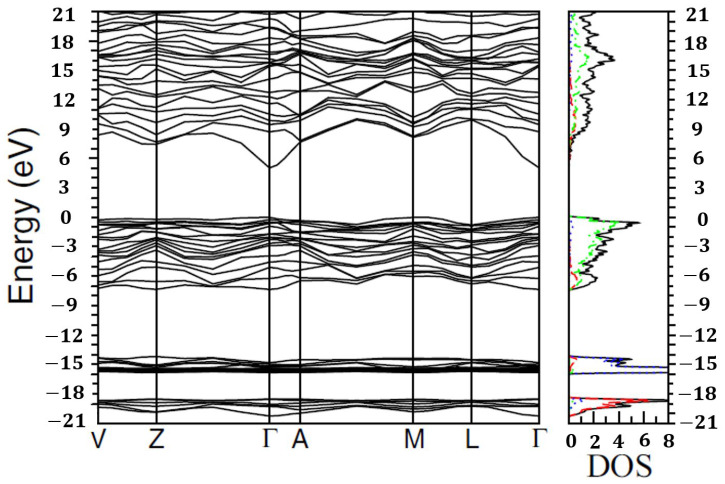
Quasiparticle band structures at HSE+G_0_W_0_ level of the β-Ga_2_O_3_ in the full energy range. In addition, the corresponding total (black full line), as well as the s- (red dashed line), p- (green dotted-dashed line), and d-projected (blue dotted line) density of states (DOS) is displayed. The valence band maximum (VBM) is defined as the zero energy. Figure reprinted with permission from [75]; copyright (2016) by the American Physical Society.

### 2.3. Other Properties

β-Ga_2_O_3_ has a wide band gap (4.8~4.9 eV) compared to SiC (3.3 eV) and GaN (3.4 eV) [82], which is intrinsically suitable for fabricating solar-blind deep ultraviolet (DUV) photodetectors [34,83,84,85]. Owing to its wide band gap, pure β-Ga_2_O_3_ crystals are colorless and extremely transparent up to UV-C range of the light spectrum. Typically, the absorbance spectra depict a steep cutoff absorption edge at approximately 255~260 nm with a shoulder approximately 270~275 nm [86,87,88,89]. This large absorption is due to the transition from the valence band to the conduction band [87]. However, β-Ga_2_O_3_ usually presents intrinsic n-type conductivity without any intentional doping because native oxygen vacancies exist in it [90]. Therefore, a slight red shift occurs in the intrinsic optical absorption edge of β-Ga_2_O_3_ [91], resulting in some minor absorption in the blue part of the visible spectrum [18]. That is why convention β-Ga_2_O_3_ crystals usually appear a light yellowish color. After intentional doping, β-Ga_2_O_3_ crystals become bluish because of increased free carriers absorbed in the red and NIR regions of the spectrum [92].

Due to its wide band gap, β-Ga_2_O_3_ has a very high critical breakdown electric field (E_br_~8 MV/cm) [47], making β-Ga_2_O_3_ a promising material for high-temperature and high-power applications [93]. In principle, β-Ga_2_O_3_ is intrinsically an insulator. However, n-type semiconductors of β-Ga_2_O_3_ can be easily achieved by doping the most commonly used donor dopants, Si and Sn elements, with shallow energy level and small activation energies of 30~80 meV in β-Ga_2_O_3_ [94]. Therefore, the electron concentration, which is proportional to the doping concentration, is controllable in a wide range of 10^15^~10^19^ cm^−3^ [79,89,95,96] and the resistivity is also tunable in an extremely wide range of 10^−3^~10^12^ Ω·cm [34,47] by changing the doping amount. As said previously, the effective electron mass of β-Ga_2_O_3_ is in the range of 0.24 m_e_~0.34 m_e_, which is relatively low and competitive with those of other wide band gap semiconductors [28,64,74]. However, the RT electron mobility (µ) in β-Ga_2_O_3_ is limited below 200 cm^2^/V·s by the LO phonon scattering [97]. Even so, its high critical breakdown electric field can compensate for this disadvantage because the Baliga’s figure of merit (BFOM), which is a fundamental parameter used to determine how well-suited a semiconductor material is for producing power devices [98,99], is proportional to the *E_br_* cubed, but is only proportional to μ to the first power.

β-Ga_2_O_3_ has anisotropic physical, optical, and electrical properties due to its unique configuration of Ga and O atoms [61,100,101,102]. To illustrate, the direction with the highest thermal conductivity, 27.0 ± 2.0 W/mK, was reported to be along the [010] direction while the lowest thermal conductivity, 10.9 ± 1.0 W/mK, was along the [100] direction at normal temperature [103]. However, even the highest thermal conductivity of β-Ga_2_O_3_ is still much smaller than those of other conventional semiconductors such as Si or GaN. Increasing temperature will only reduce its thermal conductivity [92,103]. For instance, the thermal conductivity decreases from 21 W/mK at 20 °C to 8 W/mK at 1200 °C [92]. The limited thermal conductivity of β-Ga_2_O_3_ is the most crucial potential shortcoming for high-power device fabrications because excessive heat accumulation will seriously affect the performance and reliability of devices [47].

In addition to thermal conductivity, there also exist anisotropic properties in material deformation under different temperature and pressure for β-Ga_2_O_3_. In order to define how the size of an object changes with a change in temperature and pressure, thermal expansion coefficient and elastic constant are introduced as basic properties of certain material, respectively. Evaluating thermal expansion coefficients of material is important to estimate the extent of the lattice mismatch between the substrate and the film under heteroepitaxy growth process [104]. However, there exists stress field in the interface of the substrate and the film because of the lattice misfit caused by thermal process. Therefore, knowing the elastic stiffness constants of a power-device material are also essential to fabricate such high-temperature devices [105,106,107,108].

Thermal expansion coefficients below RT of β-Ga_2_O_3_ are 1.8 × 10^−6^ K^−^^1^ for the *a*-axis and 4.2 × 10^−6^ K^−1^ for the b and c axes (with respect to RT) reported by Villora et al. [109]. The expansion along the *a*-axis is 2.4 times smaller than the one along the *b* or *c*-axis. On the other hand, thermal expansion coefficients above RT (300~700 K) of β-Ga_2_O_3_ are 1.54 × 10^−6^ K^−1^ for the a-axis, 3.37 × 10^−6^ K^−1^ for the *b*-axis, and 3.15 × 10^−6^ K^−1^ for the *c*-axis (with respect to RT) reported by Orlandi et al. [110]. The expansion coefficients for the *b* and *c* axes are almost the same and roughly double that of an even when the temperature is raised up to 1200 K. Furthermore, the increasing behavior of the thermal expansion coefficient with increasing temperature for the *a*-axis is different to those for the b and c axes [111]. These experiment results exhibit strong anisotropic behavior on the thermal expansion coefficient along the *a* axis, indicating that the (100) plane expands homogeneously [109]. In addition, when increasing temperature up to 1200 K, all three thermal expansion coefficients increase and gradually approach their high-temperature limit. This is a general saturation phenomenon that occurs when phonon modes are fully filled [111].

The mechanical properties of β-Ga_2_O_3_ are described by a set of elastic stiffness constants Cij defined in the relation: σ_i_ = C_ij_ε_j_ where σ is the stress applied to the object and ε is the strain describing the deformation amount influenced by the stress both with three tensile (i, j = 1~3) and three shear (i, j = 4~6) components, giving six components in total. Note that the elastic stiffness constants are symmetry, i.e., C_ij_ = C_ji_. Hence, there are 21 independent elastic stiffness constants. For a monoclinic symmetry crystal like β-Ga_2_O_3_, there are only thirteen independent elastic stiffness constants (C_11_, C_22_, C_33_, C_44_, C_55_, C_66_, C_12_, C_13_, C_23_, C_15_, C_25_, C_35_ and C_46_) and the other eight ones are zero. β-Ga_2_O_3_ has unusual elastic properties: strong longitudinal-modulus anisotropy (C_11_ = 242.8 ± 2.9 GPa, C_22_ = 343.8 ± 3.8 GPa, C_33_ = 347.4 ± 2.5 GPa, C_11_ ≪ C_22_, C_33_) and strong shear-modulus anisotropy (C_44_ = 47.8 ±0.2 GPa, C_55_ = 88.6 ± 0.5 GPa, C_66_ = 104.0 ± 0.5 GPa, C_44_ ≪ C55, C_66_) [112]. These anomalous elastic properties only exist in the specific space group *C2*/*m*. Monoclinic materials with other space group do not have these anomalous elastic properties [112]. The strong longitudinal-modulus anisotropic property (C_11_ ≪ C_22_, C_33_) indicates that β-Ga_2_O_3_ has relatively weak chemical bonds in the [100] direction, which means β-Ga_2_O_3_ is more compressible along the *x*-axis than that along the *y* and *z* axes [113]. In principle, most of the elastic stiffness constants of β-Ga_2_O_3_ are enhanced with increasing pressure when the hydrostatic pressure is less than 15 GPa. When the pressure is between 15 and 20 GPa, most of the elastic stiffness constants exhibit abnormal behavior, indicating that the monoclinic phase begins to transform to the rhombohedral phase of Ga_2_O_3_ (α-Ga_2_O_3_) [113], consistent with the phase transformation paragraph in the first section of the chapter.

Due to space limitations, many other basic physical properties of β-Ga_2_O_3_ have not yet been mentioned. These properties will be listed in Table 2, including refractive index, dielectric constant, thermal diffusivity, specific heat, etc. The basic physical quantities of β-Ga_2_O_3_ mentioned in this section are also listed in it.

## 3. Crystal Growth from Solution Process

A solution contains solutes and solvents. In principle, the main mission of solvents is to dissolve the solutes. However, the solvents may also hydrolyze the solutes to form insoluble substances under certain temperatures and pressure. Typical solvent of solutions could be water, various organic liquids, or their mixtures [118] depending on different requirements of hydrolyzing rate. After the hydrolysis reaction between the solutes and solvents, the solution becomes supersaturated for the insoluble substances, and crystallization occurs in the supersaturated solution [118]. A crystal is a solid in which particles are regularly arranged. Nucleation is the initial process of forming a crystal in the solution [119], in which a few ions, atoms, or molecules are dehydrated and gathered as microscopic crystalline nuclei formed either in the solution or along other surfaces. Then these nuclei will continue to grow and finally develop into large visible crystalline entities [118]. This process is often called crystal growth.

The solution process, or liquid-phase synthesis, is the most common method for preparing nanoparticles and nanostructured materials [120], whose size and shape could be well-controlled at low temperatures within a short time from minutes to hours. Furthermore, experimental processes of this kind of method are rather simple with a relatively low cost and high yield [121]. Methods for liquid-phase synthesis from compound solutions include sol-gel method, hydrothermal method, chemical bath deposition, reflux condensation method, forced hydrolysis method, successive ionic layer adsorption and reaction (SILAR) method, electrochemical deposition method, and so on. In the chapter, the basic crystal growth mechanisms of sol-gel, hydrothermal methods, and chemical bath deposition will be described in detail. Other liquid-phase methods will also be introduced briefly.

### 3.1. Sol-Gel Method

A colloidal solution is a kind of solution in which the size of the solute is between 10^−9^ m and 10^−7^ m. For a real solution, the size of the solute is less than 10^−10^ m in the form of molecules or ions. The solute and solvent are in one phase. In contrast, the solutes in the colloidal solution become dispersed colloidal particles suspended throughout the solvent, often called the dispersed medium in the colloidal solution [122]. When a light beam irradiates into the colloidal solution, the light beam is scattered by the colloidal particles. Then, a bright band of the light path in the colloidal solution can be observed from the perpendicular direction of the incident light. This phenomenon is called Tyndall effect. By using an ultra-microscope, you can see that these colloidal particles are suspended in the dispersed medium with random motion, which is called Brownian motion.

“Sol” is a kind of colloidal solution in which solid colloid particles are suspended in a liquid dispersed medium. In contrast to “sol”, a “gel” is a solid with three-dimensional network structures in which non-flowing liquid is dispersed. Solid becomes solvent and liquid becomes solute in the gel. The gel can be formed in the sol if these colloidal particles tend to agglomerate as inorganic polymers with connected porous structures in which the liquid dispersed medium is trapped. Or by drying to remove the solvents, the colloidal particles in the sol will poly-condense to form a gel [123].

Sol-gel method is a kind of liquid-phase synthesis using inorganic salts or metal alkoxide as precursors dissolved in organic solvents to form a solution [124]. By adding water, these precursors are hydrolyzed and poly-condensed simultaneously to agglomerate as colloidal particles so that the solution become a sol [122]. By aging the sol, the colloidal particles may further go through condensation reactions to polymerize as a gel with a network structure in the sol. In order to facilitate these colloidal particles to poly-condense as a gel, removing the solvents by thermal drying is a common method and finally an xerogel is obtained [123]. During the formation of the xerogel, the wet gel undergoes a large volume shrinkage and is easy to crack. To retain the original network of the gel and avoid cracking, supercritical drying is an essential way to dry the wet gel and an aerogel, whose liquid phase is replaced with gas, is obtained [125].

The sol-gel method, which is a kind of bottom-up method, is commonly used in fabricating nanomaterials of metal oxides, also called ceramic nanomaterial [126,127,128], at lower temperature between 70 and 320 °C [129,130,131,132]. Otherwise, the other methods used to produce nanomaterials need much higher temperature in the range of 1400~3600 °C [133,134,135,136]. Because the precursors in certain proportions are mixed homogeneously at a molecular scale, it is possible for the sol-gel method to make highly homogeneous nanocomposites with very high purity (99.99% purity) [122,137,138,139,140]. In addition, the sol-gel method is suitable for producing high quality nanoparticles with a narrow particle size distribution on an industrial scale [123]. However, the sol-gel method has some disadvantages such as high cost of metal alkoxides, longer processing time, toxic organic solvents for human beings, residual carbon and hydroxyl groups after drying the wet gel [141].

### 3.2. Hydrothermal Method

Hydrothermal synthesis is a classical method for preparing inorganic materials. This method was first studied by British geologist Sir Roderick Murchison (1792–1871) in the mid-19th century to simulate natural mineralization under strata in the condition of high temperature and high pressure. These naturally formed minerals are precipitated from hydrothermal ore solutions deep inside the earth. It can be seen that the term “hydrothermal” surely originated from geology [142]. The word hydrothermal is derived from the Greek words “hydros” and “thermos”, which mean water and heat, respectively [143]. Chemists, on the other hand, prefer to use the term, “solvothermal”, which is similar to hydrothermal, in which the water is replaced by any non-aqueous solvents. Here we only use the term “hydrothermal” to describe any chemical reactions, whether homogeneous (nanoparticles) or heterogeneous (bulk materials), occurring in a closed system with an aqueous (hydrothermal) or non-aqueous (solvothermal) solvent under high temperature and high pressure with the goal of dissolving and recrystallizing (recovering) substances that are by and large insoluble under normal circumstances [144].

It is well known that the most prevalent and significant solvent in nature is water. When increasing the temperature to 250~300 °C with higher pressure, ionic product (K_w_) has a maximum value (10–100 times of normal water), which is helpful for ion reaction [145]. The enhanced concentration of ${\mathrm{OH}}^{-}$ will facilitate hydrolysis of the metal salts, immediately followed by a dehydration step. Under this condition, precipitates of nano-sized particles are easy to form [146], which will be dissolved again under this high temperature and pressure. In addition, the dielectric constant decreases from 78.46 to 21 when the temperature raises from 25 °C to 300 °C [145], leading to the enhancement of the hydrolysis reaction rate based on the electrostatic theory. That is why hydrothermal synthesis is usually carried out below 300 °C [147]. However, the ionic product of water drastically decreases lower than that of normal water when the temperature and pressure reach near the critical point (374.3 °C and 22.1 MPa) because the dielectric constant and density decreases drastically [142,147], which is helpful for free radical reaction [145]. In addition, water transforms from an ionic species solvent to a nonionic species solvent under supercritical conditions [148]. Therefore, supercritical water is a particularly valuable reaction medium for organic matter and gas [145].

Due to the various properties of water under high temperature and pressure, the hydrothermal technique covers processes, such as hydrothermal transformation, hydrothermal decomposition, hydrothermal synthesis, hydrothermal dehydration, hydrothermal recycling, hydrothermal metal reduction, hydrothermal crystal growth, and so on [143]. Among them, hydrothermal crystal growth is the most frequently used process, which is done in a closed vessel under pressure called “autoclave” [149]. The principle of hydrothermal crystal growth is to transport the hydrolyzed and dissolved nutrients from the hot end to the cooler end by the convection process in order to grow seed crystals through a solubility differential within the pressure vessel induced by a temperature gradient between the dissolution zone (nutrients) and the crystallization zone (seeds) [150]. Although the technique requires a slightly longer reaction time than the vapor deposition processes or milling [144], the sealed growth vessel minimizes impurities and allows for controlling the processing conditions (temperature, pressure, oxidation potential, pH, concentration of precursors, etc.) [151] in order to ensure the production of targeted sizes and morphologies of nanoparticles with high crystallinity, purity and homogeneity [144].

### 3.3. Chemical Bath Deposition

In comparison to sol-gel and hydrothermal methods, chemical bath deposition (CBD) is the simplest solution process which does not need expensive organic precursors and solvents like the sol-gel process or a high temperature and pressure condition in a sealed autoclave like the hydrothermal process. Unlike the former two solution processes, CBD is dedicated to thin film deposition by a controlled chemical reaction in an aqueous phase which is in analogy to chemical vapor deposition in a gaseous phase [152]. As a branch of the solution processes, CBD has been developed for large-area thin film deposition as it has the following advantages: (1) it does not need additional expensive equipment, the precursors are easy to obtain and cheap. (2) it can be operated under low temperatures (<100 °C) and normal pressure [153]. CBD has produced a large number of thin films of metal chalcogenides, including sulfides, selenides, and oxides [154,155,156]. However, this method has only been used to produce a small number of thin films of divalent metal oxides (such as NiO, ZnO, and AgO) [157,158,159].

As stated at the beginning of this chapter, nucleation is the first step in the formation of crystals in a supersaturated solution, where a small number of ions aggregate to form microscopic nuclei. If nuclei are formed in solution, additional particles will be deposited on the nuclei without preferential orientation and larger spherical clusters will be formed [160]. This process is called homogeneous nucleation. If the nuclei are formed on phase boundaries, such as a surface of foreign substrates immersed in the solution, the nuclei will grow preferentially along the phase boundaries [161]. This process is called heterogeneous nucleation for which the barrier energy needed is lowered. That is why heterogeneous nucleation occurs more frequently than homogeneous nucleation unless the degree of supersaturation or supercooling rises [162].

CBD is a process for depositing thin films on surfaces that have been submerged in a dilute solution that contains metal ions and a source of hydroxide, sulfide, or selenide ions [163], which will combine to form metal chalcogenides. However, these ions in the solution do not always condense directly on the substrates by heterogeneous nucleation, which is usually called an ion-by-ion process (In general, the ion-by-ion process facilitates the growth of dense, sticky, mirror-like films [164]). These ions may first form colloidal particles and then agglomerate as clusters generated in the solution by homogeneous nucleation, which is caused by rapid hydrolysis of the metal ions to form precipitates. Finally, these clusters are absorbed by the substrates to form thin films [152]. This mechanism is usually called a cluster-by-cluster process, resulting in films that are opaque, non-uniform, and weakly adhering [164]. To reduce the cluster-by-cluster process and to enhance the ion-by-ion process, a chelating agent is often employed to reduce metal ion hydrolysis and to raise some stability to the bath [153].

### 3.4. Other Methods

In a chemical bath deposition, it is difficult to maintain a constant high temperature (<100 °C) of the bath in which chemical reactions are ongoing. It would require regular monitoring to maintain the reaction temperature not to vary too much. In addition, too much solvent is evaporated due to heating in an open vessel, causing the concentration of reactants to rise. In order to avoid losing solvent and to maintain a constant temperature, a reflux apparatus is needed for the chemical bath deposition process. In the reflux setup, solvent vapors from the liquid reaction mixture are trapped by a condenser, changing the phase from gas back to liquid form and returning to the bath. For convenience, the boiling point of the solvent is frequently chosen as the reaction temperature unless the specific temperature is essential to the reaction in which case a specialized heating apparatus would be required [165].

Forced hydrolysis of metal salt solutions is the most straightforward method for producing uniformly sized colloidal particles of metal (hydrous) oxides [166]. However, the hydrolysis rate must be slowed before forced hydrolysis because most polyvalent metal ions hydrolyze and precipitate rapidly [167]. By doing so, these metal ions can be hydrolyzed and precipitated under controlled conditions as monodispersed colloidal particles to avoid secondary nucleation, during which additional solutes are absorbed onto the existing nuclei, causing the particles to grow [168]. The forced hydrolysis method is commonly used in an acidified metal salt solution [169] or a metal salt solution with anions other than hydroxide ions [170] to prevent the hydroxide ions from rapidly combining with the metal ions. Finally, the concentration of ${\mathrm{OH}}^{-}$ ions must be enhanced to force the hydrolysis reaction, which can be accomplished by heating the solution [169] or adding a strong base [171].

To avoid the homogeneous precipitation of cationic and anionic precursors in the reaction solution, successive ionic layer adsorption and reaction (SILAR) is introduced to modify the CBD method [172]. In this technique, thin films are deposited by immersing the substrate alternately into different solutions in which cationic and anionic precursors are separately placed and rinsing the substrate with highly purified deionized water between every immersion to remove the loosely bounded ionic species [173] so that only the tightly adsorbed layer, which is a single ionic layer, stays on the substrate [174]. The SILAR method is based on a certain ionic species coated on the substrate immersed in another solution of an opposite ionic species, which will be successively adsorbed onto the original ionic layer to react ideally at the solid–liquid interface to form a single atomic layer of a new compound within a single reaction cycle. The process is then repeated to increase the thickness of the deposited thin film [175]. Therefore, SILAR is also known as a solution-based atomic layer deposition (SALD) [176].

In a reduction-oxidation reaction of CBD, electrons spontaneously go from lower to higher redox potential with no external DC power supply. In contrast, the electrochemical deposition method is based on electrolysis of a solution to induce a redox reaction which does not spontaneously happen. In a setup of an electrolysis reaction, an electrolyte solution is connected to a DC power supply through two electrodes, an anode and a cathode, dipped into the electrolyte solution. Each electrode is made by a conductor or semiconductor material. Electrons go from the anode through the circuit connected to the power supply to the cathode. Thus, the oxidation and the reduction reactions happen on the anode and the cathode, respectively. Finally, the reduction product, a thin and firmly adherent coating of the desired metal, oxide, or salt, can be deposited onto the cathode [177]. Meanwhile, the material of the anode is oxidized and dissolved into the electrolyte solution. In principle, metal ions in the solution are reduced and a metal coating is deposited on the cathode. However, choice of the anions and the pH of the solution may facilitate other reduction reactions on the cathode, depending on the redox potential [178].

## 4. Ga_2_O_3_ Materials and Thin Films

This chapter extensively discusses synthesis of pure Ga_2_O_3_, co-doped Ga_2_O_3_, Ga_2_O_3_-metal oxide composite, and Ga_2_O_3_/metal oxide heterostructure nanomaterials via solution-based methods mainly sol-gel, hydrothermal, chemical bath methods, and solvothermal, forced hydrolysis, reflux condensation, and electrochemical deposition methods.

### 4.1. Ga_2_O_3_ by Sol-Gel Process

The sol-gel process was widely used to synthesize thin films, nanorods, nanoparticles, and nano-powders of Ga_2_O_3_. The properties of synthesized Ga_2_O_3_ depend on its morphology and size and also on the precursor used and synthesis conditions. The synthesis conditions are pH of the precursor solution, ageing time, solution temperature, deposition time and also calcination temperature. The most common phase obtained at room temperature was α-GaOOH and by calcining this phase at different temperatures, different polymorphs of Ga_2_O_3_ can be obtained. Figure 7 illustrates the stepwise sol-gel synthesis of Ga_2_O_3_ and Table 3 lists the relevant literature.

The various precursors used to synthesize Ga_2_O_3_ nanomaterials were gallium acethylacetonate [12], gallium isopropoxide [179,180,181,182,183], Gallium(III) chloride aqueous solution [180], Ga metal [184,185] and gallium nitrate hydrate [14,15,186,187,188,189,190,191]. G. Sinha et al. [192] prepared the pure Ga_2_O_3_ nanocrystalline thin film on quartz substrate for the first time by the sol-gel method and analyzed the effect of annealing temperature on phase variation of the deposited film. The as-deposited GaOOH phase converted to pure α-Ga_2_O_3_ at 500 °C and pure β-Ga_2_O_3_ at 700 °C and higher temperatures. The synthesized GaO(OH) had a band gap of 5.27 eV, pure α-Ga_2_O_3_ has a band gap of 4.98 eV, and the band gap of β-Ga_2_O_3_ varied from 4.8 to 4.7 eV when annealed in the temperature range of 700 °C to 1100 °C.

M. Tadatsugu et al. [12] prepared thin films of Ga_2_O_3_:Mn for the first time via the sol-gel dip-coating method. The Ga_2_O_3_:Mn thin films annealed for 1 h at 850–1070 °C in Ar ambience and had an amorphous nature.

Y. Li et al. [179] synthesized sol-gel prepared Ga_2_O_3_ semiconducting thin film doped with Ce, Sb, W, and Zn. The spin-coated films were annealed at 600 °C for 1 h to get Ga_2_O_3_ thin film. M. Ristic et al. [180] synthesized nanoparticles of α-Ga_2_O_3_ and β-Ga_2_O_3_ using gallium(III)-isopropoxide and aqueous GaCl_3_ solution as starting materials. A dominant amorphous phase and crystalline α-GaOOH particles were obtained by addition of hot water and then aqueous TMAH solution to 2-propanol solution of gallium (III)-isopropoxide. Aggregates of amorphous α-GaOOH consist of nanoparticles. At room temperature, hydrolysis of gallium (III)-isopropoxide with pure water yielded an amorphous phase only. Polymerization and condensation of gallium (III)-isopropoxide hydrolytic products can explain the formation of this amorphous phase. The addition of aqueous TMAH solution into aqueous GaCl_3_ solution resulted in a single-phase of α-GaOOH particles (submicrometric range). The formation of α-GaOOH particles can be explained by a fast, solution-mediated transformation of hydrated Ga(OH)_3_. Y. Kokubun et al. [181] synthesized the β-Ga_2_O_3_ thin film on sapphire substrate by the sol-gel method. A major diffraction peak of ($\bar{2}$01) corresponding to β-Ga_2_O_3_ appeared in all the films when annealed at 400 °C to 1200 °C for 1 h. The optical absorption studies confirmed that the band gap of the film annealed at 800 °C has 4.95 eV. When the annealing temperature increases above 900 °C, the lattice constants of the β-Ga_2_O_3_ films decreased and the band gap increased due to diffusion of Al from sapphire substrates into Ga site in β-Ga_2_O_3_ lattice. The diffusion of Al into Ga site has been seen in the other synthesis methods, such as sputtering [193] and CVD [194]. R. Suzuki et al. [182] grew β-Ga_2_O_3_ thin films epitaxially on a 0.4 mm thick (100) β-Ga_2_O_3_ substrate using the sol-gel method similar to the method reported in [181]. R. Gopal et al. [183] synthesized β-Ga_2_O_3_ mono-crystalline nanorods by the sol-gel method using hydrolyzation of the precursor solution containing gallium(III) isopropoxide (Ga(Opr^i^)_3_) dissolved in anhydrous iso-propanol. They also synthesized γ-Ga_2_O_3_ polycrystalline nanoparticles by hydrolysis of a new modified precursor [{(H_5_C_6_) N=CH-C_6_H_4_O} Ga(Opr^i^)_2_], which was derived by a reaction between (Ga(Opr^i^)_3_ and N-phenylsalicylaldimine ([C_6_H_4_(OH)CH=N(C_6_H_5_)]) in refluxing benzene medium:(Ga(Opr^i^)_3_) + [C_6_H_4_(OH)CH=N(C_6_H_5_)] → [{(H_5_C_6_) N=CH-C_6_H_4_O} Ga(Opr^i^)_2_] + Pr^i^OH

After the hydrolysis of the above precursor solutions, β-Ga_2_O_3_ and γ-Ga_2_O_3_ phases were created by sintering the occurred products at 600 °C for 6 h. The average crystallite sizes of β-Ga_2_O_3_ nanorods and γ-Ga_2_O_3_ nanoparticles measured from XRD patterns were 120 nm and 32 nm, respectively, which are larger than TEM measured values of 100 nm (i.e., diameter of nanorods) and 10 nm (i.e., nanoparticles) due to antistrophic strains. The SEM and TEM images β-Ga_2_O_3_ nanorods and γ-Ga_2_O_3_ particles are shown in Figure 8.

G. Sinha et al. [184], prepared the Ga_2_O_3_ nanoparticles of finite size in silica matrix by the sol-gel method. The crystallite sizes of Ga_2_O_3_ in the Ga_2_O_3_:SiO_2_ composites (molar ratio: 10:90, 20:80, and 30:70) were found to be 1.2, 1.81, and 1.9 nm, respectively. They observed that β-Ga_2_O_3_ was formed in composite at a very low temperature of 400 °C, which was in contrast to their previous study [192]. This was explained by the capping effect of silica that confines the Ga_2_O_3_ to a very small size [195]. The band gap (E_g_) of Ga_2_O_3_:SiO_2_ (10:90) composite was 5.51 eV, which had a significant deviation from the bulk material (4.9 eV [196]) and was considered to be the effect of nanosized β-Ga_2_O_3_ particles in the silica matrix. The PL studies revealed that the Ga_2_O_3_:SiO_2_ composites had shown a strong blue emission peak at 460 nm, which was proved in the previous study by Binet and Gourier [196]. They also used sol-gel prepared pure β-Ga_2_O_3_ thin film as a substrate for Ga_2_O_3_ nanowire fabrication by the VLS method [185]. The film had shown a ($\bar{2}$02) diffraction peak with lower crystallinity when annealed at 700 °C for 1 h.

B. Cheng et al. [186] prepared the hollow nanotubes of crystalline β-Ga_2_O_3_ by the sol-gel method for the first time using a porous alumina template with gallium nitrate hydrate as starting materials. M.R. Mohammadi et al. [187] used the particulate sol-gel method to synthesize the mesoporous TiO_2_ and Ga_2_O_3_ thin films on quartz and alumina substrate transducers with various Ti:Ga atomic ratios (at.%/at.%) = 100:00, 75:25, 50:50 and 25:75. A polymeric fugitive agent (PFA), such as hydroxypropyl cellulose, was added to solution to enhance the porosity of the films in nanoscale. The average crystallite size of synthesized TiO_2_-Ga_2_O_3_ powders (i.e., 2–5 nm) less than that of pure TiO_2_ powders (i.e., 4–10 nm) confirms that Ga_2_O_3_ retards the anatase to rutile formation of TiO_2_ by preventing the grain growth and crystallization. A. Kaya et al. [14] synthesized the β-Ga_2_O_3_ thin films on a p-type Si substrate using the sol-gel method by annealing the as-synthesized films at a temperature of 800 °C for 2 h in Ar ambience. In XRD studies, the annealed films showed major intensities for (400) and ($\bar{1}$10) planes corresponding to β-Ga_2_O_3_. J. Gao et al. [15] synthesized β-Ga_2_O_3_ thin films coated on MOCVD grown GaN substrate by using the sol-gel method similar to their previous work [14]. H. Shen et al. [188] synthesized β-Ga_2_O_3_ thin films on c-plane sapphire substrate by a sol-gel spin coating technique. In XRD studies, the film annealed at 700 °C showed ($\bar{2}$01) and ($\bar{6}$03) planes corresponding to β-Ga_2_O_3_, and with an increase in the annealing temperature, the peak intensities were increased. All the β-Ga_2_O_3_ films showed 90% transmittance over 300 nm. The optical band gap (E_g_) of the films measured from the transmittance spectra revealed that when the annealing temperature of the films increased from 500 °C to 700 °C, E_g_ was increased monotonously from 5.07 eV to 5.24 eV and exceeded to 5.67 eV at 1100 °C. This rapid increase of band gap was explained by the diffusion of Al from alumina substrate into the Ga site in β-Ga_2_O_3_, which was already seen in a previous study [181].

M. Yu et al. [189] synthesized the α/β polycrystalline Ga_2_O_3_ thin films introducing α-Ga_2_O_3_ into β-Ga_2_O_3_ on sapphire substrate by a novel sol-gel method. When as-synthesized film annealed in air at 600; it had shown low intensities peaks corresponding to β-Ga_2_O_3_ phase. At annealing temperatures of 700 °C abd 800 °C, the (006) orientation of α-Ga_2_O_3_ co-existed with β-Ga_2_O_3_. At 900 °C, the (006) orientation of α-Ga_2_O_3_ disappeared completely making the film fully into β-Ga_2_O_3_ phase. The as-synthesized film was annealed at 800 °C to study the influence of different annealing environments (namely O_2_, N_2_ and N_2_-O_2_). It is revealed that the rich O_2_ environment would enhance the growth of α-Ga_2_O_3_ and poor O_2_ would suppress it. AFM images revealed that the sample annealed in an O_2_ atmosphere had a dense surface with a small RMS value of 4.17 nm, which was attributed to oxygen vacancy compensation and increased crystallinity. While the RMS value of the sample annealed in N_2_ ambience was 10.4 nm due to the presence of a large number of oxygen vacancies. The band gap from the Tauc plot for the films annealed in N_2_, O_2_ and N_2_-O_2_ atmospheres was 4.9 eV, 5.09 eV, and 5.04 eV, respectively. The oxygen vacancies acted as trap states at the top of the valence band, extending into the forbidden band and causing the narrow band gap. Figure 9 shows the SEM and AFM images of high quality α/β polycrystalline Ga_2_O_3_ thin film that occurred at an annealing temperature of 800 °C in O_2_ atmosphere.

S. Yu et al. [190] derived Ga_2_O_3_ sub-micro powders by the citrate sol-gel method. The Ga_2_O_3_ powders were obtained by calcination of the dried sol at 500 °C for 4 h in an O_2_ atmosphere. The particle size for these powders measured with SEM was less than 1 µm. These ceramic powders were sintered via SPS at 830–980 °C for 5 min under a pressure of 90 Mpa. The β-Ga_2_O_3_ sample that occurred by sintering at 980 °C had a transmittance higher than 40% in the wavelength range of 396–667 nm and its SEM image was shown in Figure 10.

Y. Zhu et al. [191] successfully synthesized the β-Ga_2_O_3_ thin films on (0001) sapphire substrate by a simple and effective sol-gel spin coating method. The SEM images of β-Ga_2_O_3_ films synthesized with pre-heating temperatures of 100 °C and 200 °C had surface cracks, while the 300–500 °C pre-heated films were crack-free and very flat. They reported that these cracks are due to volatilization of organic solvent under low pre-heating temperatures and oxidative decomposition under high-temperature crystallization. The AFM studies revealed that β-Ga_2_O_3_ films preheated at <300 °C had more RMS value, which was explained by obvious porosity. The β-Ga_2_O_3_ film prepared with a pre-heating temperature of 400 °C had a low RMS value of 1.982 nm. The XRD studies revealed that the synthesized β-Ga_2_O_3_ films had peaks corresponding to {$\bar{2}$01} family of planes, which proved the epitaxial relationship of β-Ga_2_O_3_ [$\bar{2}$01]//[0001]Al_2_O_3_ and the same was reported in other studies [197,198]. The XRD study of 400 °C pre-heated and 1000 °C post annealed β-Ga_2_O_3_ thin film grown on the ~7° off angled sapphire substrate revealed that the film had the major peak corresponding to (400) plane of β-Ga_2_O_3_ which supports the out-plane epitaxial relationship of β-Ga_2_O_3_ [400]//[11$\bar{2}$3]Al_2_O_3_ for the β-Ga_2_O_3_ films grown on ~7° off angled sapphire substrate [199,200].

### 4.2. Ga_2_O_3_ by Hydrothermal Process

The hydrothermal method was widely used to synthesize various morphologies of Ga_2_O_3_ materials. The properties of Ga_2_O_3_ materials and its morphologies typically depend on the type of Ga precursor, surfactant, and synthesis conditions, such as pH of the solution, ageing time, reaction temperature, and reaction time and also on calcination temperature. Morphologies of thin films by hydrothermal method could depend on the type of the seed layer. Figure 11 illustrates the stepwise hydrothermal synthesis of Ga_2_O_3_, and Table 4 lists the relevant literature.

The various precursors used to synthesize Ga_2_O_3_ nanomaterials were gallium acethylacetonate [201], commercial Ga_2_O_3_ [45], Gallium(III) chloride aqueous solution [43,201], Ga metal [202], and gallium nitrate hydrate [21,203,204,205,206,207,208,209,210,211,212,213,214,215,216,217,218,219,220,221,222,223,224,225,226,227,228,229,230,231].

S. Suman et al. [201] synthesized gallium oxide nanostructures via the hydrothermal method using three different precursors of gallium, such as gallium acetylacetonate (GO-Ga), gallium chloride (GO-Cl), and gallium nitrate (GO-Ni), and studied its effect on morphology. The morphology of Ga_2_O_3_ nanostructures prepared with gallium acetylacetonate was cuboid in shape (average size of 0.6 µm × 0.3 µm × 0.2 µm), while the nanostructures prepared with gallium chloride and gallium nitrate were of a rice-like morphology and of average size, (L × W) 1.5 µm × 0.5 µm and 1.3 µm × 0.5 µm, respectively. The β-Ga_2_O_3_ phase started forming at an annealing temperature of 800 °C.

**Figure 11 nanomaterials-12-03601-f011:**
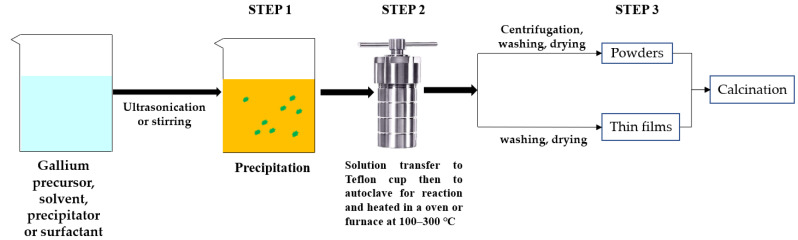
The schematic of hydrothermal preparation of gallium oxide illustrating all steps involved.

J. Zhang et al. [45] synthesized β-Ga_2_O_3_ nanorods by commercial Ga_2_O_3_ precursor. Ga_2_O_3_ was dissolved first in HCl and water or (DEG: water mixture) was added to it. The hydrothermally prepared GaOOH nanorods were calcined at 900 °C in air. The pH of the solution affected the morphology of Ga_2_O_3_. At pH = 6, the Ga_2_O_3_ quadrilateral nanorods formed, and at pH = 8, spindle-like nanorod arrays formed. The nanorods have preferential growth direction along c-axis. At pH = 6, the nanorods prepared without and with (DEG: water mixture) had the aspect ratio of 1:7, and 1:15 confirms that the addition of DEG affected the aspect ratio.

S. Krehula et al. [43] used GaCl_3_ aqueous solution and TMAH to create α-GaOOH uniform submicron particles of various shapes (rhombic rods, rhombic prisms, hierarchical structures), as shown in Figure 12. The pH of the solution (pH = 5–9), reaction temperature (60 °C or 160 °C), and aging time (5–7 days with reaction temperature 60 °C) affected the morphology of α-GaOOH. Spindle-shaped particles were formed in acidic conditions at low temperatures, as shown in Figure 12a, and rhombic rods elongated in the c-axis direction were formed at high temperatures, as shown in Figure 12b. In neutral conditions, rhombic prisms were formed, while in base conditions uniform hierarchical α-GaOOH particles were formed. The α-GaOOH prepared at 160 °C and at pH 7 (as shown in Figure 12d) has less crystallite size compared to acid or base conditions. At neutral conditions, the morphology was rhombic prisms of average length of 600 nm and edge length of (001) rhombic face of approximately 200 nm. When as-synthesized α-GaOOH calcined at 500 °C, the α-GaOOH particles changed to α-Ga_2_O_3_ retaining the morphology but at higher temperatures of 900 °C; samples collapsed severely except the sample prepared hydrothermally at 160 °C, which had retained the shape and particle structure.

J. Liu et al. [202] synthesized mesoporous mixed phase Ga_2_O_3_ using Ga metal and water precursor solution. The obtained α-GaOOH samples were calcined at 400 °C for 5 h and then calcined subsequently at 600–800 °C for 1.5 h. The XRD and TEM results showed that the Ga_2_O_3_ sample calcined at 700 ℃ was porous and had a mixed phase of α-Ga_2_O_3_ and β-Ga_2_O_3_. Both GaOOH and Ga_2_O_3_ showed rod-like morphology.

Y. Zhao et al. [203] prepared nano to micro sized β-Ga_2_O_3_ and γ-Ga_2_O_3_ with or without surfactant (PEO and CTAB) by a low temperature hydrothermal route. The quadrilateral GaOOH rods were obtained from the hydrothermal treatment of gallium hydrate precipitate (i.e., obtained from the solutions prepared by mixing of Ga(NO_3_)_3_, NaOH, and HNO_3_ in two different procedures) to which the surfactants PEO or CTAB were added. The GaOOH rods prepared from the first procedure (i.e., Ga:OH = 1:3) were changed to β-Ga_2_O_3_ phase by calcination at 900 °C. In the second procedure (i.e., Ga:OH = 1:5), the γ-Ga_2_O_3_ rods were directly synthesized after hydrothermal treatment without further need for calcination. The addition of the surfactants PEO or CTAB had no effect on the morphology of hydrothermally synthesized GaOOH rods, but there was a significant difference in pore size, mesoporosity, and pore distribution of β-Ga_2_O_3_ rods.

In addition, the effect of other surfactants, such as sodium dodecyl benzene sulfonate (SDBS) and sodium acetate (SA) on the morphology of GaOOH was explained by Y. Quan et al. [204]. As shown in Figure 13a, the morphology of α-GaOOH prepared without adding any surfactants was spindle-like particles (i.e., 1.5 µm in length). The morphology of α-GaOOH prepared with the SDBS surfactant were brush-like particles composed of the nanowires originating from a central knot, as shown in Figure 13b. The nanowire and nanobelt structured growth is related to the anisotropic growth due to selective absorption of the SDBS onto crystallographic facets of a growing crystal [232,233,234]. The products prepared with SA were agglomerated cuboid-like particles, as shown in Figure 13c.

FTIR spectral studies confirmed that the GaOOH particles were turned into pure hexagonal phase of α-Ga_2_O_3_ at 600 °C (i.e., a broad band at 669.8 cm^−1^ assigned to the valence band vibrations of Ga-O in the lattice formed by GaO_6_ octahedra) and monoclinic phase of β-Ga_2_O_3_ at 900 °C (665.4 cm^−1^-GaO_6_, 753.1 cm^−1^-GaO_4_). The PL studies of β-Ga_2_O_3_ revealed that it can emit a stable and bright blue light with a peak at 2.84 eV (437 nm) in the blue range [196] due to recombination of an electron on an oxygen vacancy donor and a hole on a gallium vacancy acceptor [235].

J. Wang et al. [205] synthesized the microspheres of β-Ga_2_O_3_ with hollow interior by calcination of hydrothermally prepared GaOOH microspheres at 1000 °C for 10 h. They proposed a 3-step vesicle template mediated self-assembling process to explain the formation of GaOOH microspheres. In aqueous solutions, the assembly of surfactant molecules having bipolar functional group are more liable to formation of micelles and closed bilayer aggregates, such as vesicles. The bilayer structure separates the aqueous interior from the exterior, providing an interface between the surfactant groups and solution, presenting a specific site for material growth from solution. Vesicles are considered as soft templates that form the hollow spherical structure [236]. As the hydrothermal time prolongs, the nuclei grow anisotropically to nanorods at the surface of templates, resembling the vesicle-template mechanism for the formation of these microspheres.

M. Muruganadham et al. [206] synthesized hollow α-GaOOH microspheres by adding oxalic acid to gallium nitrate in a hydrothermal process. The SEM studies revealed that the reaction temperature had greatly affected the morphologies of as-synthesized GaOOH, shown in Figure 14. At a reaction temperature of 175 °C—flower-like morphology stacked by nano-sheets, at 200 °C—hollow microsphere, at 225 °C—microrods with few microspheres existed. The variation in the concentration of oxalic acid does not affect the morphologies of GaOOH.

The as-synthesized GaOOH was calcined for 3 h at 450 °C to get α-Ga_2_O_3_. The GaOOH micro flower changed to a rod/plate-like morphology during calcination, while the other GaOOH morphologies retained their shape. The as-synthesized GaOOH prepared without the oxalic acid at hydrothermal temperatures of 175 °C to 225 °C did not show any microsphere morphology. The HR-TEM images of α-Ga_2_O_3_ synthesized with calcination of GaOOH prepared at a hydrothermal temperature of 200 °C showed that the nanoparticles assembled on the microsphere surface with oriented attachment could grow similar to rods with a very sharp end, as shown in the insert red circle of Figure 15a. Figure 15b shows that these nanoparticles were well-crystalline and defect free.

D. Li et al. [207] synthesized mesomorphs α-GaOOH hierarchical structures by a liquid-assisted hydrothermal method. The as-synthesized products have the morphology of a hierarchical structure with a diameter of 1 µm, self-assembled by single crystalline GaOOH nanorods (i.e., diameter 50–100 nm and 0.5 µm length) that grow along the [00$\bar{1}$] axis. C_8_H_16_N_2_O, also known as [Bmin][OH], had a key role as a soft template in the formation of the α-GaOOH hierarchical structure. As shown in Figure 16, the mesoporous α-Ga_2_O_3_ hierarchical structure was obtained by calcining the as-synthesized α-GaOOH hierarchical structure for 3 h at 450 °C.

B. Arul Prakasam et al. [208] used the hydrothermal method to synthesize self-assembled α-GaOOH microrods and micro flowers. The micro rods were prepared by adding biuret to gallium nitrate in the hydrothermal process. The as-synthesized α-GaOOH micro rods have a length of 2–5 µm. The micro flowers shown in Figure 17a were prepared by using oxalic acid (C_2_H_2_O_4_) instead of biuret. The micro flowers were made from nanoribbons and nanorods together. The as-synthesized GaOOH micro flowers had two distinct size variations. The larger were in the size of 2–4 µm whereas the smaller were 500 nm. The smaller lacked nanoribbons. The width of the nanorod in the micro flower was 50 nm. The α-Ga_2_O_3_ phase obtained by calcination of α-GaOOH for 3 h at 450 °C. The TEM image of α-Ga_2_O_3_ nanorod/ribbons in microflowers shown in Figure 17d, revealed the porous nature of α-Ga_2_O_3_ and the TEM image of microrods shown in Figure 18 revealed the mesoporous nature.

B.K. Kang et al. [209] fabricated uniform β-Ga_2_O_3_ hollow nanostructures by the hydrothermal method using carbon spheres as templates. The carbon nanospheres were prepared using a hydrothermal synthesis of a clear solution made by mixing glucose and distilled water. The carbon colloid solution was prepared by adding carbon nanospheres to a solution of ethanol and distilled water. The synthesis of Ga(OH)CO_3_ core shells around carbon spheres in the urea precipitating solution was described as follows:Ga^3+^ + (NH_2_)_2_CO + 3H_2_O → Ga (OH)CO_3_ + 2NH^4+^ + H^+^

FTIR spectra for Ga-coated core-shell structure had a sharp and intense band absorption at 1384 cm^−1^ due to the ν_3_ mode of interlamellar [CO_3_]^2−^ ions. This Ga coated core shell nanostructures were calcined between 500–900 °C. The sample calcined at 600 °C had shown low intensity peaks of β-Ga_2_O_3_. Calcination at 700 °C, confirmed the well crystallized peaks of β-Ga_2_O_3_. The calcined β-Ga_2_O_3_ hollow nanostructures have the diameter of approximately 200–250 nm.

F. Shi et al. [210] synthesized β-Ga_2_O_3_ nanorods using the urea precipitating solution. They investigated the effect of reaction time on the morphologies and properties of β-Ga_2_O_3_ nanorods. SEM morphologies of β-Ga_2_O_3_ samples showed that the products with a short reaction time of 1 h were resulting in amorphous precipitates. In reaction times of 3 h, these amorphous precipitates were converted into nanorods of uniform shape and size (i.e., length—3 µm, width—300 nm). With the 10 h reaction time, the length and width of nanorods increased to 6 µm and 1 µm, respectively. The morphology was micron-sized flowers composed of many elongated micrometer rods scattered along a common ground. The nanorods were porous and single crystalline as per TEM studies. The Raman spectrum for the sample obtained after 10 h of hydrothermal treatment has the highest intensity and narrowest full width at half-maximum, indicating the highest crystallinity. The photoluminescence spectra show that the three samples’ emission peaks appear in the blue-violet region between 375 and 425 nm, which was greatly affected by the hydrothermal time.

L. Cui et al. [211] synthesized ultrafine γ-Ga_2_O_3_ nanocrystals using urea solution by a microwave hydrothermal method. The hydrothermal products synthesized at 140 °C for a maintain time of 2 min at low urea concentrations (2 and 2.7 g) were a mixed phase of GaOOH, γ-Ga_2_O_3_, and at high concentrations (3.4 and 4.1 g) were purely γ- Ga_2_O_3_. The nanocrystals synthesized with 3.4 g of urea at 140 °C for 2 min have higher crystallinity, whereas the products of the same amount of urea at 150 °C were GaOOH. The products prepared with 3 g of urea at 140 °C for more time (3 min, 4 min) resulted in GaOOH. The optimal hydrothermal conditions to prepare γ-Ga_2_O_3_ nanocrystals with 3.4 g of urea were 140 °C for 2 min maintaining time. When these as-synthesized γ-Ga_2_O_3_ nanocrystals were calcined for 2 h at 300–700 °C, retained their γ-Ga_2_O_3_ phase up to 500 °C and changed to β-Ga_2_O_3_ at 600 °C. The γ-Ga_2_O_3_ particles were round in shape and had the crystallite size of 5–7 nm with a band gap of 4.61 eV. The PL spectroscopy of calcined γ-Ga_2_O_3_ particles at 254 nm at room temperature showed that γ-Ga_2_O_3_ could exhibit violet-blue broadband emission with a peak wavelength of 410 nm, as reported by C.C. Huang et al. [20].

W. Zhao et al. [212] synthesized mesoporous β-Ga_2_O_3_ nanorods by hydrothermal method using different M.W PEG as a template in urea precipitating solution. PEG was used as a template reagent to synthesize mesoporous nanomaterials [237]. The morphology of obtained GaOOH nanorods was similar to the quadrilateral prisms reported by Zhang et al. [45]. The obtained GaOOH white precipitates were calcined at 800 °C for 2 h in a furnace to acquire the β-Ga_2_O_3_ nanorods. Without PEG, the morphologies of produced β-Ga_2_O_3_ nanorods had no pores, but the nanorods synthesized using 1 mL PEG200 had mesoporous nature, as illustrated in Figure 19b. The band gap of β-Ga_2_O_3_ mesoporous nanorods was approximately 4.4 eV which is less than 4.7 eV as reported by Hou et al. [238]. New states created due to N doping were just above the valence band for substitutional or interstitial nitrogen, which could narrow down the band gap by combining N 2p and O 2p states.

Y. Wang et al. [213] also synthesized β-Ga_2_O_3_ nanorods by hydrothermal method using PEG in the urea precipitating solution. The as-synthesized products calcined for 10 h at 800 °C. XRD studies showed that the obtained products were highly crystalline β-Ga_2_O_3_ phase. The TEM images of the β-Ga_2_O_3_ nanorod revealed that the synthesized nanorods were porous and had diameter of 60 nm with a length of 500 nm.

S. Fujihara et al. [214] synthesized rod-like polycrystalline β-Ga_2_O_3_ films by hydrothermal method using gallium nitrate aqueous solution. The as-synthesized GaOOH rods were calcined at 700 °C to get the β-Ga_2_O_3_ phase. The formation of GaOOH precipitate in the solution was governed by the following temperature dependent reaction [239].
Ga^3+^ + 2H_2_O→GaOOH + 3H^+^

Considerable changes in the chemistry of Ga^3+^ ions occur under hydrothermal conditions due to decreased viscosity, decreased dielectric constant and increased ionic product of water [240]. The synthesized GaOOH rods were 3 µm in length with an aspect ratio of 4. The thermal decomposition of GaOOH into β-Ga_2_O_3_ increased the aspect ratio to 6–7 on average, even though morphology was retained.

L. S. Reddy et al. [21] also used gallium nitrate aqueous solution to synthesize α-Ga_2_O_3_ (at 500 °C) and β-Ga_2_O_3_ nanorods (at 800–1000 °C). The GaOOH prepared at room temperature looks like a cocoon-shaped structure formed by multi-layers of small nanoplates. At 50 °C, these small nanoplates merged and increased the thickness and width of each plate due to Ostwald ripening. At 75 °C, the stacked structure converted into a rod-like structure. At 95 °C, the length of these nanorods increased while the width decreased. TEM studies revealed that the α-Ga_2_O_3_ and β-Ga_2_O_3_ nanorods were porous and have length of 3 µm and width in the range of (280–400 nm).

H. J. Bae et al. [215] also used gallium nitrate aqueous solution to synthesize high aspect ratio β-Ga_2_O_3_ nanorods (at 1000 °C for 5 h). The spindle-like nanorods of α-GaOOH precipitated with ageing at 60 °C for 1 h while with 6 h ageing time, spindles changed to prism-like nanorods due to face-to-face anisotropic stacking of nanoplates. After the hydrothermal reaction in an autoclave at 140 °C for 10 h, the morphologies of α-GaOOH precipitate changed. High aspect ratio α-GaOOH nanorods obtained with 2 h of ageing and followed by a hydrothermal reaction. After calcination, β-Ga_2_O_3_ nanorods retained the morphologies of hydrothermally synthesized α-GaOOH.

R. Pilliadugula et al. [216] synthesized α-Ga_2_O_3_ (at 400–700 °C) and β-Ga_2_O_3_ nanorods (900, 1000 °C) using gallium nitrate aqueous solution. The band gap of as-synthesized GaOOH and β-Ga_2_O_3_-1000 °C powders measured by diffuse reflectance spectra was 4.6 eV and 5.51 eV respectively. In their another study [217], they synthesized GaOOH powder samples at various (pH = 5, 7, 9, 11, 14) and calcined at 1000 °C to get β-Ga_2_O_3_. The Gallium oxide GO-5 and GO-14 samples resulted in less crystallite size (i.e., 30.07 nm and 48.97 nm respectively) due to the existence of compressive strain in those samples. GO-5 sample had rod-like morphology, and cocoon-shaped morphology from pH = 7 to 11, and the GO-14 sample showed hierarchical structure formed from the rod morphology.

F. Shi et al. [218] also synthesized microspheres of β-Ga_2_O_3_ at different reaction temperatures (40, 80, 120 and 160 °C) using aqueous solution (i.e., at pH = 10). The β-Ga_2_O_3_ microspheres prepared at hydrothermal temperatures of 120 °C and 160 °C collapsed seriously after calcination. Crystalline particles prepared at higher hydrothermal temperatures can combine with more crystalline water. The evaporation of crystalline water at high temperatures leads to the structural collapse of the β-Ga_2_O_3_ products. The β-Ga_2_O_3_ microspheres were 1.2 µm in length and 300 nm and had dense holes on the surface. The PL spectra of β-Ga_2_O_3_ microspheres revealed emission peaks between 350 and 450 nm in the blue-violet region.

H. J. Lin [219] et al. synthesized pure β-Ga_2_O_3_ nanorod arrays (NRAs) a 50 nm thick SnO_2_ seed layer sputtered on a 1 µm SiO_2_ insulating layer Si (100) substrate by hydrothermal method. The preferential growth orientation of the GaOOH nanorod arrays was found to be perpendicular to the (111) plane as shown in Figure 20b.

B. Zhang et al. [220] also synthesized β-Ga_2_O_3_ NRAs similar to the previous study [219]. The XRD of synthesized GaOOH array had a major peak of (111) which shows that preferential growth direction perpendicular to (111) plane. The nanorods had a length of 1.8 µm and the tips of nanorods were uniform with a diagonal length of 200 nm. The SAED pattern of β-Ga_2_O_3_ nanorods confirms its growth direction perpendicular to (001) plane.

J. Zhang et al. [221] had grown β-Ga_2_O_3_ microrod arrays on a Si (100) substrate without any heterogeneous layers using a two-step hydrothermal method. The relatively larger interfacial energy between Si and GaOOH results into lack of nucleation sites, this problem was solved by the incubation of Si substrate in nucleation solution which is crucial to provide sufficient nucleation sites. The morphology of direct synthesized GaOOH microrods without nucleation stage was a one-site nucleated flower-like cluster. Ethanol reduces the surface tension, which reduces the contact angle between the solution cluster and the Si substrate. When the hydrothermal reaction was carried out directly in nucleation solution for a longer time of 12 h at 150 °C, the morphology resulted in spindle-like microrods consisting of an average length of 1.78 µm and width of 0.42 µm. The nucleation stage (with ethanol) followed by a growth stage (without ethanol) played a key role to get the array morphology of GaOOH microrods. The possibility of Ga^3+^ adsorption on the Si surface was greatly increased and density distribution sites were created. From the TEM studies, the β-Ga_2_O_3_ microrods were single crystalline and have a diameter of 768 nm with a length of 2.28 µm and the morphology of MRAs was shown in Figure 21.

A. Atilgan et al. [222] also used the same two-stage hydrothermal technique (nucleation and crystal formation) to create β-Ga_2_O_3_ nanoflakes on P Type-Si substrate. XRD studies showed strong peaks of (002), (111) related to crystalline β-Ga_2_O_3_ as same as in previous study [221].

D.Y. Guo [223] and C. Wu et al. [224] synthesized α/β-Ga_2_O_3_ phase junction NRAs hydrothermally on FTO conductive glass. The effect of concentration Ga(NO_3_)_3_ in solution and growth time on morphology of GaOOH NRAs was studied [223]. GaOOH NRAs were first annealed at 400 °C for 4 h, then at 700 °C for 20 min to produce α/β-Ga_2_O_3_ phase junction NRAs shown in Figure 22.

The tip of nanorods had a rhombus shape and the side length was in the range of 100–500 nm, with an average height of 1.5 µm. The band gaps of α-Ga_2_O_3_ and β-Ga_2_O_3_ were 4.96 eV and 4.66 eV, respectively. The flat potentials calculated according to the extrapolation of Mott-Schottky plots were −1.31 and −0.96 V. The α-Ga_2_O_3_ was more positive than that of β-Ga_2_O_3_ with the band offsets of (∆E_c_ = 0.35 eV, ∆E_ν_ = 0.05 eV), exhibiting type-Ⅱ band alignment, which indicates that the photogenerated electron-hole pairs are separated into α-Ga_2_O_3_ and β-Ga_2_O_3_ (holes in α-Ga_2_O_3_ and electrons in β-Ga_2_O_3_). Based on the band level differences, photogenerated electrons in a conduction band of α-Ga_2_O_3_ will transfer to the conduction band of β-Ga_2_O_3_ while photogenerated holes in valence band of β-Ga_2_O_3_ will transfer to the conduction band of α-Ga_2_O_3_.

C. He et al. [225] synthesized α-Ga_2_O_3_ nanorod arrays(NRAs) on SnO_2_/FTO substrate by a hydrothermal reaction and post-annealing treatment. S. Wang et al. [226] synthesized β-Ga_2_O_3_ nanorod arrays hydrothermally on a spin-coated Ga_2_O_3_ seed-layered FTO conductive glass. The seed layer was created by spinning an ethylene glycol monomethyl ether solution of ethanolamine and gallium isopropoxide at 3000 rpm for 15 s, followed by 30 min of air annealing at 450 °C. The as-prepared GaOOH NRAs were converted into β-Ga_2_O_3_ NRAs by calcination at 700 °C for 4 h. SEM studies showed that a highly dense flat surface of β-Ga_2_O_3_ NRAs aligned vertically on the FTO substrate. The nanorods have an average length of 1.287 µm and the tips of NRAs are in a diamond shape with diagonal lengths in the range of 100–500 nm. The optical band gap of NRAs estimated from UV-vis spectrum data was 4.63 eV.

J. Zhang et al. [227] synthesized a tree-like branched structure with α-Ga_2_O_3_ covered with γ-Al_2_O_3_ by a simple two-step hydrothermal method. Initially, they had synthesized GaOOH nanorod arrays hydrothermally on a spin-coated Ga_2_O_3_ seed layered FTO conductive glass. The GaOOH nanorods grown on (Ga_2_O_3_/FTO) substrate were divided into two parts. One part was annealed at 500 °C for 4 h to convert into α-Ga_2_O_3_ while the other was immersed in a solution prepared with different concentrations of Al (NO_3_)_3_.9H_2_O (i.e., 0.005 M, 0.01 M, 0.015 M) and 0.005 M C_6_H_12_N_4_ at 180 °C for 12 h as a second step in the hydrothermal process. Then, the sample was annealed at 500 °C for 4 h to obtain Ga_2_O_3_-Al_2_O_3_. Figure 23 indicates that the morphology of Ga_2_O_3_ changed significantly after hydrothermal treatment with 0.015 M aluminum nitrate. HRTEM image of Ga_2_O_3_ nanorods after the hydrothermal treatment with 0.015 M aluminum nitrate shows that the α-Ga_2_O_3_ nanorod was completely covered by γ-Al_2_O_3_; (110) peak from the XRD study indicates that the preferential growth direction of α-Ga_2_O_3_ on FTO is perpendicular to the (110) crystal plane. The presence of γ-Al_2_O_3_ may change the surface properties of α-Ga_2_O_3_, such as dangling bond configuration, surface energy, and surface roughness [241], which promotes the formation of nucleation sites for γ-Al_2_O_3_, resulting in the eventual emergence of a tree-like branch structure. The optical band gap of the α-Ga_2_O_3_ sample measured is 5.0 eV. The band gap of γ-Al_2_O_3_ is approximately 7.4 eV, and the valence band maximum position is estimated at 3.0 eV in the literature [242].

Since the band gap of γ-Al_2_O_3_ is significantly larger than that of α-Ga_2_O_3_, UV light with the wavelengths below 300 nm can penetrate the γ-Al_2_O_3_ and be absorbed by α-Ga_2_O_3_.

H. Ryou et al. [228] synthesized Sn-doped β-Ga_2_O_3_ nanostructures via a hydrothermal process. β-Ga_2_O_3_ nanostructures had Sn concentrations of 0 to 7.3 at.%. At fewer concentrations of Sn, β-Ga_2_O_3_ nanostructures had a rice-like morphology. As the Sn concentration increased, nanorods began to form and became thicker and longer at high Sn concentrations. The EDS elemental mapping of a nanostructure with 2.2 at.% Sn showed uniform distribution of Sn along β-Ga_2_O_3_ nanostructure that was beyond the maximum solid solubility of Sn in β-Ga_2_O_3_ [243]. The radii of Sn^4+^ ions (55~81 pm) comparable to that of Ga^3+^ (62 pm) can be substituted at the octahedral sites of β-Ga_2_O_3_. With an increase in Sn concentration from 0 to 7.3 at.%, the optical band gap of Sn-doped β-Ga_2_O_3_ nanostructures decreased from 4.8 eV to 4.1 eV.

R. Pilliadugula et al. [229] Sn doped β-Ga_2_O_3_ by doping Sn into a GaOOH matrix via the hydrothermal method. Sn doped β-Ga_2_O_3_ showed less weight loss in the thermogravimetric analysis compared to un-doped β-Ga_2_O_3_, which may be attributed to thermally active Sn molecules in the host material, which certainly increased the thermal stability of samples due to the strong bond nature of Sn compared to Ga [244]. In XRD studies, the 4 mol% Sn doped sample showed (110) and (310) orientations of SnO_2_ precipitate may be due to the dopant concentration exceeding the solubility limit of the host material. Sn doped samples showed minor shifts in 2θ positions of peaks due to the difference in ionic radii of Ga^3+^ and Sn^4+^. The 2 mol% Sn-doped sample had a smaller crystallite size of 12.48 nm, a cocoon-shaped morphology, and a band gap of 4.67 eV. The 4 mol%. The Sn-doped sample showed a rod-like morphology resulted from the agglomeration of nanoplate structures with an increased aspect ratio, which showed that Sn doping has altered the morphology of Ga_2_O_3_ by controlling the Ostwald ripening [245]. The surface area of the 2 mol%. Sn doped sample was 12.49 m^2^g^−1^, with a pore diameter of 11.64 nm.

J. Wang et al. [230] synthesized Ga_2_O_3_/Al_2_O_3_ composite materials by hydrothermal method. The precipitates were calcined for 3 h at 550 °C. In XRD studies of composite, no peak of Ga_2_O_3_ was observed due to the low concentration of Ga_2_O_3_ and Ga_2_O_3_ particles having good dispensability on the surface of Al_2_O_3_. TEM studies confirmed the (110) plane of Ga_2_O_3_. The morphology of Al_2_O_3_ was a nanosheet structure and Ga_2_O_3_ was nanorods of 26–28 nm long and 2.6–5 nm wide. The size of Ga_2_O_3_ in the composite was smaller because the growth of active material grains was inhibited by Al_2_O_3_ [246]. The slopes of MS plots were negative in the Mott-Schottky measurements which indicates that the Ga_2_O_3_/Al_2_O_3_ composite was P-Type.

S. Kim et al. [231] Al-doped β-Ga_2_O_3_ nanostructures by the hydrothermal method. Al-doped β-Ga_2_O_3_ nanostructures were obtained with the annealing of GaOOH nanostructures in O_2_ ambient at 1000 °C for 6 h. The compressive strain of Al-doped Ga_2_O_3_ nanostructures was compared with that of thin film β-Ga_2_O_3_ with the same Al concentration, and it was found that β-Ga_2_O_3_ nanostructures have a higher compressive strain due to a higher surface energy in nanomaterials [247]. Al doping affected the morphology of β-Ga_2_O_3_ nanostructures, with an increase in Al content the pH of the solution decreased which in turn changed the morphology of β-Ga_2_O_3_ nanostructure from spindle-like morphology to microrod structure along with a reduction in length. The BET surface area was unchanged at approximately 7.4–7.6 m^2^g^−1^ till the 1.2 Al at.%, then increases at higher Al concentrations attributed to morphology change with Al doping.

### 4.3. Ga_2_O_3_ by Chemical Bath Deposition (CBD)

The chemical bath method was a low-temperature process and widely used to synthesize thin films of Ga_2_O_3_. Morphologies depends on the type of Ga precursor and synthesizing conditions, such as pH of solution, reaction temperature, type of seed layer, and thickness of films, which depends on the deposition time. A schematic of the process is given in Figure 24, and complete literature for the synthesis of Ga_2_O_3_ by the CBD process is given in Table 5.

G. Li et al. [249] used a simple soft-chemical method to create Dy^3+^-doped gallium oxide hydroxides (GaOOH: Dy^3+^) of various morphologies (submicrospindles, submicroellipsoids, and 3D hierarchical microspheres). The morphology of GaOOH: Dy^3+^ changed from submicrospindles to 3D hierarchical microspheres of self-assembled nanoparticles as the pH increased from 4 to 9. The GaOOH: Dy^3+^ submicrospindles prepared at pH 4 had a high aspect ratio with a length 0.85 µm and a width of 0.2 µm. Due to the lowest surface defects, GaOOH: Dy^3+^ phosphors with ellipsoid shapes exhibit the highest emission intensity and Photoluminescence quantum yield (QY) among the four differently shaped samples under UV and electron beam excitation.

S. Fujihara et al. [214] synthesized c-axis oriented GaOOH films, consisting of rod-like particles deposited on various substrates through “chemical solution deposition”. Quartz glass with or without under layers of β-Ga_2_O_3_, SnO_2_, TiO_2_, MgO, and FTO coated glass were used as substrates. Heterogeneous nucleation of GaOOH resulted on all of these substrates. The nucleation resulted in a smaller number of precipitates formed on Quartz and β-Ga_2_O_3_ under layer substrates. This heterogeneous nucleation and deposition of film were enhanced with SnO_2_, TiO_2_, MgO, and FTO substrates. The rod-like particles of GaOOH were built-up closely and vertically on these substrates. The XRD studies of the GaOOH film deposited on SnO_2_ under layer confirmed that the rod-like particles were c-axis oriented perpendicular to the substrate. The XRD of annealed film confirmed that crystallinity of β-Ga_2_O_3_ increased with annealing temperature. The strong intensity peak (111) of β-Ga_2_O_3_ suggests that the film is oriented along [111] direction perpendicular to the substrate.

T. Mizunashi et al. [249] used an aqueous chemical bath deposition to synthesize β-Ga_2_O_3_:Eu^3+^ films. The presence of a relatively strong (111) peak (2 = 35.2°) suggests that the films are oriented somewhat perpendicular to the substrate along the [111] direction. When 3.0 and 5.0 at.%. Eu^3+^ doped films are exposed to 271 nm UV light, relatively strong line emissions appeared in the wavelength range of 570 to 630 nm. These emissions can be attributed to transitions of the Eu^3+^ ions from the excited 5D_0_ level to the ground 7F_J’_ (J′ = 0, 1, 2) levels. The strongest emission at 611 nm caused by electric-dipole 5D_0_ → 7F_2_ transition, indicates the presence of Eu^3+^ at a site with no inversion symmetry in β-Ga_2_O_3_. A larger Eu^3+^ ion would favor the site with a higher coordination number, but the octahedral site in β-Ga_2_O_3_ has inversion symmetry. Therefore, it is possible that the octahedral site undergoes distortion by the Eu^3+^ substitution for Ga^3+^ or Eu^3+^ is located in the tetrahedral site. A different spectrum was obtained when films were excited with a shorter wavelength of 254 nm. A broad blue-green emission band appears with a peak wavelength of approximately 451 nm, and emissive defect centers remain in the host of β-Ga_2_O_3_, demonstrating that the PL spectrum is tunable with different excitation wavelengths and Eu^3+^ concentrations.

C. Y. Yeh et al. [250] synthesized Ga_2_O_3_ thin films on glass substrates using the chemical bath method and a post-annealing process. As the concentration varied from 0.025 to 0.075 M, the aspect ratio of GaOOH nanorods was 3.23, 3.76, and 5.06, respectively. α-Ga_2_O_3_ film annealed at 500 °C had high crystallinity with less FWHM in the XRD pattern.

G. Hector et al. [251] synthesized β-Ga_2_O_3_ microrods by annealing of α-GaOOH microrods that were deposited on Si substrate by chemical bath deposition. As Ga(NO_3_)_3_.xH_2_O concentration varies (i.e., 0.015, 0.025, and 0.05 M), three types of rod-like morphologies formed directly on Si. The 0.015 M concentration resulted in plate-layered rods (length 5 µm, diameter 1.2 µm) with a hierarchical structure having a few secondary nucleation at the base. The 0.025 M concentration resulted in round-plate rods (length 7.25 µm, diameter 2 µm) with a hierarchical structure having a few secondary nucleations at the base. As the concentration increased more than 0.05 M, the morphology changed to round rods with no secondary nucleation. The aspect ratio of microrods decreased with increasing concentration and reached a maximum value of 3.30 at the highest concentration. The apparent density of rods is much less (1 rod/100 µm^2^).

J. L. Chiang et al. [252] synthesized the α-Ga_2_O_3_ nanorods on a ITO (95% In_2_O_3_ + 5% SnO_2_) coated glass substrate. A 200 nm thick ITO material was coated on a glass substrate by E-beam deposition. The GaOOH nanorods have a length of 500–700 nm. Based on XRD studies, the (111) of GaOOH with less FWHM was resulted for a 1:1 concentration ratio sample. The (104) of α-Ga_2_O_3_ was the major orientation in all the samples.

### 4.4. Other Methods

Apart from the above-mentioned methods, the other methods to synthesize Ga_2_O_3_ materials were solvothermal, forced hydrolysis, reflux condensation, and electrochemical deposition. The complete literature for the synthesis of Ga_2_O_3_ by these four processes was given in Table 6.

#### 4.4.1. Solvothermal Method

The solvothermal synthesis process was similar to the hydrothermal process, except that the solution used for synthesis was non-aqueous.

G. Sinha et al. [253] synthesized pure and rare-earth doped Ga_2_O_3_ nanoparticles by a mixture of ethylene glycol (EG) and water as solvent. Pure and co-doped (0.01% Eu^3+^ and 0.01% Tb^3+^) Ga_2_O_3_ nanoparticles were prepared with the same procedure. The PL spectra of nanoparticles showed that a bright white emission was generated due to energy transfer from (Ga_2_O_3_-blue) host to rare-earth ions (Eu^3+^ (green) and Tb^3+^ (red) ions). In their other study [254], they directly synthesized 3D micro structured β-Ga_2_O_3_ films on silicon (Si) substrate via an ethylenediamine (En)-mediated solvothermal process. The XRD analysis of films revealed that the films were well-crystalline in nature and a phase transformation occurred from α-GaOOH to β-Ga_2_O_3_ at a volume ratio of 60:40. For a given volume ratio of En to water, increasing the reaction time results in increased crystallinity. The TEM image showed that the morphology of a 40:60 sample with a reaction time of 2 h was composed of nanowires that are attached along their edges due to plastic deformation. With increasing the reaction time to 4 h, the morphology changed to branched nanoflakes of 8 nm width. The FESEM image of a 50:50 sample showed that the morphology becomes a 2D interconnected nanoflakes structure with a flake thickness of 15 nm, and these structures were assembled together to form 3D microstructures of an order of 0.4 µm in diameter for a 60:40 sample. The diameter becomes 2 µm with flakes thickness of 20−60 nm for an 80:20 sample. It was assumed that the En molecules interact with the hydroxyl groups of the precursors, allowing for complete dehydration of gallium hydroxide and the formation of β-Ga_2_O_3_ at lower temperatures of 200 °C

M. Muruganandham et al. [255] synthesized mesoporous α-GaOOH by the solvothermal method using anhydrous ethanol as a solvent. The GaOOH particles synthesized at 200 °C for 10 h had semi-nanosphere morphology with a dense core and thin shell-like structures. The particle size ranged from 100 to 500 nm and had a high surface area of 193 m^2^g^−1^. The GaOOH, prepared at less than 200 °C, did not yield semi-nanosphere morphology. The GaOOH prepared at a 5 h reaction time showed that the formation of morphology was incomplete. The semi-nanosphere morphology GaOOH had not obtained using water or water-ethanol as solvent. The high pressures resulted from the low boiling point of ethanol (78 °C) in the solvothermal process could be the reason for the different morphologies from the corresponding hydrothermal process.

W. Zhang et al. [256] synthesized γ-Ga_2_O_3_ nanoparticles by the solvothermal method and incorporated them into liquid metal/metal oxide (LM/MO) frameworks.

#### 4.4.2. Forced Hydrolysis Method

B. K. Kang et al. [257] synthesized monodispersed β-Ga_2_O_3_ nanospheres via morphology-controlled gallium precursors using a simple and reproducible urea-based forced hydrolysis method, followed by thermal calcination processes. The rod-like structures of GaOOH (i.e., diameter 200–400 nm and length of 1.5–2 µm) were obtained at a concentration ratio of 0 (i.e., without sulfate ions). The crystalline rods of GaOOH were obtained due to aggregation of Ga(OH)_4_ aqueous solution, which was a precursor obtained by decomposition of urea at 80 °C and releasing uniform ${\mathrm{OH}}^{-}$ ions into a Ga(OH)_3_ solution. As the concentration ratio R increased to 0.33, a monodispersed spherical amorphous precursor Ga_4_(OH)_10_SO_4_ formed in the solution due to the higher affinity of ${\mathrm{SO}}_{4}^{2-}$ ions than ${\mathrm{NO}}_{3}^{-}$. These spheres were agglomerated as R increased. The as-synthesized amorphous NSs were converted to uniform polycrystalline β-Ga_2_O_3_ NSs (diameter 200 nm) by annealing at 1000 °C. In their following study [258], they investigated the effect of ageing time on the morphology of gallium precursors, while fixing the concentration ratio R = 0.33 and varying the ageing time from 2–10 h. After increasing the reaction time to 6 h, the α-GaOOH microrods structure with preferential growth direction [001] was dominant. The size of α-GaOOH microrods increased as the size of NSs decreased, indicating that the NSs were dissolved in the solution and aggregated on the surfaces of the microrods. When the reaction time was increased to 10 h, the NSs vanished and the morphology changed completely to well-crystalline α-GaOOH microrods with branched rods growing along {001} planes of primary rods. This growth mechanism is illustrated in Figure 25.

E. Huang et al. [259] synthesized GaOOH via forced hydrolysis from a solution containing different ionic ratios of Ga(NO_3_)_3_·xH_2_O and sodium nitrate (NaNO_3_).
$$\mathrm{Ionic-strength}(I)=\frac{1}{2}\;\sum_{B}m_{B}{z_{B}}^{2}\;\mathrm{where}\;m_{B}\;\mathrm{is}\;\mathrm{the}\;\mathrm{molality}\;\mathrm{of}\;\mathrm{ion}\;B\;\mathrm{in}\;\mathrm{the}\;\mathrm{solution},\;Z_{B}\;\text{is its charge number (Negative for anions and positive for cations).}$$

When R increased from 0.2 to 1, the relative crystallinity of α-Ga_2_O_3_ increased distinctly and decreased with a further increase in R-value. The optimized ionic strength was 1, which has high crystallinity. The morphology of Ga_2_O_3_ without the addition of NaNO_3_ (R = ∞), was spindle-like nanorods having two narrow sides and a wide center. With R = 3 and 2, Ga_2_O_3_ appeared as nanorods of uniform width. At R = 1, Ga_2_O_3_ crystals transform into a dumbbell shape with wide sides and a narrow center. From R = 0.5 to 0.3, nanorods exfoliated to nanoplates, strips and fragments. This morphology-controlled synthesis was illustrated in Figure 26. At R = 0.2, Ga_2_O_3_ was fully broken into fragments.

The highest absorption with an edge up to 360 nm is obtained with the α-Ga_2_O_3_ sample at R = 1, indicating its highest defect site concentration in the UV-vis spectra of α-Ga_2_O_3_ synthesized at different R values. The highest photoluminescence intensity (One signal at 460 nm-blue, the other is at 600 nm-orange) is obtained for α-Ga_2_O_3_ prepared at R = 1, which is several times greater than the commercial α-Ga_2_O_3_ sample. Surface areas of Ga_2_O_3_ samples synthesized at different R values are measured to be in the range of 30–50 m^2^g^−1^, which is significantly higher than the surface area of commercial Ga_2_O_3_ (15 m^2^g^−1^).

#### 4.4.3. Reflux Condensation Method

K. Girija et al. [260] synthesized self-assembled novel floral β-Ga_2_O_3_ nanorods via the reflux condensation method. The precipitate calcined for 3 h at 500 °C was changed to crystalline α-Ga_2_O_3_ and calcined at 900 °C was changed to crystalline β-Ga_2_O_3_. Figure 27a shows the SEM image of a self-assembled pattern (i.e., size ~1 µm) made by individual nanorods of size 100 nm. As shown in Figure 27b, the TEM image suggests the single crystalline nature of nanorods. From the UV-vis diffuse reflectance spectrum, it was found that the absorption edge was at 280 nm and the band gap E_g_ of β-Ga_2_O_3_ nanorods was found to be 4.59 eV. When excited at 254 nm, the PL spectrum showed a strong UV luminescence peak centered at 360 nm and a small peak at 302 nm.

In their other study [261], they prepared single crystalline β-Ga_2_O_3_ nanorods by a facile reflux condensation method using Ga(NO_3_)_3_·nH_2_O and anionic surfactant CTAB as starting materials. Time-dependent experiments were carried out to investigate the morphology evaluation. The initial molar concentration of surfactant was as follows: metal ions were set at 5:1, and the reaction time ranged from 3 to 12 h. At 3 h, an agglomerated rod-like structure was observed, as seen in Figure 28a; when the reaction time was increased to 6 h, nanorods were formed, indicating the presence of Ostwald ripening. The nanorods were non-uniform and randomly aligned, as seen Figure 28b. When the reaction time is increased to 12 h, the nanorods become uniform and side by side aligned, as seen in Figure 28c. The nanorods measured 200 nm in length and 50 nm in diameter.

To study the effect of CTAB concentration on morphology, three concentration ratios (CTAB: metal ion—1:2, 5:1 and 50:1) were chosen while the reaction time was fixed at 12 h. At low concentrations of CTAB (ratio—1:2), the morphology was composed of agglomerated nanoparticles as shown in Figure 29a. At higher concentrations (ratio: 5:1), the morphology became side-to-side aligned nanorods and at very high concentrations the nanorods lost uniformity and alignment, as shown in Figure 29c.

#### 4.4.4. Electrochemical Deposition

J. J. M. Vequizo et al. [262] synthesized α-Ga_2_O_3_ thin film on FTO substrate by electrochemical deposition and a post-annealing process. A cyclic voltammetry (CV) experiment was performed to determine the best-applied voltage for various deposition times (2–10 min). A cathodic peak appeared at approximately −0.68 V, but no significant thickness was achieved for potentials greater than −1.0 V, indicating that a large over-potential is required to activate the reaction on the FTO substrate. As a result, the chosen potential values are greater than or equal to −1.0 V. A higher amount of H_2_O_2_ in solution results in a high non-uniformity and roughness of films due to the bubbling effect or due to oxygen gas generation during reaction. Hence, an optimal quantity (i.e., 0.13 mL) of H_2_O_2_ was chosen based on uniform thickness of the film. A deposition voltage of −1.1 V for a deposition time of 3 min and a deposition voltage of −1.0 V for a deposition time of 3–10 min produced a crack-free surface of films.
$$H_{2}O_{2}+2e^{-}\;\;\to \;2{\mathrm{OH}}^{-}$$

When a cathodic voltage was applied, the ${\mathrm{OH}}^{-}$ ions generated near the cathode by electrochemical reduction could be used to generate GaOOH at room temperature via the reaction of ${\mathrm{OH}}^{-}$ with Ga^3+^. The films converted into α-Ga_2_O_3_ at the temperature range of 500–600 °C.

N. M. Ghazali et al. [263] synthesized β-Ga_2_O_3_ nanostructures on Si (100) substrate. The possible reactions during the electrochemical deposition process are summarized below:$$\mathrm{At}\;\mathrm{cathode}:\;4{\mathrm{cl}}^{-}+4H_{2}O\;\to \;4\mathrm{HCl}+4{\mathrm{OH}}^{-},\;2\;\mathrm{Ga}3^{+}+4{\mathrm{OH}}^{-}+2e^{-}\;\to \;{\mathrm{Ga}}_{2}O_{3}+H_{2}O+H_{2}\;\mathrm{Anode}:\;H_{2}O\;\to \;½\;O_{2}+2H^{+}+2e^{-}$$

At a low molarity of 0.1 M, nanodot-like structures of Ga_2_O_3_ were deposited on Si substrate in both acidic and alkaline conditions. The dot structure has a high density in the acidic medium, as shown in Figure 30a. When molarity was increased to 0.5 and 1.0 M, nanorods started to form at a low pH level while a nanodot structure without or with less density of nanorods was obtained at a high pH level. High-density nanorods of a length of 200–500 nm and a length of 2–4 µm were obtained at pH 4, as shown in Figure 30b.

## 5. Applications

In this chapter, the applications of Ga_2_O_3_ will be discussed from various solution processes (including hydrothermal, sol-gel, chemical bath deposition, etc.). This chapter will contain deep ultraviolet photodetector, gas sensors, pH sensors, thin film transistors, and other applications. By optimizing the process parameters, Ga_2_O_3_ material synthesized through various solution processes can be applied to the sensors for better response characteristics. However, due to the influence of human operation, environment, or other factors, the reliability and repeatability of sensors fabricated by the solution process are not as stable as the vacuum process.

### 5.1. Deep-Ultraviolet Photodetectors (PD)

In the past, ultraviolet (UV) detection was applied in many fields, such as civilian, military, and scientific research. The ultraviolet spectral region is defined as λ = 400–10 nm. According to ref. [264], it is usually divided into the following subdivisions: near ultraviolet (NUV, 400–300 nm), mid ultraviolet (MUV, 300–200 nm), far ultraviolet (FUV, 200–122 nm), and extreme ultraviolet (EUV, 122–10 nm). According to another definition, it can also be divided into four regions namely ultraviolet A, B, C, and vacuum ultraviolet (VUV). A detailed description can be found in reference [265].

It is generally recognized that sunlight is the most important source of natural ultraviolet. It can emit ultraviolet radiation of various wavelengths. However, all UVC radiation from the sunlight is absorbed by the diatomic oxygen or ozone in the atmosphere [265]. In addition, vacuum UV is almost absorbed by molecular oxygen in the air. Therefore, a range of 200–280 nm of the UV region is called “solar blind”, which means that it can be detected without interference from solar radiation. In addition, the characteristics of solar-blind detectors are that they possess a cutoff wavelength below 280 nm because they only respond to ultraviolet radiation with wavelengths shorter than the solar radiation that can penetrate Earth’s atmosphere. If they are exposed to normal outdoor lighting, they will not produce a measurable signal. Solar-blind DUV-PDs with excellent thermal stability and reliability have been widely applied in many fields, such as ozone monitored, flame detection, communications, military defense, biochemistry, ultraviolet leakage, and other fields [266].

There are many scientific reports and much literature that mention solar-blind photodetectors made from wide-band gap semiconductors [266,267,268,269,270,271,272,273,274,275,276,277,278,279], such as GaN [266], ZnO [267], ZnS [268], ZnSe [269], SiC [270,271], AlN [272], diamond [273,274], BN [275], Ga_2_O_3_ [276,277,279], Al_x_Ga_1−x_N [278], and so on. These materials are also applied to high-power and high-temperature applications [280,281] because they have high breakdown field strength. However, to meet the standard for solar-blind photodetectors with a cut-off wavelength of less than 280 nm, it is necessary for alloy engineering to adjust the band gap to 4.42 eV, and Ga_2_O_3_ has the potential as a solar-blind DUV detector. Because the monoclinic structure of β-Ga_2_O_3_ has a wide band gap of 4.4–4.8 eV [276] (corresponding to a wavelength of 258–280 nm) and its solar-blind sensitivity is expected to cover most solar blind ultraviolet regions [265]. Besides the thin film type, Ga_2_O_3_ nanorods and nanowires also synthesized through hydrothermal [274], sol-gel [275], chemical bath deposition [250,251,252], and so on. This section will discuss photodetectors made from solution-based gallium oxide (Ga_2_O_3_).

Y. Kokubun et al. [181] synthesized the β-Ga_2_O_3_ on sapphire substrate by the sol-gel method at 600–1200 °C different annealing temperatures. They analyzed the effect of annealing temperature on the spectral responses of photoconductive detectors. As shown in Figure 31, the top surface of film is exposed to radiation. It can be seen that the Ga_2_O_3_ films annealed at 600 and 800 °C, the photocurrent peak appears at 250 nm, and no photocurrent was observed greater than 280 nm. As the annealing temperature exceeds 900 °C, the photocurrent peak shifted to shorter wavelengths. The maximum value of responsivity ~8 × 10^−5^ A/W occurred for the film annealing at 1000 °C. It indicated that the β-Ga_2_O_3_ films prepared by the sol-gel process have potential applications as solar-blind photodetectors.

In addition, Rikiya Suzuki et al. [178] prepared epitaxial grown β-Ga_2_O_3_ thin films on a (100) β-Ga_2_O_3_ substrate using the sol-gel method. They prepared photodiodes with and without the sol-gel prepared β-Ga_2_O_3_ cap layer. It is revealed that the Au/β-Ga_2_O_3_/β-Ga_2_O_3_ structure of photodiodes possesses a lower leakage current. From the dark I-V curves of photodiodes, it was revealed that the cap layer was highly resistive having a resistivity of 10^6^ Ω-cm. The photodiode with a cap layer operated at a 5.4 V turn-on voltage, which was more than that of the photodiode without a cap layer (i.e., 1.6 V). Figure 32 shows the spectral response of Ga_2_O_3_ photodiodes with and without a cap layer at reverse and forward biases of 3 V. Under the forward bias of 3 V, a maximum responsivity of 4.3 A/W was noted for the β-Ga_2_O_3_ photodiode with a cap layer.

H. Shen et al. [188] and M. Yu et al. [189] have also used the sol-gel technique to synthesize Ga_2_O_3_ thin films on sapphire substrate. The (I-V) test and time-resolved photo response of the photo-detector fabricated with β-Ga_2_O_3_ thin film annealed at 700 °C, showing a high I_photo_/I_dark_ ratio of 18.34 along with the fast rise time of 0.10 s and decay time of 0.10 s [188]. In addition, the photodetector made by α/β polycrystalline Ga_2_O_3_ thin films had a low-dark current (18 pA), an I_photo_/I_dark_ ratio of 1664, detectivity(D*) of 5.41 × 10^11^ Jones, and a fast photo-response speed (that is rise time(t_r_) of 0.03 s/0.23 s and decay time(t_d_) of 0.04s/0.41s) [189].

In addition, many research reports mentioned Ga_2_O_3_ nanorod arrays (NRAs) synthesized by a hydrothermal reaction [224,225,226]. S. Wang et al. [226] synthesized β-Ga_2_O_3_ nanorod arrays (NRAs) hydrothermally on a spin-coated Ga_2_O_3_ seed layered FTO conductive glass. A Ti/Au electrode was formed on β-Ga_2_O_3_ NRAs to build a solar-blind DUV photodetector. Figure 33 shows the device structure and the current-voltage characteristic curves. The PD was operated without and with light (i.e., 254 nm illumination) at various light densities. When the PD operated at 0 bias and 1.2 mW/cm^2^ light intensity, it exhibited an I_photo_/I_dark_ ratio of 9.14, a responsivity of 10.80 mA/W, an external quantum efficiency (EQE) of 5.27 × 10^−3^, and rise/decay times of 0.64 s/0.38 s, respectively. In addition, the photo response switching behaviors at various bias voltages were also observed. Obviously, the I_dark_ and I_photo_ increased with increasing the bias voltage from 0 V to 1.5 V. At 1.5 V, a I_photo_ of 17.60 pA was observed, which is approximately 11 times that at 0 V. Furthermore, the I_photo_ increased linearly with the electric field due to more electron-hole pair separation and more transport excite of photogenerated electron-hole pairs, resulting in a higher photocurrent [226]. These experimental results indicate that the PD possesses high conversion efficiency and low power consumption properties. It has the potential for use in solar-blind photodetector application.

Furthermore, α-Ga_2_O_3_ nanorod arrays (NRAs) via a hydrothermal process and thermal annealing was proposed by C. He et al. [225], where the p-type Cu_2_O microspheres were grown on the α-Ga_2_O_3_ nanorod arrays. The α-Ga_2_O_3_/Cu_2_O heterojunction was a p−n junction structure that could effectively enhance the photo-induced carriers and the range of light detection. The PEC-type photodetectors fabricated with an α-Ga_2_O_3_/Cu_2_O p−n junction structure displayed distinguishable spectral signatures with opposite photocurrent responses when illuminated by UV light at 254 and 365 nm. It has the potential to be developed into a self-powered PD capable of distinguishing between ultraviolet and visible bands. When exposed to 254–365 nm UV light, the PD showed a superior responsivity region of 0.42–0.57 mA/W. The PD showed a fast response of (t_r_~103 ms) under 365 nm UV light. C. He et al. provide a solution to distinguish different illumination bands through a photodetector with heterojunction.

D.Y. Guo [223] synthesized α/β-Ga_2_O_3_ phase junction NRAs hydrothermally on FTO conductive glass. To make a solid-state type photodetector a simple structure of Ti/Au electrode was deposited on the top of NRAs. The solar blind solid-state photodetector showed rectifying characteristics as observed in I-V curve. Photogenerated carriers cannot be separated effectively in this structure of device because only the upper end of nanorods will be in contact with electrode while the perimeter of nanorods will be contactless. Under 0 V bias and at the constant light intensity, the photocurrent from α/β-Ga_2_O_3_ photoelectrochemical-type photo detector was more than that of α-Ga_2_O_3_ photo detector. Both α-Ga_2_O_3_ and α/β-Ga_2_O_3_ phase junction photo detectors had showed linear relationship between the photo current and light intensity.

A solar-blind photodetector of vertical α/β-Ga_2_O_3_ phase junction NRAs was built on FTO glass via the hydrothermal method that has been proposed by C. Wu et al. [224]. To get a large detection range, a 0.5 × 1 cm^2^ mono layered graphene-silver nanowire (Ag NWs of 150 nm in diameter and a length several tens of hundreds of µm) hybrid film was synthesized on top of NRAs. A 0.005 × 0.05 cm^2^ circular Ag electrode was deposited on the corner of hybrid film for point connection for external circuit. The experimental results found that the current of the photodetector in the dark was 1.72 nA under 254 nm UV light with 0 V, and the current increased to 211.07 nA when the light intensity was 3000 µW/cm^2^. The responsivity decreases from 0.26 mA/W to 0.14 mA/W due to the increased probability of electron-hole pair recombination at higher light intensities. The maximum value of detectivity (D) of 2.8 × 10^9^ jones (cm Hz^1/2^ W^−1^) occurred at a light intensity of 100 µW/cm^2^ and decreased with an increase in light intensity. In addition, the photodetector showed a high I_photo_/I_dark_ ratio of 2.1 × 10^2^ and a rise time of 0.54 s. Furthermore, the photodetector operated in a vacuum environment (~1 Pa), the I_photo_/I_dark_ ratio of 3.48 was lower than that of atmospheric conditions, and it also exhibited a faster response of rise time of 0.16 s.

A. Atilgan et al. [222] mentioned that β-Ga_2_O_3_ nanoflakes were synthesized on p-Si substrate by the hydrothermal process. The back electrode covering the entire back side of the device was made by screen-printing of Ag paste. An Ag/Al/Au ohmic electrode was fabricated on the 1 × 1 cm^2^ selective area of β-Ga_2_O_3_ nanoflakes to make a heterojunction β-Ga_2_O_3_/p-Si photodiode. The nanoflake structure has a main advantage, it can effectively improve the electron-hole pairs upon UV illumination at the junction region. The device showed rectifying characteristics due to heterojunction and had a high rectifying ratio of 1387 at ± 10 V. The photoresponsivity is significantly different at 0 V (0.16 mA/W) and at 10 V (17.1 A/W). It can be shown that the β-Ga_2_O_3_ nanoflakes have potential in self-powered photodiode applications.

Recently, the tree-like branched structure of α-Ga_2_O_3_ covered with γ-Al_2_O_3_ was synthesized using a simple two-step hydrothermal method by J. Zhang et al. [227]. They succeeded in fabricating a PEC self-powered UV detector. The responsivity of the UV detector made with a Ga_2_O_3_-Al_2_O_3_ heterojunction and α-Ga_2_O_3_ nanorods was 0.174 mA/W and 0.129 mA/W, respectively. In addition, the high I_photo_/I_dark_ ratio of 51.3 and 14.5 was also determined, respectively. The experimental results found that the detection performance could be improved.

According to the above-mentioned, Ga_2_O_3_ materials by the solution process have *successfully* been applied as a photodetector. It possesses a low-cost process, low power consumption abilities, a high I_photo_/I_dark_ ratio, a fast response, and so on. Its superior properties can be a promising candidate for solar-blind photodetection applications.

### 5.2. Gas Sensors

Resistive gas sensors have been under development for a long time. Ga_2_O_3_ is one of the most important materials for high temperature gas sensing due to its high melting point (~1800 °C). S. J. Pearton et al. [7] reported on the processing and devices of gallium oxide material in 2018, which mentioned the application of gallium oxide gas sensors. In general, polycrystalline Ga_2_O_3_ materials can be used either as a gas sensor for sensing oxygen (T ≥ 900 °C) or reducing gases (T < 900 °C), depending on the working temperature. In addition, the advantage of monoclinic β-Ga_2_O_3_ is very stable, which can be easily prepared with annealing temperatures at 800–900 °C.

For high-temperature applications, such as internal combustion engines or furnace installations, Ga_2_O_3_ gas sensors have been used to monitor the composition of exhaust gases. Ga_2_O_3_ thin films possess the ability to sense oxygen by the resistivity of changes with the concentration of oxygen. S. J. Pearton et al. have summarized the gas sensor applications, operating temperature range, and detection ranges for Ga_2_O_3_ gas sensors [7]. According to many reports, the Ga_2_O_3_ thin films for gas sensors can be deposited by sputtering, reactive thermal evaporation, the sol-gel process, the synthesis method, and others. In this review, we have organized and discussed the preparation and gas applications of gallium oxide materials synthesized through solution processes.

Y. Li et al. [179] have investigated the gas sensing performance of sol-gel prepared Ga_2_O_3_ thin film doped with Ce, Sb, W, and Zn. It was concluded that the sensor doped with Zn showed O_2_ gas sensing performance at operating temperatures below 420 °C. Ce doped sensor showed a lower response yet a fast response time (40 s). The W doped sensor showed a higher response with O_2_ gas and the Sb doped sensor showed the least response compared to other sensors. In addition, the experimental results show the dynamic response of the doped Ga_2_O_3_ gas sensor when exposed to a pure N_2_ baseline and a 1000 ppm oxygen pulse at 500 °C. Furthermore, they also found that the sensor doped with Sb possesses a higher base resistance than the sensors doped with other materials. Meanwhile, sensors doped with Ce, W, and Zn possess a base resistance of similar orders of magnitude. In summary, the sensors made by Ga_2_O_3_ films doped with Sb, Ce, W, and Zn possess a superior response, stability, and repeatability when operated at various temperatures [179].

A. Trinchi et al. [282] mentioned that a kind of Schottky diode (Pt/Ga_2_O_3_/SiC) can be characterized for H_2_ sensing applications. Its sensing characteristic is correlated with the operating temperature. In this research, Schottky diodes possess more superior advantages than the pure Ga_2_O_3_ thin film of conductometric sensors. Among these, the Ga_2_O_3_ materials were prepared by the sol-gel process and deposited onto the transducers by spin-coating. The sensor placed under cycling the ambient from synthetic air (SA) to 1% H_2_ in SA air, produces repeatable changes at fixed forward bias. When the sensors were operated above 500 °C, they possessed a fast response time. In addition, the response of the sensors show good stability in H_2_ sensing applications with a 210 mV drop in bias voltage for 1% H_2_ [282].

M. R Mohammadi et al. [187] mentioned that the mesoporous TiO_2_ and Ga_2_O_3_ thin films was synthesized on quartz and alumina substrate transducers with various Ti:Ga atomic ratios using the particulate sol-gel method. Due to a high specific surface area, the gas sensor made with binary oxide (atomic ratio Ti/Ga = 1) annealed at 600 °C had shown the highest response toward the gases of CO and NO_2_.

H. J. Lin et al. [219] deposited a 50 nm-thick SnO_2_ seed layer on a 1 µm SiO_2_ insulating layer Si (100) substrate by sputtering. Afterwards, they synthesized pure β-Ga_2_O_3_ nanorod arrays (NRAs) by the hydrothermal method on the Sn/SiO_2_ substrate. Pt nanoparticles by dip-coating or LSFO (La_0.8_Sr_0.2_FeO_3_) nanoparticles by sputtering were decorated on this β-Ga_2_O_3_ nanorod array, done post-annealing to improve crystallinity. This structure was used as a gas sensor for CO gas detection at various concentrations (20, 50, 80, and 100 ppm in N_2_ balance). Those two kinds of sensors showed same performance at low concentrations. LSFO (La_0.8_Sr_0.2_FeO_3_) nanoparticles coated β-Ga_2_O_3_ nanorod arrays showed a fast response over that coated by Pt nanoparticles.

R. Pilliadugula et al. [216] synthesized β-Ga_2_O_3_ nanorods by the hydrothermal method. These calcined nanorods were mixed with ethanol at room temperature and drop casted onto a silver coated glass substrate, with a period of wait until the ethanol was completely dry. Afterwards, the films were sintered at 100 °C to improve adhesion. The morphology of calcined nanorods revealed that the porosity was increased with an increase in calcination temperature. These pores are favorable for CO_2_ gas sensing. In addition, surface area studies were obtained from BET—N_2_ adsorption-desorption studies. The BET surface areas for GaOOH and β-Ga_2_O_3_—1000 °C samples were 6.7494 m^2^g^−1^ and 6.1034 m^2^g^−1^. These two samples were compared for CO_2_ gas sensing. It was seen that the sensing response was linear for both samples. For all concentrations of CO_2_, the β-Ga_2_O_3_—1000 °C had shown a higher sensing response and better repeatability when compared to GaOOH due to its low band gap and high pore density.

Furthermore, R. Pilliadugula et al. [217] fabricated β-Ga_2_O_3_ sensing films over IDE pattern printed on glass substrates by drop casting. The films were dried at 80 °C for 10 h to allow complete evaporation of the NMP solvent from the film, and heated at 250 °C for 2 h to improve adhesion. Of all the samples, GO—5 and GO—14 have a high aspect ratio of 5.88 and 6.89. The excess alkaline environment facilitated the directional attachment of these high aspect ratio particles. It was found that the GO—5 sample has highest band gap of 4.75 eV; the band gap was influenced with pH value and decreased with increasing pH. Meanwhile, the GO—14 sample had the lowest band gap of 4.61 eV, and its high aspect ratio nanoparticles morphology showed an enhanced sensing response at all NH_3_ vapor concentrations between 25–200 ppm, except at the concentrations below 100 ppm. In addition, R. Pilliadugula et al. [229] also proposed Sn doped β-Ga_2_O_3_ by doping Sn into GaOOH matrix by the hydrothermal method. The experimental results found that the 2 mol% Sn doped sample had a superior surface area and exhibited an enhanced sensing response property at 25 ppm to 200 ppm of NH_3_.

In addition, J. Wang et al. [230] synthesized Ga_2_O_3_/Al_2_O_3_ composite materials by the hydrothermal method. The composite showed a 58.2% response to NO_x_—100 ppm at room temperature, which was 6.8 times higher than pure Ga_2_O_3_ prepared under the same hydrothermal conditions.

B. Zhang et al. [220] has successfully synthesized β-Ga_2_O_3_ NRAs. The La based LSCO perovskite oxide (La_0.8_Sr_0.2_CoO_3_) catalytical nanoparticles of two different thicknesses (3 and 8 nm) were deposited on β-Ga_2_O_3_ NRAs by RF magnetron sputtering. This structure was subjected to NO_2_ gas sensing at various concentrations at 800 °C. By LSCO nanoparticle decoration, the NO_2_ response of β-Ga_2_O_3_ NRAs can be enhanced by more than 500%. After incorporation of LSCO, it has a faster response time but a longer recovery time. For high-temperature detection, the device possesses 0.1625 ppm^−1^ at 800 °C, the sensitivity is very high. Because the LSCO/β-Ga_2_O_3_ structure is a p–n heterojunction, the charge carrier diffusion occurs at the junction, amplifying the overall n-type response of the heterojunction. Therefore, the LSCO nanoparticles facilitate detection of NO_2_ gas and improve its response speed at high temperatures.

According to the above-mentioned gas sensor applications of Ga_2_O_3_, by modifying the solution process parameters, Ga_2_O_3_ can be used to detect O_2_, H_2_, CO, CO_2_, NO_x_, NO_2_, NH_3_, and so on. Ga_2_O_3_ materials were synthesized using a solution process and have a low temperature process, fast response, high potential for use in gas sensing, and superior sensing characteristics.

### 5.3. pH Sensors

In the past decade, public health problems caused by irregular lifestyles and unhealthy diets have become a global concern. One of the established health biomarkers is a stable pH, which is associated with all tissues of the body. Hence, biosensors for pH sensing have attracted significant interest in the scientific community [283]. Therefore, several metal-oxide materials have been used as sensing materials, widely applied to pH or biochemical sensors. SnO_2_, ZnO, RuO_2_, WO_3_, ITO, TiO_2_, ZrO_2_, Ta_2_O_5_, HfO_2_, and Ga_2_O_3_ have been widely used as sensing materials for pH sensors [252,283,284]. Among oxide materials, Ga_2_O_3_ is a wide band gap material that has attracted considerable attention in very recent years because it possesses many advantages (including nontoxicity, thermal stability, and excellent chemical and ideal biocompatible properties) [252,283,284]. In addition to being widely used in photodetectors, gas sensors, photocatalysis, and photodegradation, gallium oxide is also used in pH sensors. According to ref. [284], micro- and nanostructured sensing materials have a superior ratio of surface/volume. These materials were applied in electrochemical and biomedical sciences, and other scientific areas, possessing good potential applications. They can enhance sensing response and select ability and catalytic activity; thus, as a pH sensing material, it possesses many applications in life, the medical industry, environmental pollution, and so on because of its fast response, wide testing range, high sensitivity, and easy and low-cost fabrication.

J. L. Chiang et al. [252] proposed synthesized α-Ga_2_O_3_ nanorods on an ITO-coated glass substrate via the chemical bath deposition (CBD) method. The top view of the SEM micrograph of the Ga_2_O_3_ nanorods was shown in Figure 34.

The α-Ga_2_O_3_/ITO sensing structure based on EGFET had been used to detect pH response behavior in the different (pH = 1, 3, 5, 7, 9, and 11) buffer solutions. Figure 35a shows the current–voltage (I-V) characteristic curves of the α-Ga_2_O_3_ nanorod pH-EGFET. The pH-EGFET based on α-Ga_2_O_3_ nanorod had superior sensitivity of 51.59–64.29 mV/pH. In addition, the linear response is shown in Figure 35b. The results found that α-Ga_2_O_3_ nanorod pH sensor possesses superior linear response with linearity close to 1. In addition, the long-term response of the α-Ga_2_O_3_ nanorod pH sensor was also monitored. At room temperature, a drift phenomenon was measured via an instrument amplifier circuit (AD620). The Output voltage of α-Ga_2_O_3_ pH sensor was monitored and recorded in pH 4, 7, and 10 buffer solutions over 6 h; the lower drift rate (~2.75 mV/h) in pH 7 buffer solution was observed.

In addition, the pH sensing properties of metal oxide materials (including α-Ga_2_O_3_ nanorods) have been discussed and proposed in ref. [252]. The linearity and pH sensitivity of the α-Ga_2_O_3_/ITO sensing structure was found to be superior, with a stable signal output and a fast response. The α-Ga_2_O_3_ nanorods are suitable for a wide range of pH detecting environments (pH 1~11). When compared to other materials, the Ga_2_O_3_ nanorod materials have a superior pH Nernst response. Chiang et al. provides a simple process of Ga_2_O_3_ nanomaterials for the low-temperature and low-cost fabrication of highly sensitive and disposable pH sensors.

**Figure 35 nanomaterials-12-03601-f035:**
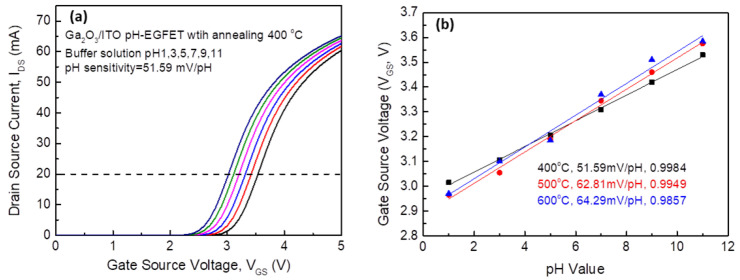
A family of current-voltage characteristic curves and pH response of α-Ga_2_O_3_ pH-EGFET, (**a**) annealed 400 °C, immersed in pH 1, 3, 5, 7, 9, 11 buffer solutions, respectively. (**b**) pH sensitivity and linearity of α-Ga_2_O_3_ pH-EGFETs were annealed at 400–600 °C. Reprinted figure with permission from [252]; copyright (2022) by Elsevier.

### 5.4. Photocatalytic and Photodegradation Applications

Recently, metal oxide materials, such as TiO_2_, ZnO, and Ga_2_O_3_ have been considered promising for photocatalytic and photodegradation applications [21,285,286,287,288] because the metal oxides possess excellent physical and chemical properties. The photocatalytic processes include two steps: one is hydroxyl radical oxidation (∙OH) and the other is photo-generated holes oxidation [289,290]. The generation of (∙OH) from water is controlled by the relative humidity of water molecules or gases on the photocatalyst surface. The oxidative properties of photo generated holes were greatly influenced by the structural and electronic properties of photocatalysts. Ga_2_O_3_ has a wide band gap (4.2–4.9 eV), and it has many superior properties that are important for semiconductor devices and industrial applications.

In addition, according to the above-mentioned (Section 5.1), the Ga_2_O_3_ materials synthesized solution process possesses a superior response to UV and DUV. In terms of photocatalytic and photodegradation applications, also the Ga_2_O_3_ materials have been investigated and published, as described below.

L. S. Reddy et al. [21] synthesized the β-Ga_2_O_3_ nanorods by a simple hydrothermal synthesis. The photocatalytic activity of samples explained in Figure 36a shows the photodegradation absorbance spectra of Rhodamine B (RhB) solution under UV light irradiation.

The maximum absorbance of RhB under UV light at different absorption times occurred at 553 nm for α-Ga_2_O_3_ and β-Ga_2_O_3_. The photographic of the RhB solution for various UV irradiation times (i.e., 0–180 min) is also shown in the inset of Figure 36b. Under the UV irradiation time of 180 min, photodegradation efficiency reached 62% for α-Ga_2_O_3_ and 79% for β-Ga_2_O_3_, as shown in Figure 36b.

W. Zhao et al. [212] synthesized mesoporous β-Ga_2_O_3_ nanorods by the hydrothermal method using different M.W PEG as a template. These β-Ga_2_O_3_ mesoporous nanorods were used as a catalyst for photodegradation of toluene. It was concluded from their studies that mesoporous β-Ga_2_O_3_ nanorods synthesized with PEG (M.W 200, 20 mL) can obtain the largest BET surface area (29 m^2^g^−1^) and mean pore size (30 nm). The percentage of dangerous pollutant intermediates (i.e., acetaldehyde and formaldehyde) produced from the photocatalytic activity of β-Ga_2_O_3_ was less than that of conventional TiO_2_ as a catalyst.

M. Muruganadham et al. [206] reported the synthesis of α-Ga_2_O_3_ microspheres by using hydrothermal method. The experimental results of photocatalytic degradation showed that α-Ga_2_O_3_ microsphere is more efficient than conventional TiO_2_ for AO7 dye degradation and Cr (VI) reduction.

D. Li et al. [207] synthesized mesomorphs α-Ga_2_O_3_ hierarchical structures by a using liquid assisted hydrothermal method. The synthesized α-Ga_2_O_3_ subjected to photodegradation of RhB dye under UV irradiation. The absorption spectra of RhB solution under different irradiation times showed that the maximum peak occurred at 556 nm. Under exposure via UV light irradiation for 100min, the as-prepared α-Ga_2_O_3_ removed 95% of the dye, while the commercial α-Ga_2_O_3_ only removed 60%. The greater absorption ability mostly attributed to the larger surface area of BET (35.94 m^2^g^−1^).

Y. Wang et al. [213] synthesized β-Ga_2_O_3_ nanorods by the hydrothermal method. The photocatalytic activity of β-Ga_2_O_3_ nanorods was studied by photocatalytic degradation of dye under solar irradiation. According to experimental results, with simulated solar irradiation after 4 h, the degradation rate of methylene blue (MB) and methyl orange (MO) reaches 95.32% and 90.47%, respectively.

J. Liu et al. [202] synthesized mesoporous mixed phase Ga_2_O_3_ by calcination of α-GaOOH at 600–700 °C. The absorption spectrum of photocatalytic degradation of Ga_2_O_3_-700 °C sample in metronidazole solution has a characteristic peak at 320 nm, and the peak decreases with the increase of irradiation time, indicating that the degradation of metronidazole solution occurred. Under UV irradiation for 120 min, 95.55% of the solution was removed with mesoporous mixed phase Ga_2_O_3_ photocatalyst

L. Cui et al. [211] synthesized ultrafine γ-Ga_2_O_3_ nanocrystals by the microwave hydrothermal method. Under UV light illumination, with RhB the calcined nanocrystals of γ-Ga_2_O_3_ exhibited a higher photodegradation efficiency of 90.7% compared to that of 68.2% for the as-synthesized pure γ-Ga_2_O_3_ nanocrystals.

H. Ryou et al. [228] synthesized Sn-doped β-Ga_2_O_3_ nanostructures via the hydrothermal process. The intrinsic β-Ga_2_O_3_ nanostructure had 40% photocatalytic activity and increased to 90% with 0.7 at.% Sn, decreased with an increase in Sn concentration, and flattened out from 3.2 to 7.3 at.% Sn, which was attributed to absorption capacity limitation due to a reduction in quantum yield because of absorption or scattering of UVC light by precipitated Sn oxides. The presence of SnO or SnO_2_ degraded the photocatalytic performance of β-Ga_2_O_3_ nanostructure.

S. Kim et al. [231] synthesized Al-doped β-Ga_2_O_3_ nanostructures by the hydrothermal method. These Al doped β-Ga_2_O_3_ nanostructures were used as a photocatalyst for degradation of the MB solution. The redox potential and surface area of β-Ga_2_O_3_ nanostructures increased with the increase in Al content but the defects increased. The defects in photocatalyst can either enhance or degrade the activity by acting as carrier separation and recombination sites [291]. With the increase in the Al content from (0.6–3.2 at.%) in β-Ga_2_O_3_ nanostructures the photocatalytic activity decreased. This decrease in photocatalytic activity was attributed to the generation of deep trap levels due to GB/crystallography [77,292] and oxygen vacancy defects [293].

In addition, the Ga_2_O_3_ materials were synthesized on glass substrates using the chemical bath method with a post-annealing process by C. Y. Yeh et al. [250]. A 0.075 M concentration of Ga(NO_3_)_3_ was used to prepare GaOOH materials. Afterwards, α-Ga_2_O_3_ materials were obtained by annealing, and the MB in the solution was photodegraded by more than 90% after 5 h of UV irradiation.

W. Zhang et al. [256] incorporated the solvent-synthesized γ-Ga_2_O_3_ nanoparticles into liquid metal/metal oxide (LM/MO) frameworks. The quantity of LM/MO frameworks (5.45 mg) and Ga_2_O_3_ nanoparticles (3.3, 16.3, and 32.6 μL) were regulated to yield 0.2, 1, and 2 wt.%, respectively. The photocatalytic performance of LM/MO frameworks incorporating γ-Ga_2_O_3_ nanoparticles was evaluated using Congo red (CR) as the indicator dye, in which the LM/MO framework incorporating 1 wt.% Ga_2_O_3_ nanoparticles was demonstrated to exhibit the best photocatalytic efficiency of ~100% h^−1^ within 1 h with high reusability.

K. Girija et al. [260] synthesized self-assembled novel floral β-Ga_2_O_3_ nanorods via the reflux condensation method. The floral β-Ga_2_O_3_ nanorods possess the BET-specific surface area of 40.8 m^2^g^−1^. In comparison with commercial Degussa TiO_2_ (P25), the photodegradation of RhB by the synthesized β-Ga_2_O_3_ had a high efficiency of 90%. In their other study [261], they prepared single crystalline β-Ga_2_O_3_ nanorods by a facile reflux condensation method using Ga(NO_3_)_3_.nH_2_O and anionic surfactant CTAB as the precursor materials. The as prepared β-Ga_2_O_3_ nanorods possess a large surface area of 49.5 m^2^g^−1^ and a band gap of 4.50 eV; β-Ga_2_O_3_ nanorods showed a good photo degradation efficiency of 92% and RhB degradation was achieved.

According to the above-mentioned, it can be found that the Ga_2_O_3_ material has excellent photodegradation characteristics, and the parameters affecting photodegradation characteristics include process parameters, annealing temperature, types of added materials, formulation ratios, and the micro-nano structure and crystal phase of the gallium oxide material.

### 5.5. Other Devices

In addition to the above-mentioned applications, some reports also propose other applications of gallium oxide materials via solution process method. Here, it is described as follows:

M. Tadatsugu et al. [12] prepared the first Ga_2_O_3_ based thin film via the sol-gel method. The Ga_2_O_3_: Mn thin film emitting layer (TFEL) was used to make a TFEL device combined with BaTiO_3_ substrate in which they concluded that the film annealed at 900 °C and had an optimal performance. The TFEL device exhibited luminance of 1271 and 401 cd/m^2^ when driven by sinusoidal wave voltages of 1 kHz and 60 Hz, respectively.

H. J. Bae et al. [215] synthesized high aspect ratio β-Ga_2_O_3_ nanorods by the hydrothermal method. Hydrothermally prepared β-Ga_2_O_3_ nanorods were used to fabricate FET by mechanically transferring nanorods onto back gated SiO_2_/P^+^ Si substrate. It revealed the charge transfer properties of an n-type semiconductor. This unintentional n-type behavior in un-doped β-Ga_2_O_3_ was also reported by Kohei Sasaki et al. [294].

A. Kaya et al. [14] synthesized the β-Ga_2_O_3_ thin films on p-Si substrate using the sol-gel method. The electrical properties, such as break-down voltage, interface trap density, series resistance, and dielectric properties, were investigated by fabricating a MOS capacitor using sol-gel prepared β-Ga_2_O_3_ as oxide, p-Si as a semiconductor, and Ni/Au as a metal contact.

S. Gao et al. [15] synthesized β-Ga_2_O_3_ thin films coated on MOCVD grown GaN substrate by using the sol-gel method. They prepared a Ni/sol-gel β-Ga_2_O_3_/GaN MOS structure and found that its electrical parameters, such as *I_o_*, n, and *Φ_ap_* were strongly affected by temperature due the lateral inhomogeneity of barrier height.

### 5.6. Limitations and Challenges

There has been significant research on Ga_2_O_3_ material via solution-based synthesis methods and their applications to the present day. Even though growth processes in solution-based synthesis cannot be precisely controlled, they are considerably cheaper than vacuum-based synthesis methods. When compared to vacuum-based methods such as CVD and PVD, thin films prepared by solution methods have several defects and nanoparticles that were relatively large in size. The surface of thin films prepared by CVD and PVD is much smoother than that of solution methods. The surface area to volume ratio of nanomaterials synthesized using solution-based methods was high, which is important in sensor [187,216,217,220] and photocatalyst [212,231,260] applications. Large-scale synthesis and cost-effective approaches are critical for photoelectrodes and photocatalysis [295]. Process simplification and defect minimization, as well as growth and size control of solution processes can promote large-scale production and in-turn facilitates practical demonstration.

## 6. Conclusions

In this review, we have summarized the characteristics comparison and application fields of gallium oxide materials prepared by the hydrothermal method, sol-gel method, chemical bath method, and so on. Gallium oxide materials have shown a potential development in emerging technologies based on varied nanostructures via solution processes. These structures are influenced by various techniques, materials, and process parameters employed in those corresponding synthesis methods. These used parameters also have a major role, as with change in parameters a change in formation was observed. In addition, gallium oxide materials can be effectively used as DUV detectors, gas sensors, pH sensors, photocatalytic and photodegradation applications, and so on. Hence, we hope that this review can provide a reference for researchers when preparing gallium oxide materials using solutions processes.

## Figures and Tables

**Figure 1 nanomaterials-12-03601-f001:**
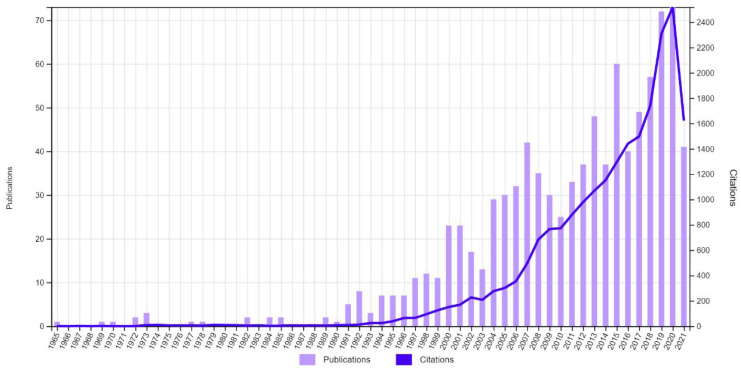
Number of publications and citations on gallium oxide from solution process since 1965. A total of 936 papers have been published and the number of citations is 15821 in the last 55 years (Data: Web of Science).

**Figure 2 nanomaterials-12-03601-f002:**
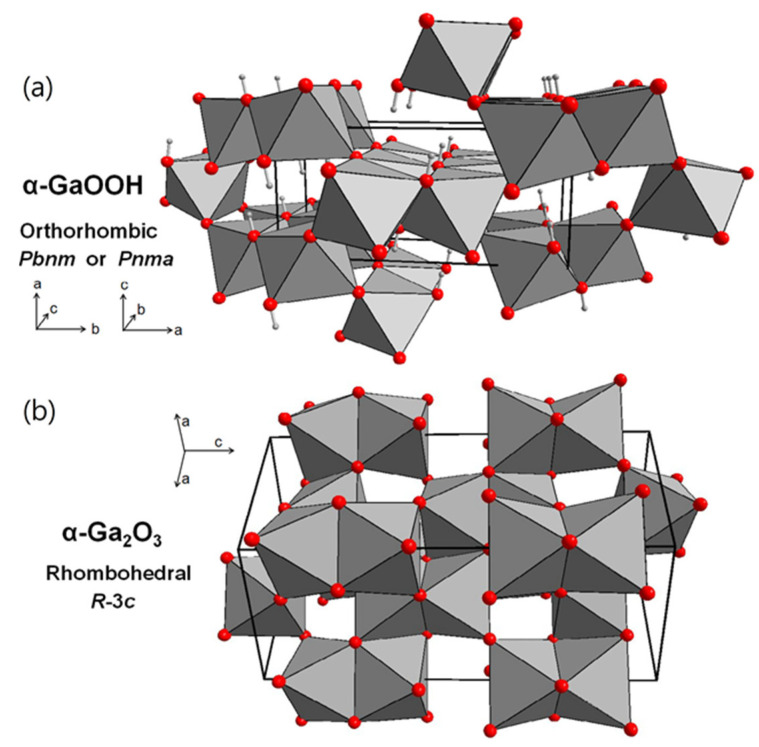
Crystal structures of (**a**) α-GaOOH and (**b**) α-Ga_2_O_3_ phases represented by the cation polyhedra. The space group, crystallographic axes, and unit cell are designated. Reprinted figure with permission from [43]; copyright (2015) by Elsevier.

**Figure 3 nanomaterials-12-03601-f003:**
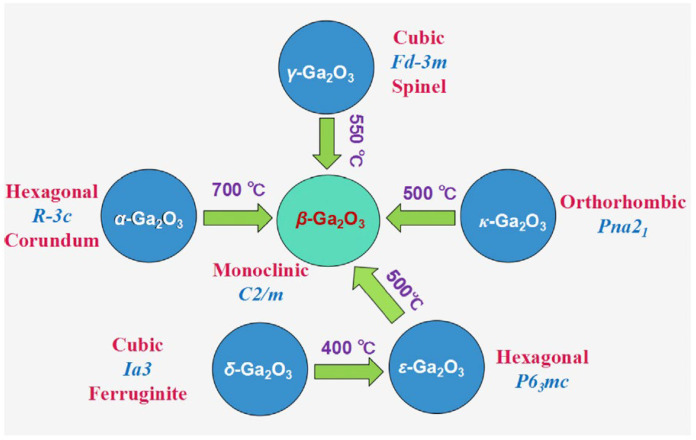
Interconversion relation of Ga_2_O_3_ polymorphs. Reprinted figure with permission from [48]; copyright (2019) by Elsevier.

**Figure 4 nanomaterials-12-03601-f004:**
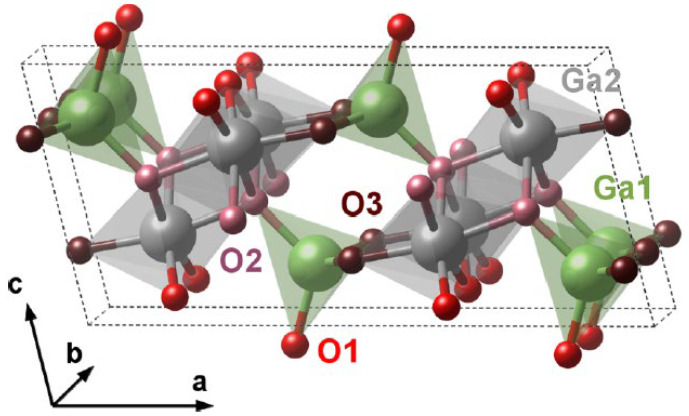
Unit cell of β-Ga_2_O_3_. Tetrahedrally coordinated Ga1 are green while octahedrally coordinated Ga2 are gray. Threefold O1 and O2 are indicated in red and pink, respectively, while fourfold O3 are indicated in maroon. Reprinted figure with permission from [49]; copyright (2016) by the American Physical Society.

**Figure 5 nanomaterials-12-03601-f005:**
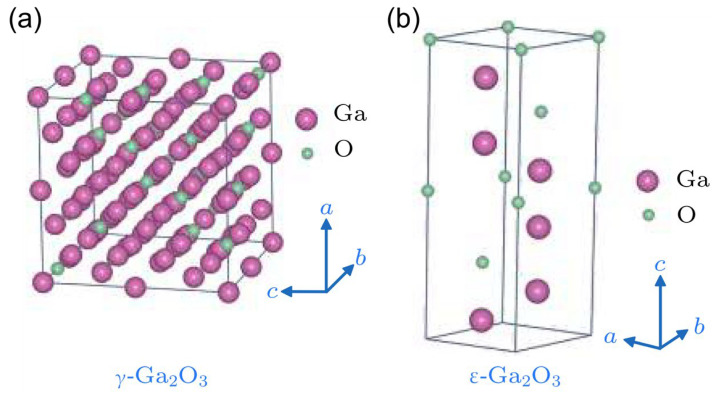
Crystal structures of polymorphs of (**a**) γ-Ga_2_O_3_ and (**b**) ε-Ga_2_O_3_. Reprinted figure with permission from [48]; copyright (2019) by Elsevier.

**Figure 7 nanomaterials-12-03601-f007:**
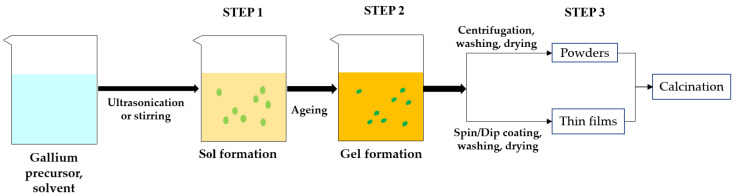
A schematic of sol-gel preparation of gallium oxide illustrating all steps involved.

**Figure 8 nanomaterials-12-03601-f008:**
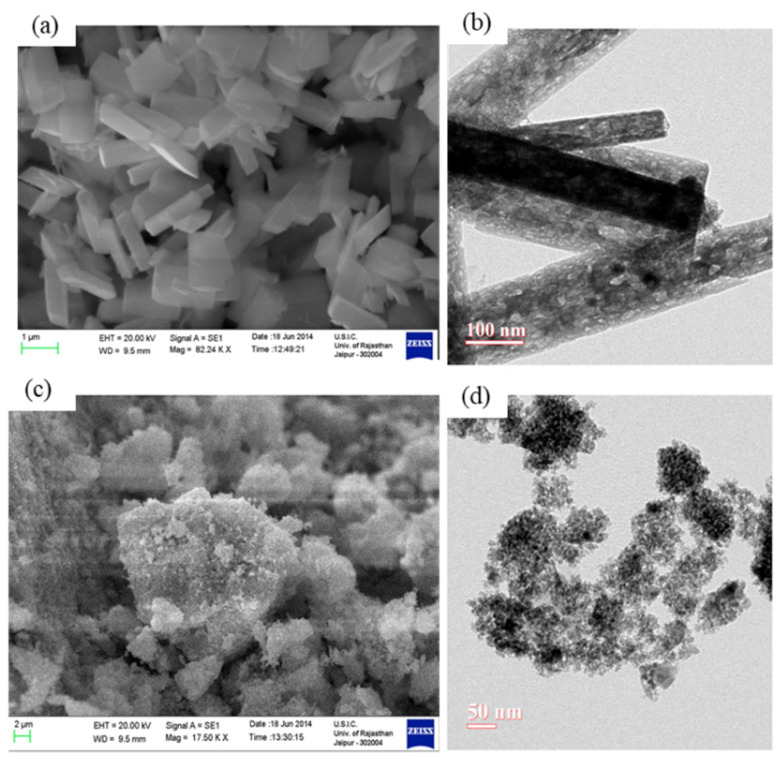
(**a**,**c**) SEM and (**b**,**d**) TEM images of β-Ga_2_O_3_ (**a**,**b**) and γ-Ga_2_O_3_ (**c**,**d**). Reprinted figure with permission from [183]; copyright (2018) by Elsevier.

**Figure 9 nanomaterials-12-03601-f009:**
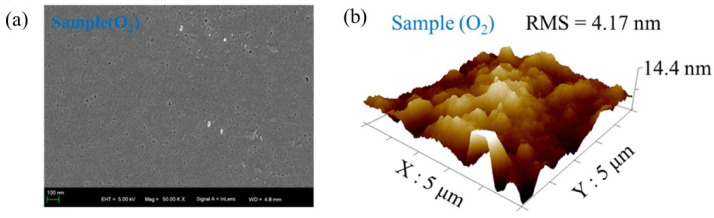
(**a**) SEM and (**b**) AFM images of high quality α/β polycrystalline Ga_2_O_3_ thin film annealed at 800 °C in O_2_ atmosphere. Reprinted figure with permission from [189]; copyright (2020) by Elsevier.

**Figure 10 nanomaterials-12-03601-f010:**
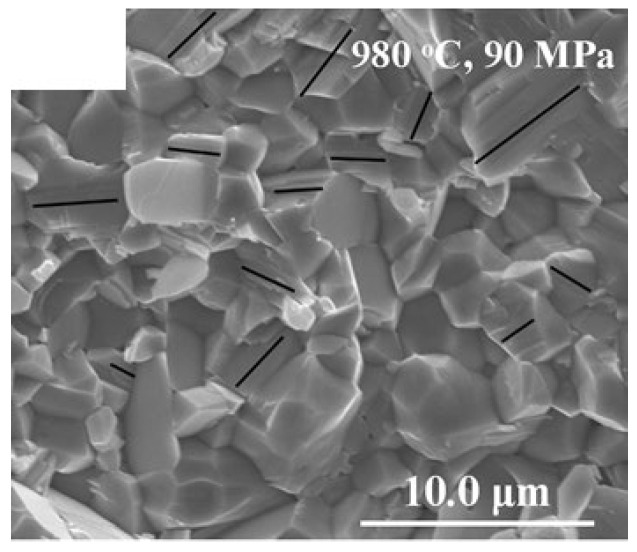
SEM image of β-Ga_2_O_3_ sample sintering obtained via SPS at 980 °C ^@^ 90 Mpa. Reprinted figure with permission from [190]; copyright (2020) by Elsevier.

**Figure 12 nanomaterials-12-03601-f012:**
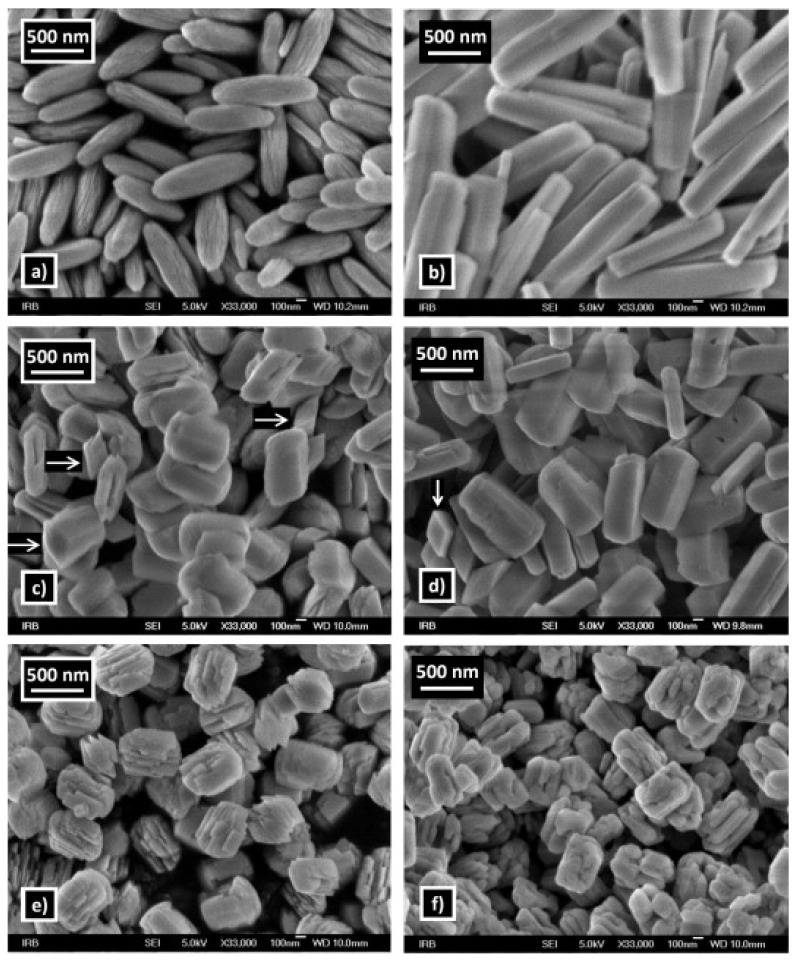
Morphology of α-GaOOH samples (**a**) A-60, (**b**) A-160, (**c**) N-60, (**d**) N-160, (**e**) B-60, and (**f**) B-160. Presence of rhombic faces shown with arrows. Reprinted figure with permission from [43]; copyright (2015) by Elsevier.

**Figure 13 nanomaterials-12-03601-f013:**
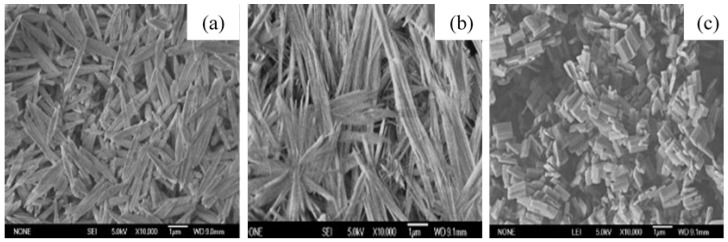
The morphologies of prepared GaOOH (**a**) without surfactant and with (**b**) SDBS and (**c**) SA. Reprinted figure with permission from [204]; copyright (2010) by Elsevier.

**Figure 14 nanomaterials-12-03601-f014:**
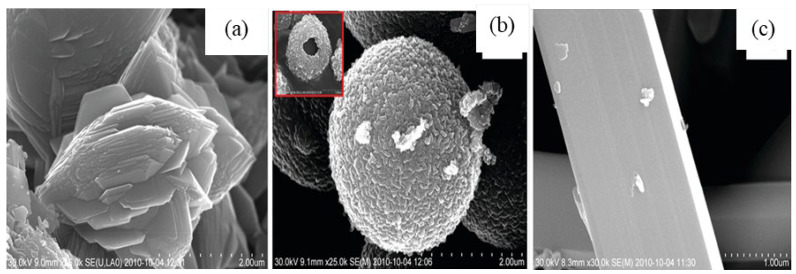
FESEM images of GaOOH morphologies prepared at various hydrothermal temperatures (**a**) 175 °C (**b**) 200 °C and (**c**) 225 °C. Reprinted figure with permission from [206]; copyright (2012) by ACS Publications.

**Figure 15 nanomaterials-12-03601-f015:**
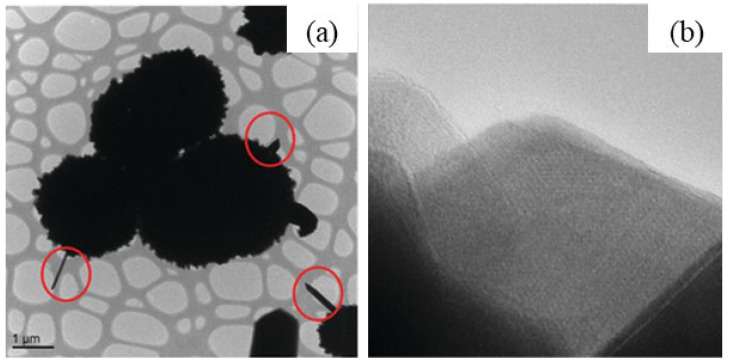
HR-TEM images of α-Ga_2_O_3_ (**a**) microspheres (Insert red circle shows shard-end rods with oriented attachment grown on microsphere surface) (**b**) nanoparticles assembled on the microsphere surface. Reprinted figure with permission from [206]; copyright (2012) by ACS Publications.

**Figure 16 nanomaterials-12-03601-f016:**
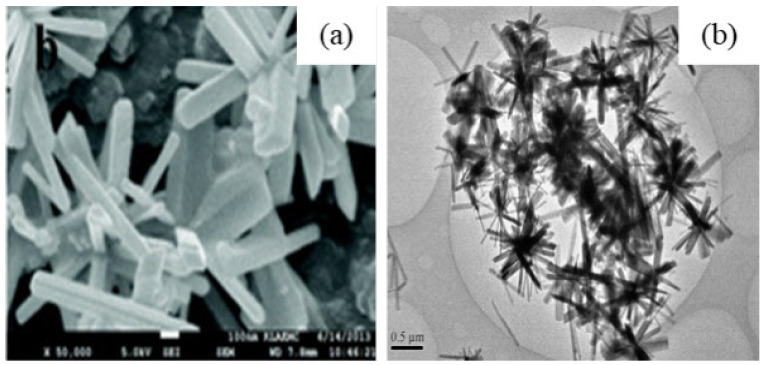
α-Ga_2_O_3_ hierarchical structure by (**a**) SEM and (**b**) TEM. Reprinted figure with permission from [207]; copyright (2013) by the Royal Society of Chemistry.

**Figure 17 nanomaterials-12-03601-f017:**
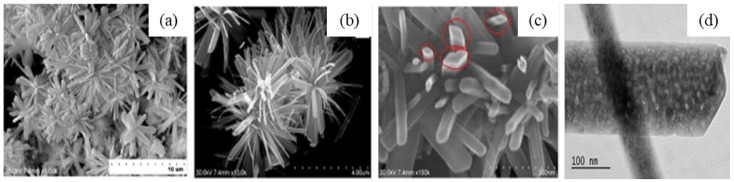
SEM images of (**a**) micro flowers of α-Ga_2_O_3_ prepared with oxalic acid in solution (**b**) nanoribbons (**c**) nanorods (**d**) TEM image of α-Ga_2_O_3_ nanorod/ribbon. Reprinted figure with permission from [208]; copyright (2015) by Elsevier.

**Figure 18 nanomaterials-12-03601-f018:**
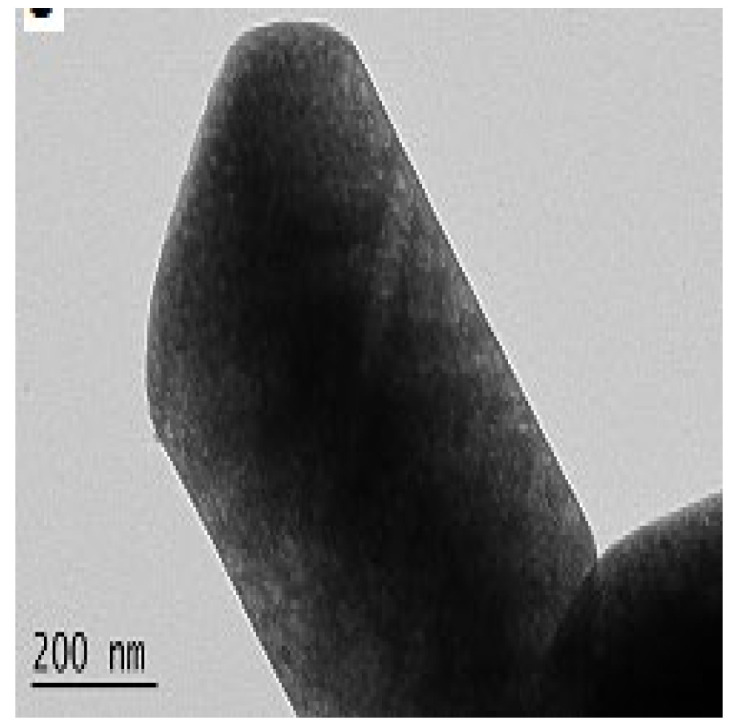
TEM image of α-Ga_2_O_3_ micro rods prepared with biuret in solution. Reprinted figure with permission from [208]; copyright (2015) by Elsevier.

**Figure 19 nanomaterials-12-03601-f019:**
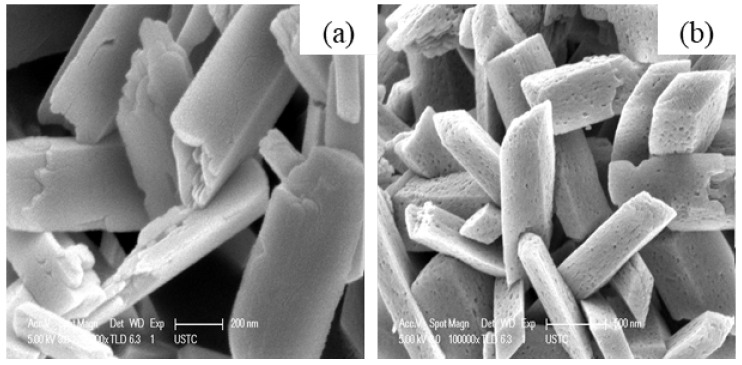
The morphologies of β-Ga_2_O_3_ prepared (**a**) without and (**b**) with 20 mL PEG200. Reprinted figure with permission from [212], Copyright (2011) by Elsevier.

**Figure 20 nanomaterials-12-03601-f020:**
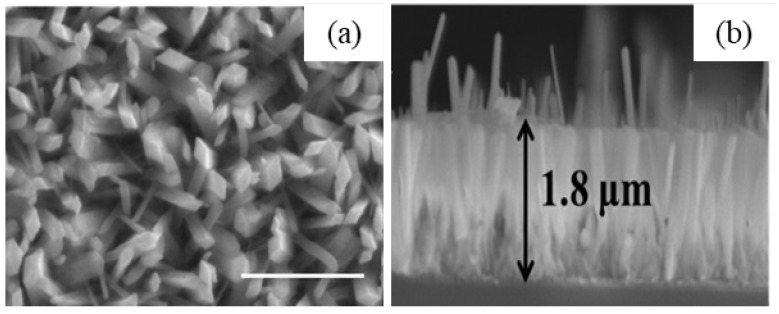
SEM images of (**a**) GaOOH NRA’s on SnO_2_/SiO_2_/Si substrate (**b**) side view. Reprinted figure with permission from [219], Copyright (2016) by ACS Publications.

**Figure 21 nanomaterials-12-03601-f021:**
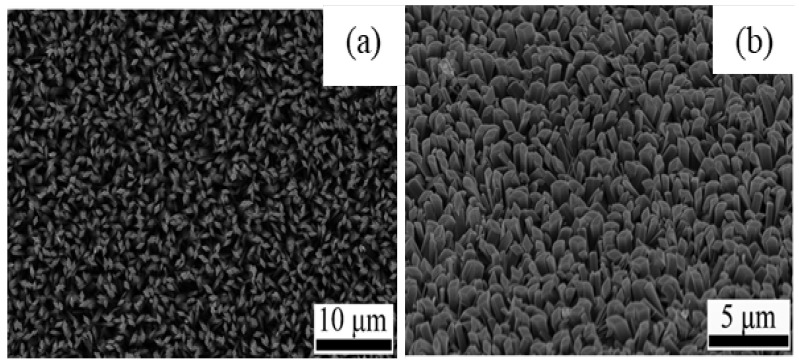
SEM images of (**a**) β-Ga_2_O_3_ microrod arrays on Si substrate (**b**) 45° tilt view. Reprinted figure with permission from [221]; copyright (2018) by the Royal Society of Chemistry.

**Figure 22 nanomaterials-12-03601-f022:**
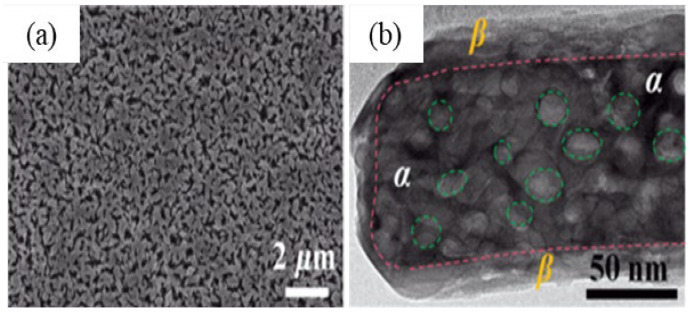
(**a**) FE-SEM and (**b**) TEM images of a α/β-Ga_2_O_3_ nanorod. Reprinted figure with permission from [223]; copyright (2020) by APS Physics.

**Figure 23 nanomaterials-12-03601-f023:**
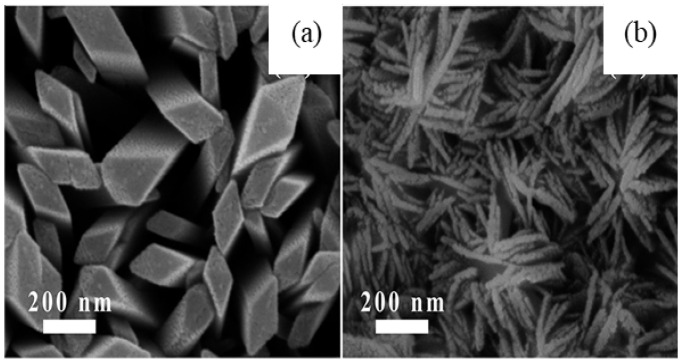
SEM images of pure α-Ga_2_O_3_ nanorods (**a**) and tree liked branched structure Ga_2_O_3_ nanorods (**b**) after treatment with 0.015 M aluminum nitrate. Reprinted figure with permission from [227]; copyright (2021) by Elsevier.

**Figure 24 nanomaterials-12-03601-f024:**
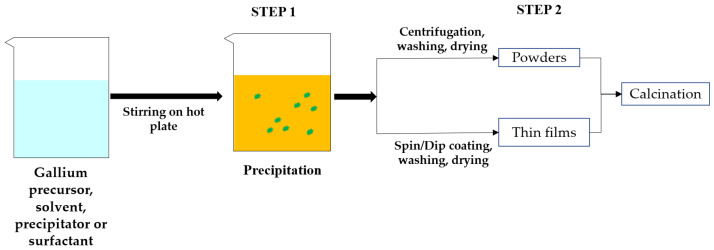
The schematic of chemical bath preparation of gallium oxide illustrating all steps involved.

**Figure 25 nanomaterials-12-03601-f025:**
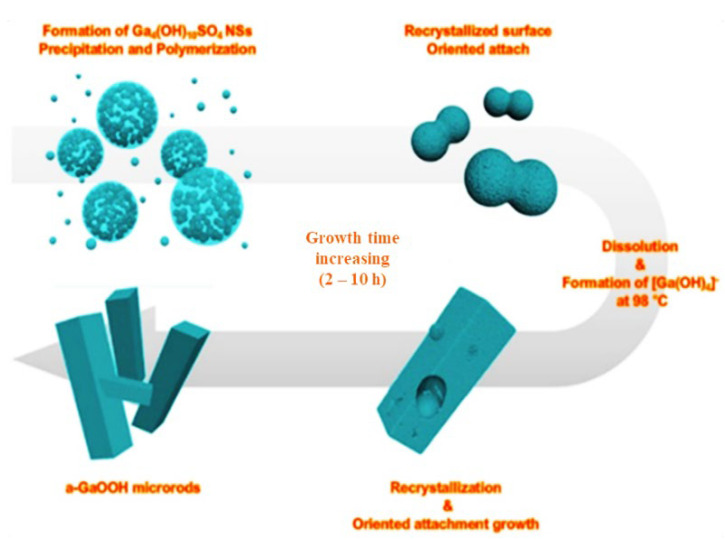
Illustration of time dependent growth of gallium precursors. Reprinted figure with permission from [258]; copyright (2016) by Elsevier.

**Figure 26 nanomaterials-12-03601-f026:**
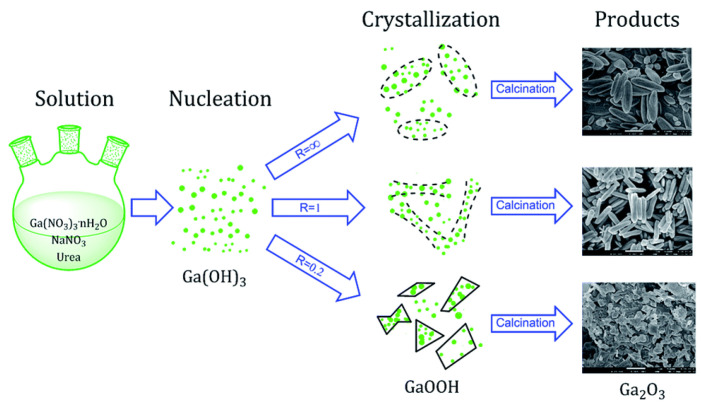
Schematic of morphology-controlled synthesis of Ga_2_O_3_. Reprinted figure with permission from [259]; copyright (2017) by the Royal Society of Chemistry.

**Figure 27 nanomaterials-12-03601-f027:**
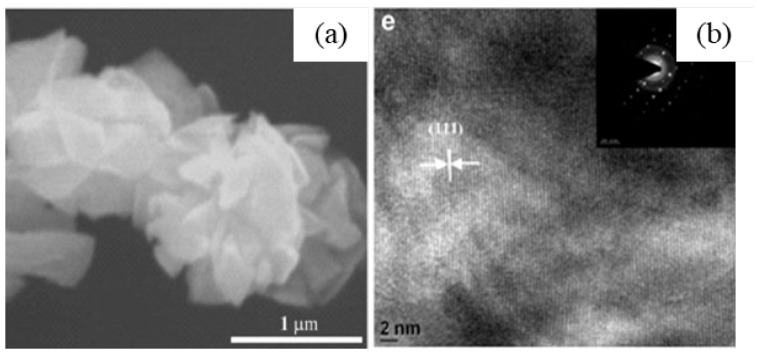
(**a**) SEM image of Floral β-Ga_2_O_3_ (**b**) TEM image of nanorod inside the floral structure of β-Ga_2_O_3_. Reprinted figure with permission from [260], copyright (2013) by Elsevier.

**Figure 28 nanomaterials-12-03601-f028:**
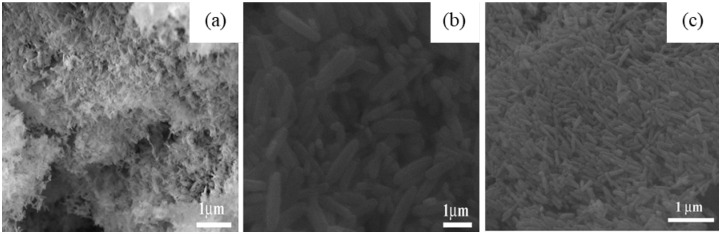
Time dependent morphology evolution of β-Ga_2_O_3_. (**a**) agglomerated rod-like structure (**b**) Randomly aligned non-uniform nanorods (**c**) Side-by-side aligned uniform nanorods. Reprinted figure with permission from [261]; copyright (2014) by Elsevier.

**Figure 29 nanomaterials-12-03601-f029:**
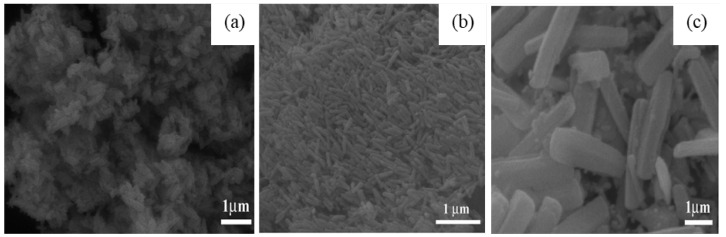
Concentration dependent morphology evolution of β-Ga_2_O_3_. (**a**) agglomerated nanoparticles (**b**) Side-by-side aligned uniform nanorods (**c**) Randomly aligned non-uniform nanorods. Reprinted figure with permission from [261]; copyright (2014) by Elsevier.

**Figure 30 nanomaterials-12-03601-f030:**
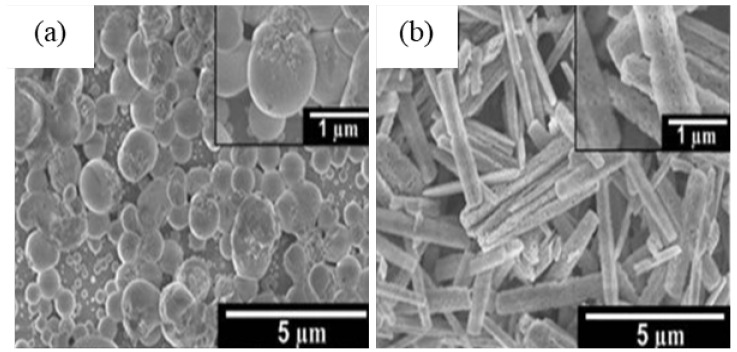
(**a**) Nanodot structure of Ga_2_O_3_ prepared at molarity of 0.1 M (**b**) Nanorod structure of Ga_2_O_3_ prepared molarity of 1.0 M in acidic medium (i.e., at pH 4). Reprinted figure with permission from [263]; copyright (2014) by Springer.

**Figure 31 nanomaterials-12-03601-f031:**
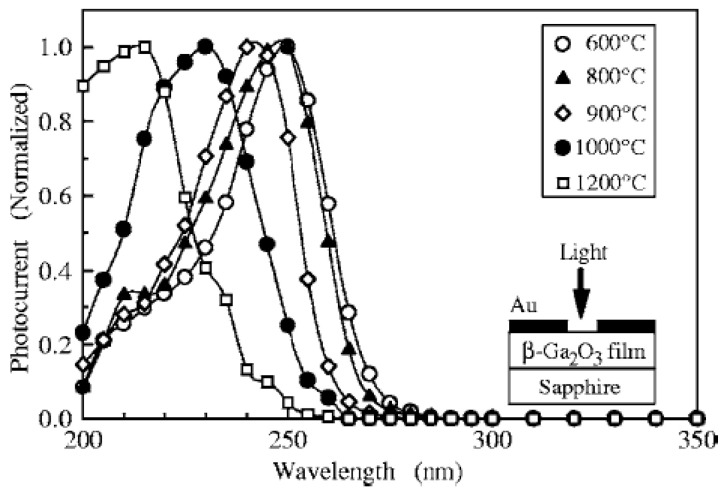
Photocurrent characteristics responses of β-Ga_2_O_3_ photodetectors prepared at 600–900 °C temperatures. Reprinted figure with permission from [181]; copyright (2007) by the American Institute of Physics.

**Figure 32 nanomaterials-12-03601-f032:**
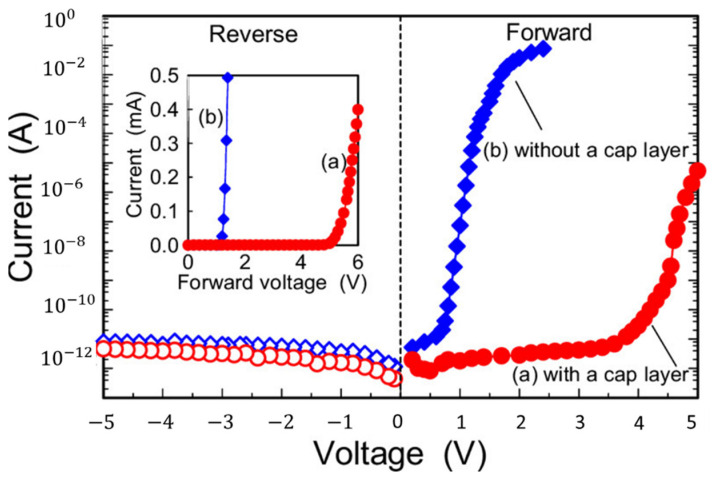
(Color online) Spectral response of Ga_2_O_3_ photodiodes with and without a cap layer at reverse and forward biases of 3 V. The inset shows the incident light intensity dependence of the photocurrent at (**a**) forward and (**b**) reverse biases of 3 V under illumination with 250 nm light. Reprinted figure with permission from [182]; copyright (2011) by the American Institute of Physics.

**Figure 33 nanomaterials-12-03601-f033:**
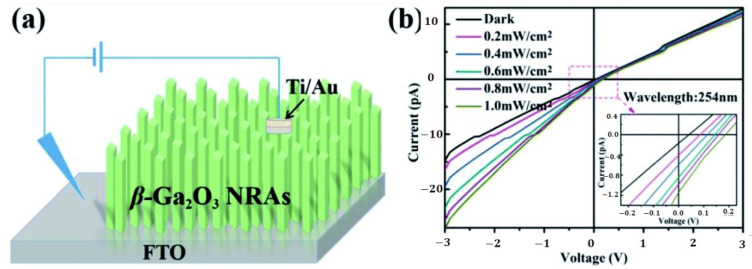
(**a**) The β-Ga_2_O_3_ NRAs solar-blind photodetector. (**b**) Current–voltage curves of the device in the dark and under 254 nm illumination with various light power densities. Reprinted figure with permission from [226]; copyright (2019) by the Royal Society of Chemistry.

**Figure 34 nanomaterials-12-03601-f034:**
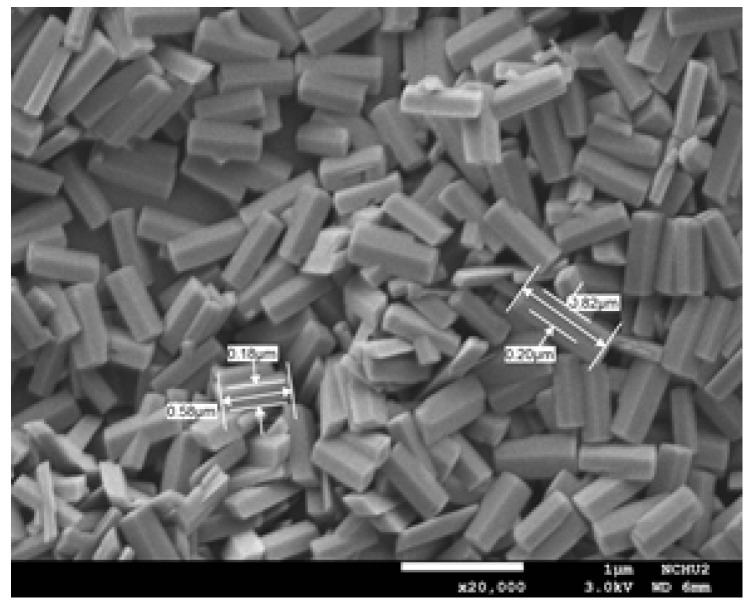
The as-deposited surface morphology of GaOOH nanorods, with the same molar concentration ratios (Ga(NO_3_)_3_/HMT = 1).

**Figure 36 nanomaterials-12-03601-f036:**
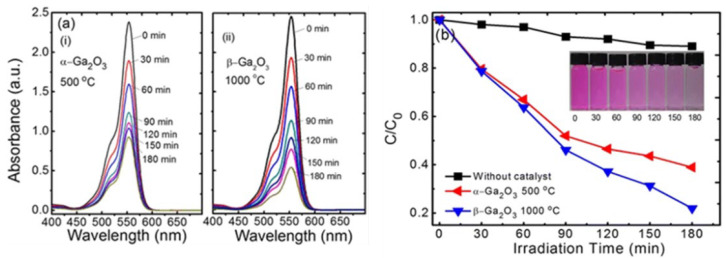
Photocatalytic activity of α-Ga_2_O_3_ and β-Ga_2_O_3_ under different UV irradiation. (**a**) (i) absorbance spectra of α-Ga_2_O_3_ and (ii) absorbance spectra of β-Ga_2_O_3_ nanorods. (**b**) Calculated constant reaction rate of the photodegradation of RhB solution (C/C_0_) as a function of UV irradiation time for α-Ga_2_O_3_ and β-Ga_2_O_3_ nanorods. Reprinted figure with permission from [21]; copyright (2015) by Springer.

**Table 1 nanomaterials-12-03601-t001:** Crystal structures and lattice parameters of different Ga_2_O_3_ polymorphs.

Polymorph Type	Crystal Structure	Space Group	Lattice Parameters	Ref.
α	Hexagonal or rhombohedral	R-3c	a = b = 4.98 Å, c = 13.43 Å, α = β = 90°, γ = 120°	[25]
β	Monoclinic	C2/m	a = 12.23 Å, b = 3.04 Å, c = 5.80 Å, α = γ = 90°, β = 103.7°	[35]
γ	Cubic	Fd-3m	a = b = c = 8.24 Å, α = β = γ = 90°	[36]
δ	Cubic	Ia3	a = b = c = 9.402 Å, α = β = γ = 90°	[37]
ε	Hexagonal	P6_3_mc	a = b = 2.90 Å, c = 9.26 Å, α = β = 90°, γ = 120°	[32]

**Table 2 nanomaterials-12-03601-t002:** Basic physical properties of β-Ga_2_O_3_.

Property	Value	Ref.
Band gap (eV)	4.8~4.9	[82]
breakdown electric field (MV/cm)	~8	[47]
Saturation velocity (10^7^ cm/s)	1.8~2.0	[7]
Melting point (°C)	1793	[48]
Specific heat (J·g^−1^·K^−1^)	0.56	[92]
Thermal conductivity (W·m^−1^·K^−1^)	10.9 ± 1.0 along [100] 27.0 ± 2.0 along [010] 15.0 along [001]	[103]
Thermal diffusivity (mm^2^·s^−1^)	3.7 ± 0.4 along [100] 9.6 ± 0.5 along [010] 7.1 ± 0.4 along [001]	[114]
Thermal expansion (K^−1^)	1.54 × 10−6 along [100] 3.37 × 10−6 along [010] 3.15 × 10−6 along [001] 2.23 × 10−5 for β angle	[110]
Cleavage plane	(100), (001)	[115]
Absorption edge (nm)	270~275	[86,87,88,89]
Emission bands at RT (eV)	UV (3.2–3.6) blue (2.8–3.0) green (2.4)	[18]
Refractive index (in the visible spectrum)	1.98~2.1	[6]
High-frequency dielectric constant	3.57	[116]
Low-frequency (or static) dielectric constant	10.2 ± 0.3	[117]

**Table 3 nanomaterials-12-03601-t003:** Sol-gel synthesis of Ga_2_O_3_.

Method	Substrate/ Template	Precursor	Synthesis Conditions	Properties	Application	Ref.
Sol-gel dip coating	BaTiO_3_ (Thickness-0.2 mm)	Gallium acethylacetonate (Ga(C_5_H_7_O_2_)_3_)—A) + manganese chloride (MnCl_2_) + methanol (CH_3_OH)—(B). Molar ratio = ($\frac{A}{A+B})$ = 0.08, Mn content: 0.3 at.%	Room Temperature (RT) stirring for 30 min, then added HCl and H_2_O. pH of solution: 3.4, Stirring at 50 °C for 5 h under N_2_ ambient, Dip coated films dried at RT for 5 min, Pre-heat: 10 min, 600–100 °C. Process repetition: 25 times, Calcination: 850–1070 °C for 1 h in Ar ambient	Amorphous thin film of Ga_2_O_3_: Mn. Thickness: 2 µm	Green-Light emitting TFEL device	[12]
Sol-gel spin coating	Sapphire transducers	Gallium isopropoxide (0.15 M) + i. Ce doping—cerium isopropoxide, or ii. W doping—tungsten ethoxide, or iii. Sb doping—antimony butoxide, or iv. Zn doping—zinc acetyleacetonate hydrate. Each element doping level—3 mol%	Solutions prepared in dry N_2_ ambient, Ultra sonication of mixed solution: 1 h, Aging time: 24 h, Spin coating at 3000 rpm for 30 s, dried at RT for 24 h, Calcination: 600 °C for 1 h	Ga_2_O_3_ thin film doped with Ce, Sb, W and Zn. Thickness: 200 nm	O_2_ gas sensor	[179]
Sol-gel		Gallium(III)-isopropoxide (2.714 g) + 2-Propanol—200 mL + Hot water—200 mL (90—100 °C) + 25% *w*/*w* aqueous TMAH solution (2 mL)	Ultra-speed centrifuge: 20,000 rpm, washed with ethanol and water then dried, Calcination: 500 °C for 4 h, 900 °C for 2 h	Amorphous α and β-Ga_2_O_3_ at 500 °C, β-Ga_2_O_3_ at 900 °C (sub-micrometer size)		[180]
		Gallium(III)-isopropoxide (1.796 g) + Twice distilled water—50 mL (20 °C)	Ultrasonication for 1 h, liquid phase evaporation at 70 °C for 24 h. Calcination: 500 °C for 4 h, 900 °C for 2 h	(10–20 nm) sized particles of β-Ga_2_O_3_ at 500 °C, sub-micromere size particles of β-Ga_2_O_3_ at 900 °C		
		GaCl_3_ aqueous solution—25 mL (0.284 M) + Twice distilled water—175 mL (20 °C) + 25 % *w*/*w* TMAH aqueous solution	pH: 7.82, Ageing time: 2 h, Calcination: 500 °C for 4 h, 900 °C for 2 h	Elongated and uniform sub-micron sized particles of α-Ga_2_O_3_ at 500 °C, big agglomerates of β-Ga_2_O_3_ with less size than that of α-Ga_2_O_3_		
Sol-gel spin coating	Sapphire (0001)	(Gallium isopropoxide (A) + (2-Methoxyethanol + monoethanolamine (B)))—0.4 mol/L Molar ratio (C_m_ ratio): ($\frac{B}{A}$ = 1)	Solution stirred for 1 h at 60 °C till transparent sol appear, Spin coating at 3000 rpm then dried at 90 °C for 10 min, Pre heat at 300 °C for 20 min, 6 times process repetition Calcination: 1000 °C for 1 h	Thickness: 150–200 nm The band gap of β-Ga_2_O_3_ thin films increased due to Al doping into Ga_2_O_3_ at 900 °C	Solar-blind Photo-detector	[181]
Sol-gel spin coating	β-Ga_2_O_3_ (100) substrate(Thickness: 0.4 mm)	(Gallium isopropoxide (A) + (2-Methoxyethanol + monoethanolamine (B)))—0.4 mol/L Molar ratio C_m_ ratio: ($\frac{B}{A}$ = 1)	Solution stirred for 1 h at 60 °C till transparent Sol appear, Spin coating at 3000 rpm then dried at 90 °C for 10 min, Pre heat at 300 °C for 20 min, 6 times process repetition Calcination: 1000 °C for 1 h	β-Ga_2_O_3_ thin film grown epitaxially on β-Ga_2_O_3_ substrate, Thickness: 120 nm	Solar-blind Photo-diode	[182]
Sol-gel		(Gallium (III) isopropoxide (Ga (Opr^i^)_3_) (2 g) + anhydrous iso-propanol (20 mL)) + 2 drops of water-isopropanol	Pre-stirring for 2 h, aged for 2 days with continuous stirring, dried in oven at 100 °C then washed with acetone & water Calcination: 600 °C for 6 h	β-Ga_2_O_3_ mono-crystalline nanorods, Length = 100 nm		[183]
		N-phenylsalicylaldimine modified gallium (III) isopropoxide (2 g) + anhydrous iso-propanol (20 mL)) + 2 drops of water-isopropanol	Pre-stirring for 2 h, aged for 2 days with continuous stirring, dried in oven at 100 °C then washed with acetone & water Calcination: 600 °C for 6 h	γ-Ga_2_O_3_ polycrystalline nanoparticles, Size = 10 nm		
Sol-gel		Solution part 1: Gallium metal + HNO_3_—(Total pH: 1–2), Solution part 2: Tetraethyl orthosilicate + (ethanol-water mixture) + Few drops of 0.1 N HCL then stirred for 1 h	Molar ratio: Ga_2_O_3_ to SiO_2_—10:90, 20:80 and 30:70. Both solutions mixed (pH: 1–2) then stirred for 1 h, Heated at 70 °C for 3.5 h, died at 200 °C for 4 h. Calcination: 400 °C for 11 h, 500 °C for 5 h and 900 °C for 8 h	Ga_2_O_3_:SiO_2_ composite nanoparticles, β-Ga_2_O_3_ phase formation at a low temp of 400 °C for each molar ratio		[184]
Sol-gel dip coating	Amorphous quartz or Silicon	((Gallium metal + HCl) + dry ethanol)—0.075 mol/L + few drops of acetic acid	Solution stirred until it is appearing clear, Dip coated films dried at 100 °C for 5 min, Calcination: 700 °C for 1 h	β-Ga_2_O_3_ thin film		[185]
Sol-gel	Porous alumina Template (Pore size: 200 nm)	Gallium nitrate hydrate (0.4 M) + ethanol (5 M) + concentrated aqueous ammonia diluted in ethanol (50% vol.) (0.25 M) added drop wise	Precipitates separated centrifugally then washed with DI water then peptized in nitric acid to form stable sol, Template immersed in solution for 5 s at RT then dried in air for 30 min, Calcination: 500 °C for 12 h	Hollow nano-tubes of Ga_2_O_3_, Length 50 µm, Inside diameter: 100 nm and outside diameter: 200 nm		[186]
Sol-gel dip coating	Quartz and alumina substrate transducers	Solution part 1: Titanium tetraisopropoxide (A) + HCl (B) + H_2_O (C). C_m_ ratio: (A:B:C=0.4:0.2:48.8) Solution part 2: Gallium (III) nitrate hydrate + hydroxypropyl cellulose (1.5 g/100 mL) +DI water	Ti:Ga atomic ratios (at.%/at.%) = 100:00, 75:25, 50:50 and 25:75. Part 1 peptized at 70 °C for 2 h. Part 2 stirred for 30 min, Mixed sol stirred for 2 h at 70 °C, dip coated films dried at 150 °C for 1 h Calcination: 600, 800 and 1000 °C for 1 h	Ga_2_O_3_ retards the anatase to rutile formation of TiO_2,_ High SSA for Ti:Ga = 50:50 annealed at 600 °C	CO and NO_2_ gas sensor	[187]
Sol-gel drop casting	(100) p-Si wafer doped with boron (Thickness: 280 µm)	Gallium nitrate hydrate + ethanol	After drop casting on substrate, films spun at 3000 rpm for 30 s, Heated at 100 °C for 30 min to evaporate ethanol, Calcination: 800 °C for 2 h in Ar ambience	β-Ga_2_O_3_ thin film	MOS capacitor	[14]
Sol-gel drop casting	MOCVD grown GaN on sapphire	Gallium nitrate hydrate + ethanol	After drop casting on substrate, films spun at 3000 rpm for 30 s, Heated at 100 °C for 30 min to evaporate ethanol, Calcination: 800 °C for 2 h in Ar ambience	β-Ga_2_O_3_ thin film	MOS structure	[15]
Sol-gel spin coating	Sapphire (0001)	(Gallium nitrate hydrate (A) + (2-Methoxyethanol + monoethanolamine (B)))—0.5 mol/L Molar ratio C_m_ ratio: ($\frac{B}{A}\;$ = 1)	Solution stirred for 1 h at 60 °C till transparent sol appear, aged at RT for 36 h, after spin coating film kept on hot plate for 10 min, Pre heat at 500 °C for 15 min, Process repetition: 6 times, Calcination: 500 °C to 1100 °C for 2 h	Crystalline β-Ga_2_O_3_ thin film at 700 °C and higher, Thickness: 150–200 nm	Solar-blind ultra-violet photo-detectors	[188]
Sol-gel spin coating	Sapphire (0001)	(Gallium nitrate hydrate + ethanol)—0.6 mol/L	Solution stirred at 60 °C for 90 min and aged at RT for 36 h, spin coated substrates were heated on hot plate at 100 °C for 15 min, Pre heated at 300 °C for 25 min, Process repetition: 4 times, Calcination: 500 °C–900 °C for 2 h and later at 800 °C in O_2_, N_2_ and N_2_-O_2_ environments for 2 h	At 600 °C—low intensity β-Ga_2_O_3_. At 700–800 °C—α/β polycrystalline Ga_2_O_3_ thin film, At 900 °C—polycrystalline β-Ga_2_O_3_. Thickness: 240–280 nm	Photo-detector	[189]
Citrate sol-gel		Gallium nitrate + (citric acid (C_6_H_8_O_7_)—1.5 mole times than cations)	Stirred in water bath at 90 °C, transparent Sol dried at 200 °C, Calcination: 500 °C for 4 h in O_2_ ambience	Ga_2_O_3_ sub-micro powders		[190]
Sol-gel spin coating	Sapphire (0001)	(Gallium nitrate(A) + ethanol)—0.4 mol/L) + monoethanolamine (B), C_m_ ratio: ($\frac{B}{A}$ = 1)	Spin coated at 1000 rpm for 50 s, Pre-heating of films at 100–500 °C in O_2_ ambient for 10 min, Calcination: 1000 °C in O_2_ ambient for 2 h	β-Ga_2_O_3_ thin films		[191]

**Table 4 nanomaterials-12-03601-t004:** Hydrothermal synthesis of Ga_2_O_3_.

Method	Substrate/ Template	Precursor	Synthesis Conditions				Properties	Applications	Ref.
hydrothermal method		Gallium acetylacetonate (0.1 M) + DI water	NH_4_OH added to the transparent solution dropwise till pH-10, Solution stirred at 65 °C for 5 h continuously, Reaction at 140 °C for 10 h, dried at 70 °C for 6 h, Calcination: 600 °C, 800 °C, 850 °C, 950 °C and 1000 °C				Cuboid shape β-Ga_2_O_3_ starting from 800–850 °C, Pore size was maximum at 950 °C		[201]
		Gallium chloride (0.1 M) + DI water					Rice like morphology β-Ga_2_O_3_		
		Gallium nitrate (0.1 M) + DI water					Rice like morphology β-Ga_2_O_3_		
hydrothermal method		(Commercial Ga_2_O_3_ + HCl) + DI water-35 mL	NaOH solution added till pH = 6–8 Reaction in autoclave at 180 °C for 24 h, hydrothermal crystals filtered, washed and dried at 60 °C for 6 h, Calcination: 900 °C				-At pH 6, β-Ga_2_O_3_ regular quadrilateral nanorods (width: 200–300 nm) -At pH 8 spindle like β-Ga_2_O_3_ nanorod arrays formed		[45]
		(Commercial Ga_2_O_3_ + HCl) + DI water—35 mL + solvent of diethylene glycol (DEG) and water (1:1)							
hydrothermal method		(GaCl_3_ aqueous solution + water) + (TMAH (CH_3_)_4_NOH) aqueous solution (25% *w*/*w*))	-pH was adjusted to 5, 7 and 9 by adding TMAH gradually, shaken for 5 min, ✓ Reaction at 60 °C (Aging time: pH-5: 5–7 days, pH-7: 1 day and pH-9: 2 days) or ✓ Reaction at 160 °C for 2 h. Calcination: 500 and 900 °C for 2 h				Uniform submicron particles of different shapes (rhombic rods, rhombic prisms, hierarchical structures) of α-Ga_2_O_3_ at 500 °C and β-Ga_2_O_3_ at 900 °C		[43]
hydrothermal method		Gallium metal—0.2 g + DI water—60 mL	Reaction in autoclave at 160 °C for 12 h, precipitate collected by centrifugation then dried at 70 °C, Calcination: Pre heat at 400 °C for 5 h then at 600–800 °C for 1.5 h				Rod-like morphology mixed phase of α-Ga_2_O_3_ and β-Ga_2_O_3_ at 700 °C	Photocatalyst	[202]
hydrothermal method		(((Ga(NO_3_)_3_ + water)—0.012 mol/L) + NaOH solution (1.5 M) + 0.036 mol NaOH), kept in shaking bath (100 rmp) at 80 °C for 2 h	PEO or CTAB (0.0048 mol) added		Stirring at RT for 2 h. Reaction at 100 °C for 48 h. Dried at 80 °C	Calcination: 900 °C for 2 h.	PEO-β-Ga_2_O_3_ quadrilateral rods length—2.56 µm, CTAB-β-Ga_2_O_3_ quadrilateral prisms length—2.56 µm CTAB sample has larger pore size than PEO sample_._		[203]
		((0.003 mol Ga(NO_3_)_3_ + water (10 mL)) + (0.015 mol NaNO_3_ + water—10 mL)), kept in shaking bath (100 rmp) at 80 °C for 2 h	PEO or CTAB (0.0025 mol) added	HNO_3_ added dropwise at 80 °C to adjust pH = 9.5		No calcination	Amorphous agglomerates and nanotubes of γ$-{\mathrm{Ga}}_{2}O_{3}\;(\mathrm{length}\;\leq \;60\;\mathrm{nm},\;dia_{ext}$ = 0.8–3 nm), More number of nanotubes with PEO than CTAB		
hydrothermal method		((Ga(NO_3_)_3_·nH_2_O (A) + DI water)-0.01 mol/L) + SDBS (B), C_m_ ratio: ($\frac{B}{A}$ = $\frac{1}{10}$)	Reaction in autoclave at 140 °C for 10 h, Products separated, washed with DI water then dried in atmosphere ambient, Calcination: 600 °C or 900 °C for 5 h				Brush-like particles composed with the nanowires		[204]
		((Ga(NO_3_)_3_.nH_2_O (A) + DI water)-0.01 mol/L) + SA (B), C_m_ ratio: ($\frac{B}{A}$ = $\frac{1}{10}$)					Cuboid-like particles		
		((Ga(NO_3_)_3_·nH_2_O + DI water)— 0.01 mol/L)					Spindle like particles		
hydrothermal method		(((Ga(NO_3_)_3_·xH_2_O + DI water—8 mL +NaOH—207 mg) + oleic acid—1.68 mL) + 1.5 mL of oleic acid + 6 mL of ethanol) + Na_2_S·9H_2_O—96 mg	Reaction in autoclave at 140 °C for 4, 6, 8 and 16 h. Products collected, washed then dried under vacuum at 70 °C for 8 h. Calcination: 1000 °C for 10 h				β-Ga_2_O_3_ microspheres with hollow interior		[205]
hydrothermal method		(50 mL of Ga(NO_3_)_3_—0.0508 mol/L + 50 mL of oxalic acid (C_2_H_2_O_4_) (0.155–0.666 mol/L)) using water as solvent	Rigorous stirring on a hot plate at 90 °C, Reaction in autoclave at temperatures varying from (175–225 °C) for 10 h. Calcination: 450 °C for 3 h				Calcination of GaOOH prepared at reaction temp 200 °C were resulted into α-Ga_2_O_3_ microspheres	Photocatalyst	[206]
hydrothermal method		(Ga(NO_3_)_3_·xH_2_O—0.01 g) + water—20 mL + C_3_H_7_NO—5 mL + C_8_H_16_N_2_O ([Bmin][OH])—0.05 g	Precipitates collected, washed and dried at 60 °C, Reaction in autoclave at 180 °C for 24 h. Calcination: 450 °C for 3 h				Mesomorphs α-Ga_2_O_3_ hierarchical structures	Photo degradation	[207]
hydrothermal method		(Ga(NO_3_)_3_—2.55 g (10 mmol)) + (biuret—6.18 g (60 mmol)), Each solution prepared using 50 mL of water as solvent	Heated up to boiling, then stirred for 30 min, Reaction in autoclave at 200 °C for 10 h, dried at 120 °C for 2 h, Calcination: 450 °C for 3 h				Mesoporous α-Ga_2_O_3_ microrods		[208]
		(Ga(NO_3_)_3_—2.55 g (10 mmol)) + (oxalic acid—7.56 g (60 mmol)), Each solution prepared using 50 mL of water as solvent	Solution heated up to boiling, then stirred for 30 min, Reaction in autoclave at 200 °C for 14 h, dried at 120 °C for 2 h, Calcination: 450 °C for 3 h				Mesoporous α-Ga_2_O_3_ micro flowers		
hydrothermal method	Carbon spheres as templates	(Ga(NO_3_)_3_·xH_2_O—1.3544 g (6 mmol) + urea—1.8 g (30 mmol) + carbon colloid solution	Reaction in autoclave at 90 °C for 48 h, Dried at RT for 24 h, Calcination: 500 °C (1 °C/min), 600–800 °C (10 °C/min) for 1 h				Uniform β-Ga_2_O_3_ hollow nanostructures at 700 °C.		[209]
		Ga(NO_3_)_3_·xH_2_O—15 mL (0.3 mol/L) + (urea—3.5 g + DI water—15 mL)	Urea solution was kept 90 °C for 1 h then added to solution, Reaction at 140 °C for 1, 3 and 10 h, solution centrifugated, precipitates collected then washed and dried, Calcination: 900 °C for 3 h				β-Ga_2_O_3_ nanorods		[210]
Micro-wave hydrothermal method		Ga(NO_3_)_3_—10 mL (0.166 mol/L) + (urea (2.0, 2.7, 3.4, 4.1 g) + DI water—10 mL)	Solution stirred at 60 °C for 30 min, Heated to 100 °C in 3 min @ 400 W then to 130–150 °C in 2 min @ 600W and maintained there for 1–4 min, Calcination: 300–700 °C for 2 h				Ultrafine γ-Ga_2_O_3_ nanocrystals at 140 °C for 2 min maintaining time, Phase change to β-Ga_2_O_3_ at 600 °C	Photo degradation	[211]
hydrothermal method		(Ga(NO_3_)_3_·9H_2_O—1.6 g + urea—2.64 g + PEG—20 mL + 70 mL—DI water)	Solution stirred vigorously for 2 h at 25±1 ℃, Reaction in autoclave at 140 °C for 6 h, Precipitates filtered, washed with ethanol, Calcination: 800 °C for 2 h				Mesoporous β-Ga_2_O_3_ nanorods	Photocatalyst	[212]
		(Ga(NO_3_)_3_·9H_2_O (0.01 mol) + urea (0.1 mol) + PEG—200) using DI water as solvent	Stirred at RT for 1 h, Reaction in autoclave at 160 °C for 8 h, Dried at 100 °C for 24 h, Products separated by centrifugation, washed with alcohol then dried at 100 °C for 24 h, Calcination: 800 °C for 10 h				β-Ga_2_O_3_ nanorods	Photocatalytic degradation	[213]
hydrothermal method		(Ga(NO_3_)_3_·nH_2_O + DI water)— 0.015 mol/L	Reaction in autoclave at 150 °C for 24 h, Solid products collected and washed and dried at RT, Calcination: 700 °C for 1 h.				Rod like morphology polycrystalline β-Ga_2_O_3_ films		[214]
		Ga(NO_3_)_3_·nH_2_O (0.1 M) + DI water—100 mL	NH_4_OH added to adjust solution pH-9, Reaction at RT to 95 °C for 5 h, Calcination: 500, 800 and 1000 °C for 3 h				α-Ga_2_O_3_ nanorods at 500 °C, β-Ga_2_O_3_ nanorods at 800 and 1000 °C	Photocatalytic degradation	[21]
		Ga(NO_3_)_3_·xH_2_O (0.1M) + DI water—50 mL	NH_4_OH- (28−30% NH_3_ in solution) added to make pH-10, Stirred at 60 °C while aging for 10 min–6 h, Reaction at 140 °C for 10 h, dried at 70 °C for 6 h, Calcination: 1000 °C for 5 h				β-Ga_2_O_3_ nanorods	FET	[215]
		Ga(NO_3_)_3_—1.02292 g (0.05 M) + DI water—80 mL	NH_4_OH solution added till pH-9, Reaction in autoclave at 95 °C for 5 h, Precipitates separated by centrifugation then dispersed in 5 mL DI water and dried at 75 °C, Calcination: 150 °C, 400–1000 °C in ambient air for 5 h				α-Ga_2_O_3_ nanorods at 400–700 °C, β-Ga_2_O_3_ nanorods at 900 °C and 1000 °C	CO_2_ gas sensing	[216]
		((Ga(NO_3_)_3_·xH_2_O (0.05 M) + DI water)—160 mL)	Various amounts (1, 1.5, 2, 3 and 6 mL) of NH_4_OH was added to get different pH = 5, 7, 9, 11 and 14, Reaction at 100 °C for 5 h, dried overnight at 75 °C, Calcination: 1000 °C for 5 h				β-Ga_2_O_3_ nano powder	NH_3_ gas sensor	[217]
		Ga(NO_3_)_3_·9H_2_O—15 mL (0.3 moL/L)	NH_4_OH added dropwise till pH-10, Reaction at 40, 80, 120 and 160 °C for 18 h, Centrifugated, washed and dried, Calcination: 900 °C for 3 h				β-Ga_2_O_3_ microspheres		[218]
hydrothermal method	Sputter coated 50 nm thick SnO_2_ seed layer on 1 µm SiO_2_/Si (100)	Ga(NO_3_)_3_·9H_2_O—0.6 g + DI water—40 mL	Seed layer annealed at 900 °C for 2 h, Solution pH-2, Substrates incubated in solution at 150 °C for 12 h in autoclave, Calcination: 1000 °C for 4 h				β-Ga_2_O_3_ nanorod arrays (NRAs)	CO gas detection	[219]
	Sputter coated 50 nm thick SnO_2_ seed layer on 1 µm SiO_2_/Si (100)	Ga(NO_3_)_3_·9H_2_O—0.6 g + DI water—40 mL	Seed layer annealed at 900 °C for 2 h, Solution pH-2, Reaction at 150 °C for 12 h, dried overnight at 80 °C Calcination: 1000 °C for 4 h				β-Ga_2_O_3_ nanorod arrays (NRAs)	NO_2_ gas sensor	[220]
	Si (100)	Nucleation (solution-1): Ga(NO_3_)_3_·9H_2_O—0.2 M + ethanol—10 mL + DI water—30 mL. Crystal Growth (solution-2): Ga(NO_3_)_3_·9H_2_O—0.2 M + DI water—30 mL.	SiO_2_ oxide layer eliminated by etching with HF, Nucleation: Reaction in solution-1 at 100 °C for 30 min, Growth: Reaction in solution-2 at 150 °C for 12 h, dried at 50 °C for 60 min, Calcination: 800 °C for 4 h				β-Ga_2_O_3_ micro rod arrays (MRAs)		[221]
	Si (100)	Nucleation (solution-1): Ga(NO_3_)_3_·9H_2_O—0.2 M + ethanol—10 mL + DDI water—30 mL. Crystal Growth (solution-2): Ga(NO_3_)_3_·9H_2_O—0.2 M + DDI water—30 mL.	SiO_2_ oxide layer eliminated by etching with HF, Nucleation: Reaction in solution-1 at 100 °C for 60 min, Growth: Reaction in solution-2 at 150 °C for 12 h, dried at 70 °C for 2 h, Calcination: 800 °C for 4 h				β-Ga_2_O_3_ nanoflakes	β-Ga_2_O_3_/p-Si hetero junction self-powered photodiode	[222]
	FTO glass	(Ga(NO_3_)_3_—0.3 g + DI water)—30 mL	Reaction at 150 °C for 12 h, Calcination: Pre-annealed at 400 °C for 4 h then annealed at 700 °C for various times or annealed from 770–830 °C for 20 min				α-Ga_2_O_3_ NRAs and α/β-Ga_2_O_3_ phase junction NRAs	Solid-state type photodetector, photoelectron-chemical type photodetector	[223]
	FTO glass	(Ga(NO_3_)_3_·9H_2_O—0.3 g + DI water—30 mL)—0.0239 M	Reaction at 150 °C for a night, dried at 80 °C, Calcination: Pre-annealed 400 °C for 4 then annealed at 700 °C for 20 min				α/β-Ga_2_O_3_ phase junction NRAs	Self-powered solar-blind photodetector	[224]
	A layer of SnO_2_ on the surface of the FTO glass	(Ga(NO_3_)_3_ + DI water)—0.39 mol/L	Hydrothermal reaction at 150 °C for 12 h, dried Calcination: 400 °C for 4 h				Hexagonal prism like α-Ga_2_O_3_ NRA, average length—1.62 µm and average diameter 80–200 nm	Self-powered spectrum-distinguish- able α-Ga_2_O_3_ NRA/Cu_2_O microsphere (MS) p-n junction electro-chemical photodetector	[225]
	Spin coated Ga_2_O_3_ seed layered FTO glass	(Ga(NO_3_)_3_·9H_2_O—0.3 g + DI water—30 mL)—0.0239 M	Seed layer annealed at 450 °C for 30 min, Reaction at 150 °C for 12 h, washed with DI water then dried at 80 °C, Calcination: 700 °C for 4 h				β-Ga_2_O_3_ NRAs	Solar-blind deep UV photodetector	[226]
	Spin coated Ga_2_O_3_ seed layer on FTO substrate	Ga(NO_3_)_3_·9H_2_O—30 mL (0.03 M) + hexamine (C_6_H_12_N_4_) (0.005 M)	Reaction at 180 °C for 12 h, Calcination: 500 °C for 4 h				α-Ga_2_O_3_ nanorod arrays (NRAs)		[227]
	GaOOH NRAs prepared by hydrothermal method as above	Al(NO_3_)_3_·9H_2_O (0.005 M, 0.01 M, 0.015 M) + C_6_H_12_N_4_ (0.005 M)					α-Ga_2_O_3_ nanorod was completely covered by γ-Al_2_O_3_	Ga_2_O_3_-Al_2_O_3_ heterojunction PEC Self-powered UV detector	
hydrothermal method		((Ga(NO_3_)_3_·xH_2_O (0.1 M) + DI water—50 mL) + SnCl_4_—0, 0.026, 0.052, 0.130 and 0.260 g)	NH_4_OH (28% in H_2_O) added to solution at 60 °C to till pH-10, Reaction at 140 °C for 10 h, dried at 70 °C for 6 h, Calcination:1000 °C in O_2_ ambient for 6 h				β-Ga_2_O_3_ nanostructures	Photocatalyst	[228]
		Ga_2_(NO_3_)_3_·xH_2_O (0.05 M) + DI water—80 mL + SnCl_2_·2H_2_O (0 mol%, 2 mol%, 4 mol%)	NH_4_OH added to attain pH-7, Reaction at 100 °C for 5 h, dried overnight at 75 °C, Calcination: 1000 °C for 5 h				β-Ga_2_O_3_ nano powder	NH_3_ gas sensor	[229]
hydrothermal method		(Al(NO_3_)_3_·9H_2_O—2.01 g + Ga(NO_3_)_3_·9H_2_O—0.11 g + DI water—40 mL) + methenamine (HMT)-3.2 g	Solution stirred ultrasonically for 1 h, Reaction at 180 °C for 9 h, Calcination: 550 °C for 3 h				Ga_2_O_3_/Al_2_O_3_ composite materials	NO_x_ gas sensing	[230]
		Ga(NO_3_)_3_·xH_2_O (0.1 M) + DI water—50 mL + Al (NO_3_)_3_·9H_2_O (0.02, 0.064, 0.110 and 0.210 g)	NH_4_OH added dropwise till pH-10.34, Reaction at 140 °C for 10 h, dried at 70 °C for 6 h, Calcination: 1000 °C for 6 h in O_2_ ambient				β-Ga_2_O_3_ nanostructures with spindle like morphology to microrod structure	Photocatalyst	[231]

**Table 5 nanomaterials-12-03601-t005:** Chemical bath synthesis of Ga_2_O_3_.

Method	Substrate/Template	Precursor	Synthesis Conditions	Properties	Applications	Ref.
Chemical bath method		$[((({\mathrm{Ga}}_{2}O_{3}+{\mathrm{Dy}}_{2}O_{3}+\mathrm{Dilute}\;\mathrm{HCl})+\mathrm{DI}\;\mathrm{water})-500\;\mathrm{mL}\;(0.015\;\mathrm{mol}/L))+\mathrm{urea}\;(0.025\;\mathrm{mol})+{\mathrm{OH}}^{-}$source]—0.5 mol/L	Stirring at RT for 2 h, then to 90 °C in 1 h, films deposition at 90 °C for 4 h, precipitates centrifugated, washed then dried at 90 °C for 24 h. Calcination: 1000 °C for 4 h	At pH-4, submicrospindles.At pH-9, Self-assembled nanoparticle hierarchical microspheres.		[248]
Chemical bath method	Quartz glass, β-Ga_2_O_3_/Quartz (70 nm)	(Ga(NO_3_)_3_·xH_2_O + DI water)—0.015 mol/L	Substrates put in solution bottles, sealed and kept in oven at 60 °C for 1–48 h, Calcination: 600–900 °C for 1 h	A small number of heterogeneous nucleation GaOOH precipitates		[214]
	SnO_2_, TiO_2_, MgO coated Quartz glass (Thickness: 70 nm), and FTO glass			The rod like particles of GaOOH were build-up closely and vertically on substrates		
Chemical bath method	SnO_2_ coated quartz glass	(Ga(NO_3_)_3_·nH_2_O + Eu(CH_3_COO)_3_·4H_2_O+ DI water)—0.015 mol/L + (urea—0.5 mol/L) concentrations of Eu^3+^ dopants against Ga^3+^ (1, 3, 5 and 10 at.%)	Substrates put in solution bottles, sealed and kept in oven at 90 °C for 24 h, Calcination: 900 °C for 1 h	β-Ga_2_O_3_:Eu^3+^ oriented along [111] perpendicular to substrate		[249]
Chemical bath method	Glass	Ga(NO_3_)_3_·nH_2_O (0.025 M, 0.05 M, 0.075 M) + HMT (0.5 M) + DI water —1 L	Solution under stirring while reaction at 95 °C for 5 h, films dried at 70 °C for 5 min, Calcination: 400 °C, 500 °C, or 600 °C for 3 h	α-Ga_2_O_3_ film had high crystallinity at 500 °C	Photodegradation	[250]
Chemical bath method	Si (001)	(Ga(NO_3_)_3_·xH_2_O + ultra-pure water)—(0.015–0.1 M)	Solution stirred at 70 °C for 12 h, Films deposited at 70 °C for 24 h, Films washed and dried under N_2_, Calcination: 900 °C for 1 h	3 types of rod-like morphologies of β-Ga_2_O_3_ formed directly on Si		[251]
Chemical bath method	ITO/glass (E-beam deposition) -200 nm	Ga(NO_3_)_3_·$H_{2}O\;(A)+\mathrm{HMT}\;(B)+\mathrm{DI}\;\mathrm{water},\;C_{m}\;\mathrm{ratio}:\;(\frac{A}{B}=0.5,\;1,\;2)$	A seed layer deposition first 10, 20 and 30 min at 95 °C, Deposition at 95 °C for 2 h, Calcination: 400–600 °C for 1 h	α-Ga_2_O_3_ film of thickness 3.5 µm, Film’s crystallinity is high for C_m_ = 1:1	pH sensor	[252]

**Table 6 nanomaterials-12-03601-t006:** Other methods to synthesize Ga_2_O_3_.

Method	Substrate/Template	Precursor	Synthesis Conditions	Properties	Applications	Ref.
Solvothermal		GaCl_3_ aqueous solution (5.79 mmol) + Eu(NO)_3_.6H_2_O (4.06 × 10^−2^ mmol) + Tb(NO)_3_.6H_2_O (2.32 × 10^−2^ mmol) + (ethylene glycol (EG) and water mixture—2:3)	NaOH (1 mol/L) added dropwise till pH = 10, stirred for 2 h, Reaction in autoclave at 200 °C for 4 h, Products separated by centrifugation then washed and dried at 100 °C in vacuum for 4 h.	PL spectra: Un doped β-Ga_2_O_3_—Bright blue emission, 0.01% Tb ^3+^ doped—Green, 0.01% Eu ^3+^ doped—Red, Co-doped—White		[253]
Solvothermal	Si	(Ga metal + HF) + (Solvent: mixture of ethylenediamine (En) and water (volume ratio—80:20, 60:40, 50:50, 40:60 and 30:70))	Selected amount of precipitate and solvent then stirred for 1 h, Films deposition in autoclave at 200 °C for 4 h, films washed then dried at 100 °C for 1 h	50:50 sample—2D interconnected nanoflakes structure of flake thickness of 15 nm, 60:40 sample—3D microstructures of order 0.4 µm diameter		[254]
Solvothermal		(Gallium nitrate—0.65 g) + ethanol—50 mL) + (Sodium acetate- 0.6 g + ethanol—50 mL)	Reaction in autoclave at 200 °C for 5h and 10 h, washed with DI water and ethanol then dried in oven at 120 °C for 2 h, No calcination	5 h—GaOOH morphology was incomplete, 10 h—GaOOH semi nanosphere morphology (size: 100–500 nm)		[255]
Solvothermal		(Ga(NO_3_)_3_·nH_2_O—0.5 g + HCl—mL + H_2_O) + ethylene glycol—4.5 mL, (water to EG ratio-2:3)	pH of solution = 10, Reaction in autoclave at 195 °C for 6 h, Precipitates washed with water and ethanol then dried at 80 °C	Ga_2_O_3_ nanoparticles size range: 5–10 nm, liquid metal/metal oxide (LM/MO) frameworks with Ga_2_O_3_ via ultrasonication	Photocatalyst	[256]
Forced hydrolysis		$({\mathrm{Ga}}_{2}({\mathrm{SO}}_{4}{)}_{3}\cdot {\mathrm{xH}}_{2}O+\mathrm{Ga}({\mathrm{NO}}_{3}{)}_{3}\cdot {\mathrm{xH}}_{2}O)\text{—}0.01\;M+\mathrm{urea}\text{—}0.1\;M+\mathrm{DI}\;\mathrm{water}\text{—}100\;\mathrm{mL},\;\mathrm{Concentration}\;\mathrm{ratio}:\;R=SO_{4}^{2-}/NO_{3}^{-}=0–\infty$	Solution mixture aged in oil bath at 98 ± 1 °C for 2 h, Precipitates separated by centrifugation and washed then freeze dried at −110 °C, Calcination: 500–1000 °C for 1 h	Uniform polycrystalline β-Ga_2_O_3_ Ns (diameter 200 nm) by calcination at 1000 °C ^@^ R = 0.33		[257]
Forced hydrolysis		$({\mathrm{Ga}}_{2}({\mathrm{SO}}_{4}{)}_{3}\cdot {\mathrm{xH}}_{2}O+\mathrm{Ga}({\mathrm{NO}}_{3}{)}_{3}\cdot {\mathrm{xH}}_{2}O)\text{—}0.01\;M+\mathrm{urea}\text{—}0.1\;M+\mathrm{DI}\;\mathrm{water}\text{—}100\;\mathrm{mL},\;\mathrm{Concentration}\;\mathrm{ratio}:\;R=SO_{4}^{2-}/NO_{3}^{-}=0.33$	Solution mixture aged in oil bath at 98 ± 1 °C for 2–10 h, Precipitates separated by centrifugation and washed then freeze dried at −110 °C, Calcination: 500−1000 °C for 1 h	Aging time: 2 h—NSs morphology, 10 h- microrods morphology, CL spectra: β-Ga_2_O_3_ NSs—UV blue emission peak at 375 nm, β-Ga_2_O_3_ microrods—Strong blue emission peak at 416 nm		[258]
Forced hydrolysis		$(\mathrm{Ga}({\mathrm{NO}}_{3}{)}_{3}\cdot {\mathrm{xH}}_{2}O\;(A\text{—}0.03\;\mathrm{mol}/L)+\mathrm{urea}\;(B\text{—}0.18\;\mathrm{mol}/L)+\mathrm{DI}\;\mathrm{water})+\mathrm{DI}\;\mathrm{water}+{\mathrm{NaNO}}_{3}(C),\;C_{m}\;\mathrm{ratio}:\;(\frac{A}{B}=\frac{1}{6})$, $\mathrm{Ionic}\;\mathrm{strength}\;\mathrm{ratio}\;(R=\frac{I_{A}}{I_{C}})$ = ∞, 3, 2, 1, 0.5, 0.3, 0.2	Hydrolysis at 90 °C for 9 h, Products centrifuged and washed, Calcination: 450 °C in vacuum	Morphology: At R = ∞—spindle like nanorods having two narrow sides and wide center, R = 1 —dumbbell shape with wide sides and narrow center, R = 0.2—Ga_2_O_3_ broken into fragments		[259]
Reflux condensation		Ga(NO_3_)_3_·xH_2_O—0.01 M + DI water—30 mL + urea—0.1 M	Solution refluxed with continuous stirring at 90 °C for 12 h, Precipitate centrifuged and washed then dried at 100 °C, Calcination: 500 °C and 900 °C for 3 h	Self-assembled pattern of β-Ga_2_O_3_ nanorods	Photocatalyst	[260]
Reflux condensation		Ga(NO_3_)_3_·xH_2_O (A)—0.01 M + DI water—30 min + CTAB (B)—stoichiometric amount) C_m_ ratio: ($\frac{B}{A}$ = $\frac{1}{2}$,$\;\frac{5}{1}$, $\frac{50}{1}$)	Solution under stirring for 30 min, solution refluxed with continuous stirring at 90 °C for 12 h, precipitate washed and dried at 100 °C, Calcination: 900 °C for 3 h	At optimal C_m_ ratio = 5/1, uniform and side to side aligned of β-Ga_2_O_3_ nanorods (length = 200 nm, diameter = 50 nm)	Photocatalyst	[261]
Electro-deposition	FTO glass	Ga_2_(SO_4_)_3_—20 mM + H_2_O_2_ (9.79 M H_2_O_2_ solution)—0.13 mL	Applied potential: −1.0 to −1.2, Cathode: FTO glass, Cathode: Pt Deposition time: 2–10 min, Calcination: 500–600 °C	At −1.0 and −1.1 V, deposition rate = 30–60 nm/min, At −1.2 V, deposition rate = 1 µm/min		[262]
Electro-deposition	Si (100)	(Ga_2_O_3_ (0.1 M, 0.5 M, 1.M) + HCl—1.5 mL) + DI water—6.5 mL	NH_4_OH added to vary the pH = 4–10, deposition time = 2 h, Cathode: Pt wire current density = 0.15 A/cm^2^	At pH = 4: High density nanodots at 0.1 M, High density nanorods at 1.0 M		[263]

## Data Availability

The data that support the findings of this study are available from the corresponding author upon reasonable request.

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
