# Peer review of "A Review on Gallium Oxide Materials from Solution Processes"

_nanomaterials, 2022, doi:10.3390/nano12203601_

Round 1

Reviewer 1 Report

In this review article, Authors have summarized the solution-based methods for fabrication of gallium oxide nanostructures and thin films. The article will serve as a good reference for the scientific community who want to prepare gallium oxide thin films and materials by solution-based approaches. Reviewer supports the publication with following recommendations.

·         In introduction authors should highlight how their review article is different from other reviews on gallium oxide materials which partially covers the topic

o   https://www.mdpi.com/2079-4991/12/12/2061

o   https://www.intechopen.com/chapters/58433

o   https://onlinelibrary.wiley.com/doi/full/10.1002/nano.202100149

·         In fig. 32 results are from a method that is from VPT, the original publication used liquid gallium metal and tube CVD like setup, can author explain how it fits as a solution-based method and their review

·         Qualities of the thin films and nanostructure from solution remains inferior to the ones obtained by vacuum-based methods and at present will be impractical to be used in quality heterostructure devices and high-power applications. Author could add following sections or comments in their review about  

o   Comparison between qualities of the thin films and nano structures form solutions and vacuum based methods.

o   Limitations in application of the materials and thin films obtained by solution-based methods

Also, authors can improve the story flow in section 4 and 5 of the article, as the paragraphs appears more like an independent summary of reference articles rather than making transition from one paragraph to another, which would be more appealing to their readers.

Reviewer 2 Report

The manuscript is written well with relevant references and data. There are several questions to answer to achieve the confidence from the reviewer. Here are the comments.

1. The author should add the variability of density of states with different process conditions.

2. Is this material suitable for memory applications?

3. The sensor applications using Ga2O3 are reliable with reproducible? please provide such information in the revised manuscript.

Reviewer 3 Report

The article is a review of the current state of research in the field of nanomaterials based on gallium oxide, in particular, special attention is paid to chemical methods of synthesis. A large amount of data on experimental work, synthesis methods, as well as possible practical applications is given. Special attention should be paid to the tables, which summarize the main results for the relevant sections of the article in a short form.

As a note, I will note a large number of typos and inaccuracies in all sections, for example, table 1 (Space group for the hexagonal phase); lines 98, 212-213, 300 (semiconductors such as sapphire?), 907-908, 1706, 1718 (units?) and many many others. The article should be carefully read again by authors.

Despite the remarks, this review is of great scientific and practical interest, and can be published after correcting minor comments.

Reviewer 4 Report

The review article under consideration provides a solid basis in the field of gallium oxide structures fabrication by the wet (or solution) chemistry methods. It also gives a rather comprehensive and update information on gallium oxide phases with their properties as well as on applications of the gallium oxide structures in UV detectors, gas and pH sensors, and as photocatalysts. 

Some issues, however, need to be clarified before accepting the article.

1. Section 3, which precedes the example of fabricated gallium oxide structures, provides a general description of synthesis methods for all possible materials, and not specifically for gallium oxide-based ones. I believe that the review can become more informative if section 3 will have materials presenting the specific features of the synthesis of gallium oxide by these methods.

2. There are no figures illustrating gallium oxide structures processed via sol-gel methods. Perhaps such Figures can be added to Section 4.1.

3. It could be good to make Figure captions more informative, e.g. indicating for each case the technique of sample fabrication.

4. The first paragraph in Section 5 just repeats the information presented earlier in the text. This paragraph should be better omitted.

There are also some technical minors that should be checked and fixed. For example, sentences in lines 49-51 and 1346-1347 have no verbs; line 98 shows an error in referencing; the meaning of sentences in lines 1377-1379 is not clear.  

I recommend publishing this review article, considering the above edits.
